# Biological carbon pump estimate based on multidecadal hydrographic data

Wei-Lei Wang[1✉], Weiwei Fu[2,5], Frédéric A. C. Le Moigne[3], Robert T. Letscher[4], Yi Liu[2,6], Jin-Ming Tang[1] & François W. Primeau[2✉]

The transfer of photosynthetically produced organic carbon from surface to mesopelagic waters draws carbon dioxide from the atmosphere[1]. However, current observation-based estimates disagree on the strength of this biological carbon pump (BCP)[2]. Earth system models (ESMs) also exhibit a large spread of BCP estimates, indicating limited representations of the known carbon export pathways[3]. Here we use several decades of hydrographic observations to produce a top-down estimate of the strength of the BCP with an inverse biogeochemical model that implicitly accounts for all known export pathways. Our estimate of total organic carbon (TOC) export at 73.4 m (model euphotic zone depth) is $15.00 \pm 1.12$ Pg C year$^{-1}$, with only two-thirds reaching 100 m depth owing to rapid remineralization of organic matter in the upper water column. Partitioned by sequestration time below the euphotic zone, $\tau$, the globally integrated organic carbon production rate with $\tau > 3$ months is $11.09 \pm 1.02$ Pg C year$^{-1}$, dropping to $8.25 \pm 0.30$ Pg C year$^{-1}$ for $\tau > 1$ year, with 81% contributed by the non-advective-diffusive vertical flux owing to sinking particles and vertically migrating zooplankton. Nevertheless, export of organic carbon by mixing and other fluid transport of dissolved matter and suspended particles remains regionally important for meeting the respiratory carbon demand. Furthermore, the temperature dependence of the sequestration efficiency inferred from our inversion suggests that future global warming may intensify the recycling of organic matter in the upper ocean, potentially weakening the BCP.

The downward flux of biologically produced organic carbon draws $CO_2$ out of the atmosphere, contributing to the maintenance of a vertical gradient of dissolved inorganic carbon (DIC) in the ocean[4]. Much of the primary production occurring in sunlit waters is respired in surface waters without greatly affecting the partitioning of $CO_2$ between the atmosphere and ocean[2]. Attention, therefore, focuses on the fraction of the net primary production (NPP) exported to deeper waters before being respired. Considerable effort has focused on discovering the processes responsible for the regional differences in the so-called ef-ratio[2,5-7], defined as export, or new production, divided by NPP (e-ratio and f-ratio, respectively). Oceanographers rely on empirical relationships between the ef-ratio and satellite-based measurements of NPP and sea surface temperature (SST) to obtain global-scale export patterns. Unfortunately, different versions of these empirical relationships, which typically assume that the ef-ratio is positively correlated with NPP and negatively correlated with SST, produce globally integrated estimates of carbon export that can vary by as much as a factor of three (5–12 Pg C year$^{-1}$), although part of the spread may be caused by the different choices of export depth and data-coverage issues[2,8]. Moreover, several field observations in highly productive regions such as the Southern Ocean often contradict the assumption that the ef-ratio is positively correlated with NPP (refs. 9–13).

One cause for these discrepancies is that most observations provide only snapshots of the ocean at the time of collection, whereas episodic signals may be missed in models. Another explanation is that empirical algorithms focus almost entirely on the contribution from sinking particles, neglecting possibly important contributions from vertically migrating zooplankton and the transport of dissolved and non-sinking particulate organic carbon (POC) by subducting and overturning water masses (also known as the particle injection pump)[1]. For instance, support for the importance of non-sinking particles is provided by the work of Emerson[14], who estimated annual net community production (ANCP) at three time-series sites (ALOHA, Hawaii Ocean Time-series station; BATS, Bermuda Atlantic Time-series Study; and OSP, Ocean Station Papa). He found that sinking POC flux is 3–4 times lower than required by mass-balance analyses. Indeed, Boyd et al.[1] suggest that non-gravitational export pathways acting on suspended particles can account for as much carbon export as the gravitational carbon pump, although the strength of these export pathways remains uncertain.

Regardless of how organic matter produced in surface waters is transferred to depth (gravitationally or not, in particulate form or not), most of it eventually remineralizes to inorganic carbon and nutrients, consuming dissolved oxygen ($O_2$) along the way. The resulting imprint on the dissolved oxygen, inorganic carbon, dissolved organic carbon

[1]State Key Laboratory of Marine Environmental Science, College of Ocean and Earth Sciences, Xiamen University, Xiamen, China. [2]Department of Earth System Science, University of California, Irvine, Irvine, CA, USA. [3]Univ Brest, CNRS, IRD, Ifremer, LEMAR, Plouzané, France. [4]Earth Sciences and Ocean Process Analysis Laboratory, University of New Hampshire, Durham, NH, USA. [5]Present address: Department of Atmospheric and Oceanic Science, Fudan University, Shanghai, China. [6]Present address: Department of Geosciences, Princeton University, Princeton, NJ, USA. ✉e-mail: weilei.wang@xmu.edu.cn; fprimeau@uci.edu

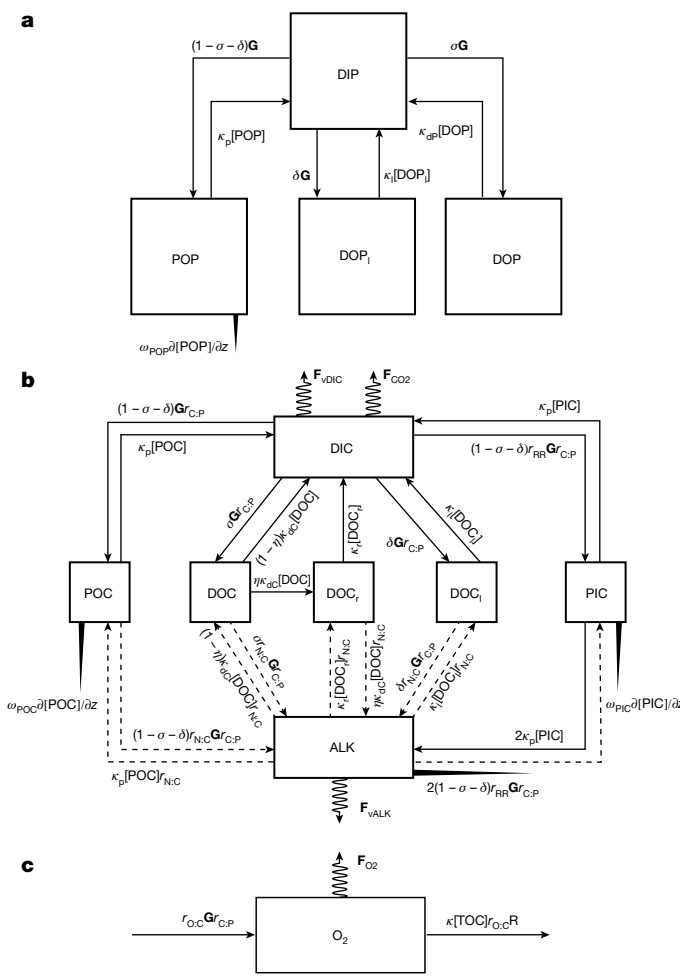

**Fig. 1 | Schematic representations of phosphorus, carbon and oxygen cycling in the ocean. a**, Phosphorus cycling. **b**, Carbon cycling. **c**, Oxygen cycling. The dissolved tracers, oxygen ($O_2$), DIP and DIC, DOP and DOC, and ALK are transported by advection and diffusion. POP and POC are transported vertically downward and remineralized in the water column. The downward particulate flux attenuation is modelled so as to produce a power-law depth dependence. PIC also sinks and dissolves in the water column. Its downward flux attenuation is modelled so as to produce an exponential depth dependence. DIC and $O_2$ experience sea-to-air gas fluxes in the surface ocean, which are represented by the coiled arrows above DIC and $O_2$, respectively ($F_{CO2}$ and $F_{O2}$). The DIC and ALK concentrations are influenced by evaporation and precipitation. Thus, a virtual flux (coiled arrows above DIC and below ALK; $F_{vDIC}$ and $F_{vALK}$, respectively) is applied to DIC and ALK to account for the concentrating and diluting effects of precipitation and evaporation. The solid lines in the schematic for the carbon-cycle model represent the pools that are connected by means of sink–source relationships. The dashed lines indicate the pools that are related by indirect source–sink relationships. For example, the formation of POC does not directly release ALK but instead changes the chemical form of nitrogen, which leads to changes in the ALK. See Methods for the symbol definitions and the Supplementary Information for their numerical values. $\kappa$[TOC] in the $O_2$ model represents the remineralization of TOC (see Methods for its full expression).

(DOC) and dissolved organic phosphate (DIP) concentrations, for which there exist global databases (GLODAPv2.2021 (ref. 15) and an updated DOC database[16]) collected over several decades of ship-based campaigns (Extended Data Figs. 1 and 2), allows us to infer the total carbon export and its regional variations. Here, using an inverse biogeochemical model for the cycling of phosphorus (P), carbon (C) and oxygen (O) (Fig. 1), we estimate the global distribution of the export flux separated into contributions from advective-diffusive flux, which encompasses fluxes mediated by physical transports such as the mixed-layer pump[17]

and the subduction pump[18], and DOC contribution to the biological pump[19,20], and non-advective-diffusive vertical flux that includes contributions from the gravitational pump[1], zooplankton migration pump[21] and seasonal lipid pump[22]. The model has 21 adjustable parameters whose values are constrained from global databases of DIP, total alkalinity (ALK), DIC, DOC and $O_2$ using a Bayesian inversion procedure.

After fitting the 21 parameters (Extended Data Table 1), the model captures most of the spatial variance in the three-dimensional distribution of DIP ($R^2 = 0.93$, $n = 76,480$), DIC ($R^2 = 0.94$, $n = 63,085$), ALK ($R^2 = 0.87$, $n = 59,093$), $O_2$ ($R^2 = 0.88$, $n = 83,732$) and total DOC ($R^2 = 0.80$, $n = 21,295$) (Extended Data Figs. 3 and 4). What distinguishes our model from previous inverse models[23,24] is the small number of adjustable parameters and the simultaneous use of several tracers to constrain the inversion. In particular, our inverse model uses DIC measurements, which provide the most natural constraint on the BCP. Previous inverse models did not use DIC observations to avoid the need to explicitly model the transient anthropogenic carbon signal in the hydrographic DIC dataset. Here we explicitly simulated the transient DIC signal and found that it contributes an approximately 20% decline in the vertical DIC gradient produced by the biological pump (Extended Data Fig. 5c; Methods). Furthermore, by combining ALK and DIC data with an accurate representation of the anthropogenic DIC signal, our model captures the respiration of organic carbon not oxidized by $O_2$ (ref. 25). In a sensitivity test in which we followed refs. 23,24 by using only $O_2$ and DOC to constrain the model, we found a substantial deterioration of the fits to other tracers (Extended Data Fig. 3). Our model results are further validated using deep-water POC fluxes measured using sediment traps at time-series stations and the ANCP estimated using several geochemical tracers (see text below). The model is able to match different satellite-based NPP products (CbPM and CAFE) by adjusting labile DOC production (Extended Data Fig. 6; Methods) without greatly affecting the goodness of fit to tracers or the estimated carbon fluxes with residence times greater than about 1 year.

## Organic carbon fluxes

In our model, which has a horizontal mesh resolution of 2° × 2° and 24 vertical layers, we define export according to the timescale for the vertical transfer of the organic carbon. Fluxes by fast-sinking POC (gravitational pump) and vertical zooplankton movements (vertical migration pump and seasonal lipid pump), which transport carbon vertically with no appreciable lateral transport, are assigned to non-advective-diffusive vertical export. Fluxes induced by organic carbon detrainment caused by changes of mixed-layer depth (mixed-layer-depth pump)[17] and physical subduction (subduction pump)[18] are assigned to advective-diffusive export. We note that, although the DOC pool of our model includes what would be characterized as suspended POC in field measurements and therefore is missing from the DOC measurement database, we believe that the difference is negligible for most of the ocean because the concentration of suspended POC is much lower (less than a few µM) than that of DOC (dozens of µM)[19]. We infer the strength and distribution of the total BCP from tracer distributions, which avoids counting the same export pathways several times[1].

Globally integrated, our estimated non-advective-diffusive vertical flux, which is calculated by integrating POC remineralization below 73.4 m, the euphotic zone depth of the model, is 10.63 ± 0.14 Pg C year$^{-1}$. For comparison, simulated export production in the Coupled Model Intercomparison Project Phase 5 (CMIP5) models ranged from approximately 4.5 to 7.5 Pg C year$^{-1}$ (ref. 26). The spread in the newer CMIP6 models is even larger, ranging from approximately 5 to 12 Pg C year$^{-1}$ at about 100 m (ref. 3). Our most probable estimate is almost triple that obtained from the $^{234}$Th method (4 Pg C year$^{-1}$)[2]. That our estimated export flux is larger than the $^{234}$Th-based estimate is not surprising because our flux includes not only the gravitational pump but also

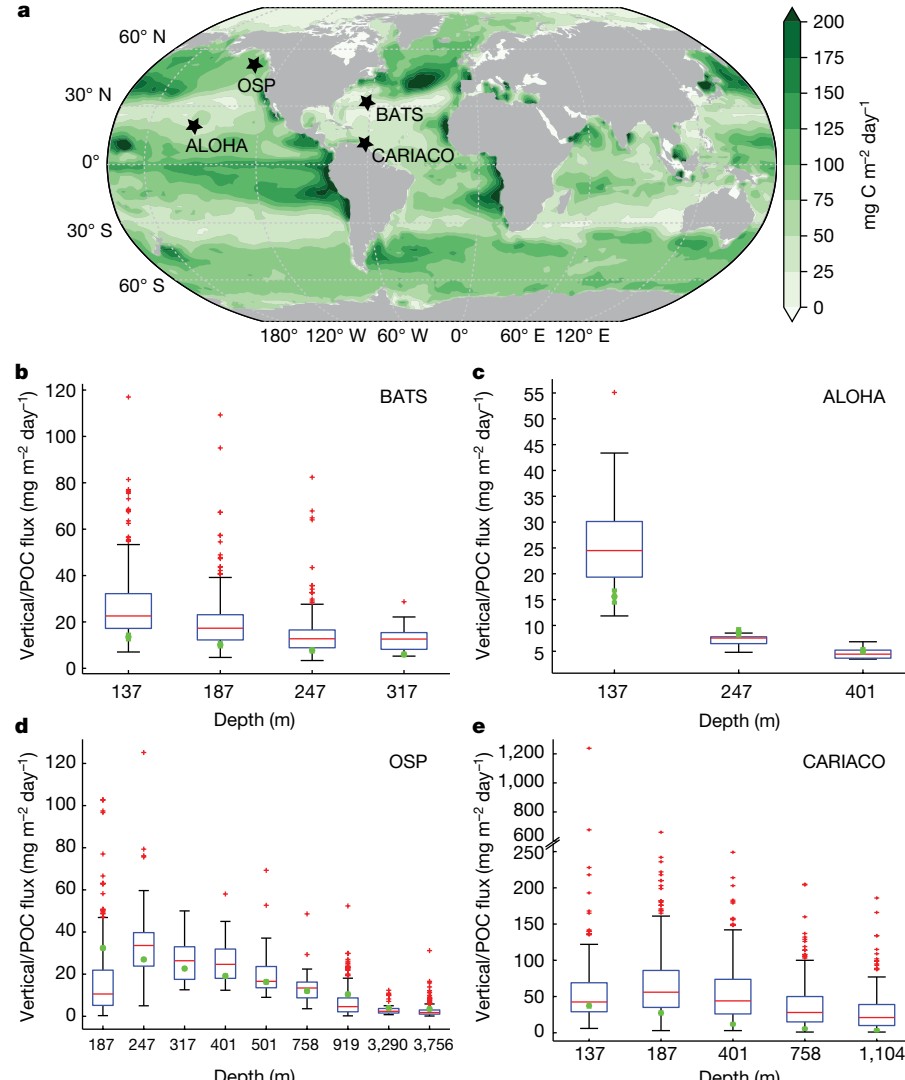

**Fig. 2 | Non-advective-diffusive vertical flux. a**, Contour plot of non-advective-diffusive vertical flux (mg C m⁻² day⁻¹) exiting the base of the euphotic zone, with the location of the four ocean stations (OSP, ALOHA, BATS and CARIACO) marked with black stars. **b**–**e**, Model-derived non-advective-diffusive vertical flux at different depths compared with trap-determined POC flux. The box plots represent sediment trap and the green circles are model predictions, with error bars representing ±1σ derived from different model configurations. The box plots summarize the distributions of in situ measurements of POC flux, which show the 25th, 50th and 75th percentiles binned according to the POC flux. The whiskers cover 99.3% of the data, with the remaining points shown as red crosses. In **b**–**e**, the sediment-trap data presented are multiyear collections covering a sampling period of 1988–2011 for the BATS station, 1988–2010 for the ALOHA station, 1987–2006 for the OSP and 1995–2012 for the CARIACO station. Because sediment traps are deployed in the water for several months, their measurements represent an average for a relatively extended period instead of a snapshot. The results of **a** are based on the CbPM NPP product and an e-folding remineralization time of 12 h for labile DOC.

fluxes mediated by zooplankton migration. By contrast, the ²³⁴Th method constrains only the flux of sinking POC.

Geographically, our estimated non-advective-diffusive vertical export rate is high in coastal upwelling regions, the Southern Ocean convergent zones, subpolar North Pacific and Atlantic oceans and low in the subtropics (Fig. 2 and Extended Data Fig. 7). The non-advective-diffusive vertical flux is consistent with measurements from deep-water sediment traps[27] at ocean stations ALOHA, OSP, BATS and CARIACO, in which extensive measurements exist (Fig. 2), even though such POC-flux measurements only partially include contributions from zooplankton migration (faecal pellets). The similarity may be because the migration pump is weak in oligotrophic oceans[24,28,29], in which ALOHA and BATS are located. For the mid-latitude OSP site, our estimates for the upper 200 m are higher than the median values of in situ sediment-trap measurements, probably because of the contribution from the migration pump. For the coastal CARIACO station,

the higher fluxes from sediment-trap measurements have several possible explanations. First, our model may not have adequate resolution. Second, the bias may be the result of blooms, which may be poorly represented in our climatological-mean model. Last, sediment traps may overestimate particle flux in coastal regions because of augmented 'statistical funnels' of particle collection[30] or catchment of large aggregates mediated by a range of physical and biological processes[1].

Our advective-diffusive export is calculated by tracking subsurface organic carbon respiration rates back to the base of the euphotic zone using an adjoint method[31]. The semi-labile and labile organic carbon fluxes are 1.67 ± 0.02 and 2.70 ± 1.04 Pg C year⁻¹, respectively, at 73.4 m. The export of refractory organic carbon (e-folding decay time about 5,500 and about 11,000 years in and below the euphotic zone, respectively) is two orders of magnitude lower than that of labile and semi-labile ones and, thus, is ignored in the following discussion. Our advective-diffusive flux of semi-labile organic carbon is close to

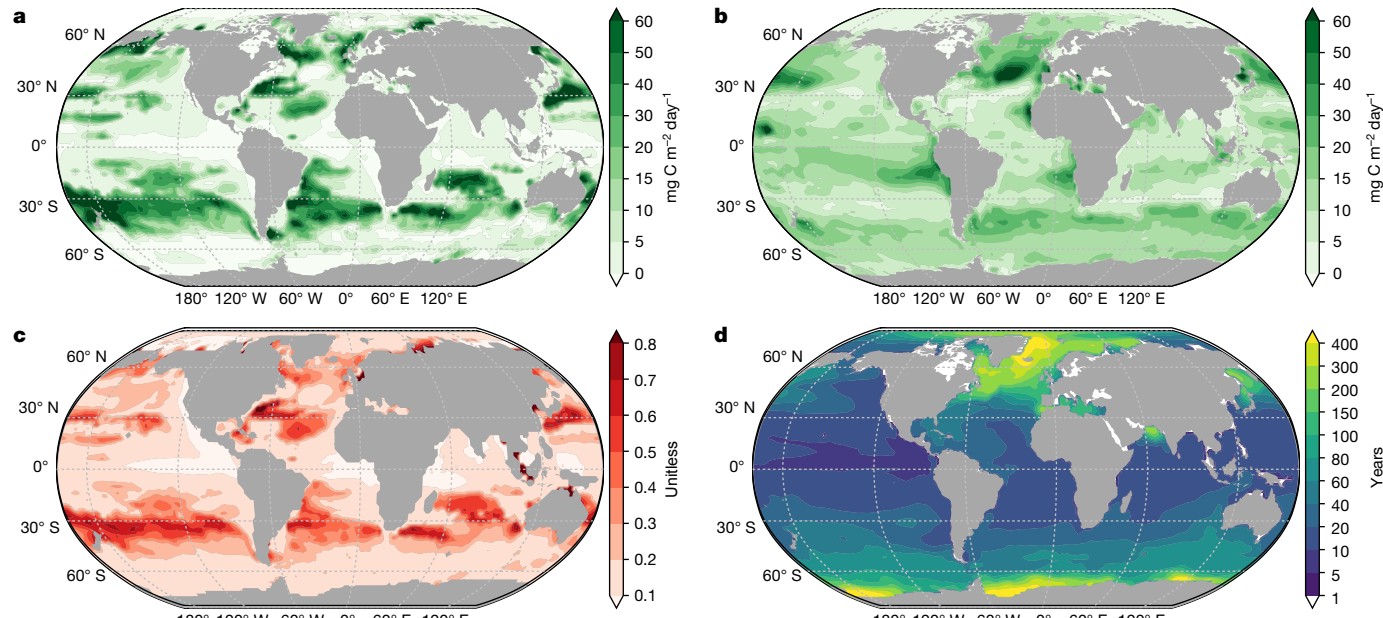

**Fig. 3 | Contour plots of advective-diffusive export flux at the base of the model euphotic zone. a**, Distribution of advective-diffusive flux by labile organic carbon (mg C m$^{-2}$ day$^{-1}$). **b**, Distribution of advective-diffusive flux by semi-labile organic carbon (mg C m$^{-2}$ day$^{-1}$). **c**, Distribution of the ratio of advective-diffusive flux to TOC flux. **d**, Distribution of DOC residence time in

years at the bottom of the euphotic zone. The residence time is defined as the time elapsed for DOC to be upwelled to the surface ocean following its export below the euphotic zone at that grid box. The results are based on the CbPM NPP product and an e-folding remineralization time of 12 h for labile DOC.

a previous estimate of 1.8 Pg C year$^{-1}$ at 100 m reference depth[19], but lower than the estimated 2.31 ± 0.6 Pg C year$^{-1}$ at the same depth of 73.4 m obtained from interpolated DOC observations and a circulation model[20]. When we include the export of labile organic carbon, our estimate surpasses any previous estimates. The previous estimate[20], which considered only one DOC pool, may have included signals from both labile and semi-labile organic carbon, explaining its intermediate value.

The labile and semi-labile organic carbon have distinct export patterns (Fig. 3a,b and Extended Data Fig. 7). Two factors contribute to this spatial pattern. One is the biological production pattern and the other is the spatially variable export efficiency. To explain the latter effect, we computed the mean DOC sequestration time for each water column in the model ('DOC sequestration time' in Methods). The mean of these residence-time distribution functions is contoured in Fig. 3d. For semi-labile organic carbon, the high export regions are in the Southern Ocean convergence zone, subarctic North Pacific and North Atlantic, with relatively long DOC residence time (Fig. 3b,d). These are important subduction and deep-water formation regions in which water masses are transferred from the mixed layer into the thermocline and deep ocean. For labile organic carbon, the high export regions (Fig. 3a) are located in the periphery of where it is produced, for example, in the subtropical gyres. However, there is no apparent export in the equatorial oceans and coastal upwelling regions (for example, the Arabian Sea and eastern tropical Pacific), in which its production is the highest (Extended Data Fig. 6c,d). This is because strong upwelling retains the labile organic carbon in the surface ocean long enough for it to be respired. Another interesting region is the high-latitude North Atlantic Ocean, in which export is high even though production is low. This is because strong vertical mixing reinforces the export of short-lived organic carbon (Fig. 3d).

Regionally, the contribution of advective-diffusive export (labile + semi-labile) to total carbon export can be higher than 50% (ref. 4) (Fig. 3c and Extended Data Fig. 7). The high-contribution regions are mainly in the middle-latitude and high-latitude oceans, such as the subtropical North Atlantic and South Atlantic oceans, and high-latitude North Atlantic Ocean and the Southern Ocean convergence zones,

whereas in the equatorial upwelling zones, the contribution of advective-diffusive export is less than 10%. Overall, our estimated pattern of advective-diffusive flux is in close agreement with the results estimated on the basis of an inverse model constrained using the US Climate Variability and Predictability (CLIVAR) DOC observations[32]. The zonally averaged advective-diffusive export proportion (sum of advective-diffusive fluxes by labile and semi-labile organic carbon over TOC flux) increases from about 15% in equatorial regions (0–15°) to about 37%, about 39% and about 29% in subtropical (15–30°), temperate (30–45°) and subpolar (45–60°) areas, respectively. The poleward increase of advective-diffusive export ratios is consistent with the mechanisms of the mixed-layer pump[17,33], eddy subduction pump[18] and large-scale subduction pump[34].

Combining the non-advective-diffusive and advective-diffusive fluxes, our globally integrated TOC flux at the base of the euphotic zone is 15.00 ± 1.12 Pg C year$^{-1}$ (Fig. 4a). This number is sensitive to the export horizon owing to strong remineralization in the upper ocean. For example, the export flux decreases by roughly 30% from 73 m to the 100-m-depth horizon typically used by ESMs as a reference export depth. An alternative perspective on this sensitivity is provided by distribution functions for the sequestration time, $\tau$, of organic carbon production and for the stock of regenerated DIC (Fig. 5). The TOC production with $\tau > 3$ months is 11.09 ± 1.02 Pg C year$^{-1}$. For $\tau > 1$ year, the total export flux decreases to 8.25 ± 0.30 Pg C year$^{-1}$ and for $\tau > 3$ years, it is only 6.30 ± 0.09 Pg C year$^{-1}$. The distribution functions show that the total flux is dominated by small residence-time export, but that the small residence-time fluxes contribute negligibly to the standing stock of regenerated DIC, pointing to the rapid recycling of much of the organic matter production on short timescales. For $\tau < 1$ year (yellow regions in Fig. 5), the accuracy of export fluxes is highly uncertain as a result of three main factors. First, the circulation model lacks representation of the seasonal cycle. Second, the short residence-time fluxes are sensitive to the mathematical formulation of the biological production and respiration models. Last, the inverse model, which is constrained by carbon, oxygen and nutrient stocks, is insensitive to the part of the export-flux distribution that does not affect these stocks.

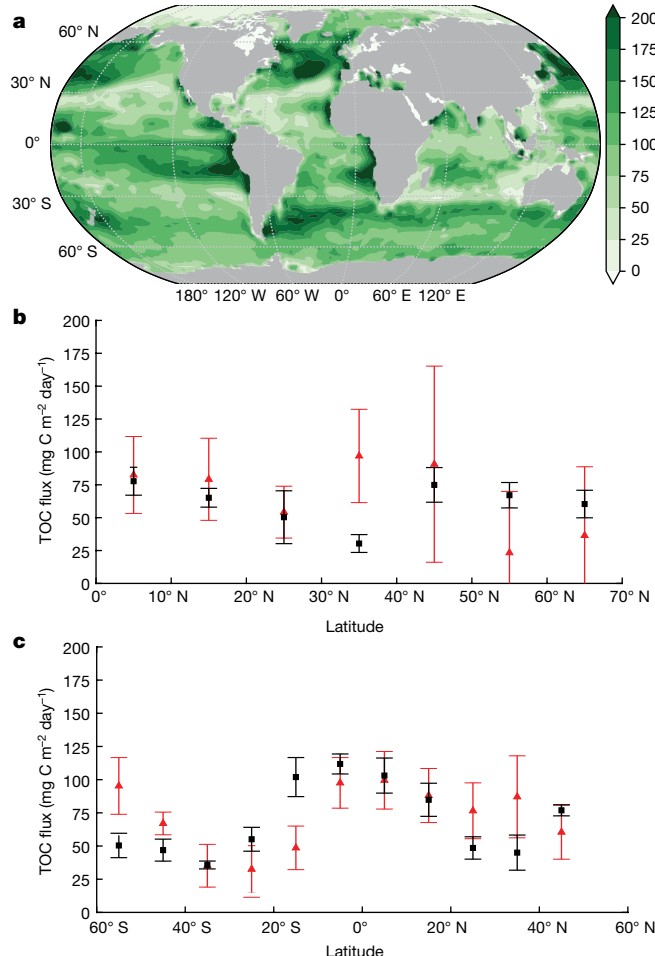

**Fig. 4 | TOC flux at the base of the model euphotic zone. a**, Distribution of TOC (non-advective-diffusive + advective-diffusive) flux (mg C m$^{-2}$ day$^{-1}$) at the base of the euphotic zone. **b,c**, Comparisons of TOC flux with geochemical ANCP estimates at the North Atlantic and Pacific oceans, respectively. The black squares represent the mean ($\pm 1\sigma$) of TOC flux over 10° latitude bands in this study. The red triangles correspond to the mean ($\pm 1\sigma$) of geochemical ANCP estimates[33], which is computed at the base of a spatially varying mMLD obtained from a CESM simulation. For a fair comparison, we extend our flux to the bottom of mMLD at places in which mMLD is deeper than the model euphotic zone depth (Methods). Our TOC flux is the sum of non-advective-diffusive and advective-diffusive flux at places in which mMLD is shallower than the euphotic depth. The results are based on the CbPM NPP product and an e-folding remineralization time of 12 h for labile DOC.

Indeed, marked contributions to the standing stock (Fig. 5b) only become apparent when residence times approach approximately 1 year.

Our estimated TOC flux rate at 100 m (10.64 ± 0.80 Pg C year$^{-1}$) falls into the range of the previous model and satellite-based predictions (5–12 Pg C year$^{-1}$, summarized in ref. 14 in their table 1) and is in close agreement with the 'baseline' estimate of 10.2 Pg C year$^{-1}$ using an ensemble numerical model constrained with $O_2$ and DOC observations[24]. There are no direct global-scale annual TOC flux measurements because extensive samplings would be needed to resolve the seasonal cycle of all export pathways. Reliable ANCP (equivalent to TOC flux at steady state) estimates are only available at time-series stations and some basins based on regional ARGO float data[35]. On basin scales, our results align with the geochemical ANCP in the Pacific Ocean and North Atlantic Ocean[35] (Fig. 4b,c). The magnitude of our estimated TOC export flux varies meridionally by approximately a factor of three, indicating a smaller gradient compared with previous ESM-based or satellite-based estimates, which typically suggest that TOC export varies by up to a

factor of ten[35]. We further compared our TOC flux with those measured using mass-balance calculations at ALOHA, BATS and OSP. Our model results (mean with ±$\sigma$) at the base of maximum mixed-layer depth (mMLD) have overlapping error bars with mass-balance estimates at ALOHA (45.99 ± 23.00 this study versus 82.15 ± 23.00 mg C m$^{-2}$ day$^{-1}$) and at OSP (52.65 ± 3.29 this study versus 75.56 ± 19.71 mg C m$^{-2}$ day$^{-1}$)[14]. Our estimate at the BATS station (23.00 ± 3.28 mg C m$^{-2}$ day$^{-1}$ at mMLD) is much lower than the ANCP by Emerson[14] (124.83 ± 39.42 mg C m$^{-2}$ day$^{-1}$ at 150 m), but twofold higher than the ANCP determined using $O_2$ and DI$^{13}$C in the western North Atlantic around the BATS station at 100 m depth (82.13 ± 13.14 this study versus 39.42 mg C m$^{-2}$ day$^{-1}$ (ref. 36)).

## Biogeochemical implications

Budgets based on in situ observations often struggle to establish a balance between community production and respiration (for example, refs. 37,38), either because they fail to account for all processes that deliver organic carbon to the mesopelagic ocean or because they are limited to measurements during a specific season. Our model, which represents an annual-mean balance between community production and respiration, is able to simultaneously fit full water-column observations of DIC, DOC, ALK and $O_2$, showing that there is no difficulty in closing the budget provided one accounts for both advective-diffusive and non-advective-diffusive export pathways. At the Porcupine Abyssal Plain site in the North Atlantic Ocean, our TOC export (201.5 ± 29.4 mg C m$^{-2}$ day$^{-1}$ between 73 and 1,000 m) exceeds the in situ community respiration (48–167 mg C m$^{-2}$ day$^{-1}$ between 50 and 1,000 m) measured in the summer season[39] when net community production is relatively low[38]. At station ALOHA, our annual TOC flux between mMLD and 1,000 m (45.1 ± 4.0 mg C m$^{-2}$ day$^{-1}$) overlaps with in situ measurements of heterotrophic respiration rates between 150 and 1,000 m (32.5–96.6 mg C m$^{-2}$ day$^{-1}$)[37]. However, at the Japanese time-series site K2 station, also in the Pacific, our TOC flux between mMLD and 1,000 m (82.1 ± 2.4 mg C m$^{-2}$ day$^{-1}$) falls short of the lower end of in situ determinations (106.1–249.8 mg C m$^{-2}$ day$^{-1}$)[37] at the depth interval of 150–1,000 m. Such disparities could potentially arise because our model represents an annual mean, whereas the in situ measurements were conducted during specific seasons. Future development of a seasonal inverse model could contribute to narrowing this difference. The disparities might also be influenced by the inherent uncertainties associated with in situ measurements. In light of these potential factors, we advocate for an increased number of in situ observations focused on year-round whole-community carbon demand within the twilight zone.

Numerous mechanisms have been proposed to explain the spatial variations of carbon flux, with prominent factors including particle size and sinking velocities, community structure, remineralization dependence on temperature and oxygen, and ballast effect[40,41]. ESMs that incorporate these mechanisms in varying degrees exhibit a wide range of carbon flux (approximately 5–12 Pg C year$^{-1}$)[3] and have clearly identifiable biases in their simulated oxygen and carbon distributions. By contrast, our inverse model avoids overparameterization, by not including explicit representations of each of these processes. Nevertheless, it provides a good fit to the tracer data with a simple temperature-dependent parameterization for the remineralization of organic carbon. Specifically, our model adopts a power-law parameterization with a temperature-dependent exponent $b = b_{C\theta}T + b_C$ for non-advective-diffusive carbon fluxes (Methods). Our inversion infers a temperature dependence, $b_{C\theta} = 0.03\ °C^{-1}$ (Extended Data Table 1) that is approximately 50% smaller than the value estimated using a limited sediment trap dataset of POC fluxes[42], but is otherwise in agreement with the sign of the temperature effect. Geographically, non-advective-diffusive vertical fluxes attenuate faster when surface waters are warmer and penetrate deeper in the water column when surface waters are cold (Extended Data Fig. 8). Notably, our non-advective-diffusive vertical

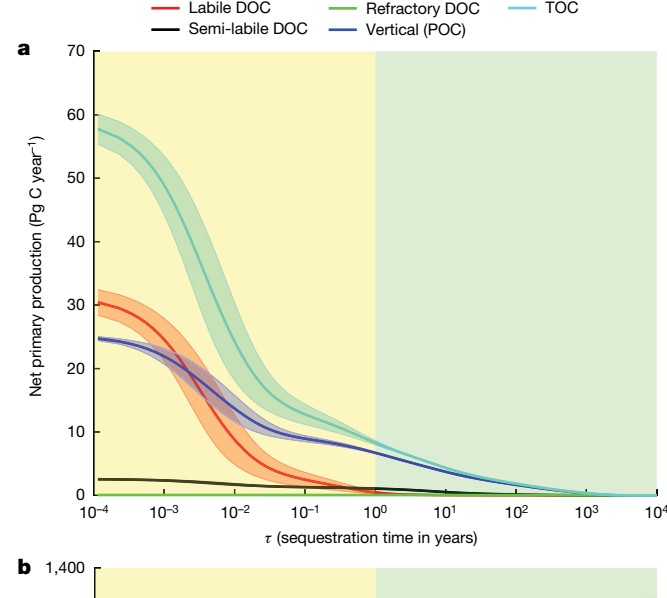

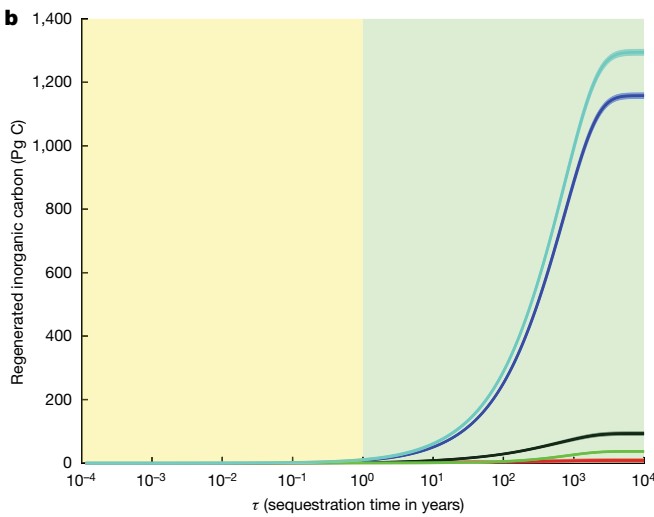

**Fig. 5 | Sequestration-time distribution functions for the organic carbon flux and the stock of regenerated DIC. a**, Sequestration-time-partitioned organic carbon production. The curves show the cumulative NPP fluxes with sequestration times greater than $\tau$ separated into contributions from labile DOC (red), semi-labile DOC (black), refractory DOC (green) and POC (blue). The sequestration times are measured from the time when the organic carbon is respired into DIC to the time when the regenerated DIC is transported back to the 36.1-m-thick surface layer of the model. **b**, Sequestration-time-partitioned standing stock of regenerated DIC. The curves show the cumulative stock with sequestration times less than $\tau$. All curves correspond to climatological-mean estimates integrated over the whole ocean volume. The error bars, indicated by the shaded regions, correspond to ±1$\sigma$ computed from four inverse models in which the e-folding lifetime of labile DOC was either 12 h or 24 h and the biological carbon production was patterned using either the CbPM or the CAFE NPP products. The posterior parametric uncertainty makes a negligible contribution to the shown error bars. For $\tau > 1$ year (green regions), the inverse model produces a robust estimate of the export-flux distribution.

flux includes not only the classical gravitational POC flux but also any fluxes with substantial non-advective-diffusive vertical transport, such as fluxes related to seasonal lipid pump[22] and zooplankton migration pump[21]. It is also worth noting that, in high-latitude low-temperature oceans, the prevalence of large phytoplankton with ballast shells and shorter food webs promotes non-advective-diffusive vertical fluxes. Conversely, in warm subtropical gyres, the prevalence of small phytoplankton and longer food webs reduces non-advective-diffusive flux[40,41]. The deeper penetration in higher latitudes, coupled with an

overall lower temperature dependence compared with the trap-derived value (0.03 °C$^{-1}$ this study versus 0.062 °C$^{-1}$ (ref. 42)) underscores the intricate interplay of different mechanisms. In our inverse model, the dependence of the power-law exponent on temperature serves as a proxy for any mechanism that correlates with surface temperature. Future research will need to unravel these mechanisms. But if we assume that the contemporary relationships persist into the future, we can expect that global warming will cause stronger non-advective-diffusive vertical-flux attenuation (increased $b$-value in Extended Data Fig. 8c,d), which would leave more carbon in the upper ocean and atmosphere[43]. The same mechanism could help to explain atmospheric $CO_2$ variations during glacial–interglacial cycles[44]. The more efficient downward carbon transfer in cold waters compared with warm waters (evidenced by lower $b$-value in high latitudes; Extended Data Fig. 8) suggests a stronger removal of $CO_2$ from the atmosphere during cold climates.

Our results emphasize the role played by advective-diffusive export. Only a few global-data-constrained estimates of carbon export[23,24] and algorithms account for advective-diffusive export of DOC and suspended POC (refs. 7,45,46) or export mediated by zooplankton migration[3]. Previously, the contribution from DOC was typically included by simply scaling up the POC flux by an assumed amount[2]. However, Emerson[14] found that sinking POC export is a small fraction of the ANCP at three time-series stations (BATS, ALOHA and OSP), suggesting that other export pathways are important. Indeed, we find that the export of DOC and suspended POC can be regionally important, especially in subtropical gyres in which DOC production is high and Ekman convergence transports DOC downward[4,20] and in high-latitude oceans in which the subduction pump and mixed-layer pump are strong[17,18] (Fig. 3c). More importantly, in situ observations often miss such mixing events because sea-going measurements usually take place during the summer, when there is less vertical mixing in the water column. This is a possible reason why POC export ratios determined in situ are negatively correlated with NPP in the Southern Ocean[10]. Indeed, we find that up to 70% of the production is exported by means of the advective-diffusive pathway in the latitudes between the subtropical and subantarctic fronts (Fig. 3c). The negative correlation between POC export ratio and NPP contradicts the empirical relationships that relate the ef-ratio to temperature and NPP (refs. 6,45) by assuming a positive relationship between NPP and the ef-ratio.

Furthermore, the export of DOC is not associated with the export of particulate inorganic carbon (PIC) as the POC export may be. Such export can therefore be more efficient at sequestering $CO_2$ by avoiding the effects of the carbonate counter pump[47]. However, a more sluggish circulation[48,49] and stronger stratification[50] expected as a result of future warming may decrease the export of DOC and suspended POC and thus contribute a positive feedback to climate warming. An improved mechanistic understanding of the various pathways associated with the BCP should help to decipher what controls carbon export efficiency and improve predictions of future carbon exports[11–13]. Our results highlight the importance of including the advective-diffusive flux of DOC and suspended POC when estimating the strength of the BCP and motivate the need to improve satellite-based carbon export algorithms so that they better account for export mediated by mixing and other fluid transport.

One strength of our inverse model is that the estimated export fluxes are not sensitive to satellite-estimated NPP. This is a substantial difference from export estimates based on the ef-ratio, which suffer from the compound uncertainties in the ef-ratio and in the algorithm used to estimate the NPP (ref. 28). By contrast, our inverse model infers carbon export from the respiration signal imprinted in the full water column DIC, DOC, DIP, ALK and oxygen observations. Unlike prognostic ESMs, our top-down inverse estimate avoids the need for incorporating uncertain and possibly incorrect parameterizations of complex processes for which we have insufficient understanding. However, our model has its own limitations. For example, our advection–diffusion

transport model represents the climatological annual-mean circulation and lacks seasonality. Therefore, we are unable to diagnose how export changes seasonally. Future developments of our inverse model should consider the effect of seasonal variation. Finally, the successful integration of DIC and oxygen measurements in our model was contingent on an accurate estimation of the transient anthropogenic carbon signal. Our estimate shows that the vertical DIC gradient in the ocean has decreased by approximately 20% owing to the invasion of anthropogenic $CO_2$ (Extended Data Fig. 5c and 'Anthropogenic DIC' in Methods). We therefore expect that future improvements in anthropogenic carbon-uptake estimates will need to take into account the multitracer constraints we used here.

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

## Methods

### Data

Observational concentrations of DIP, DIC, ALK and O$_2$ were downloaded from the Global Ocean Data Analysis Project website, the second version (GLODAPv2 (ref. 15); Extended Data Fig. 1). DOC observations were from ref. 16. The data were then binned into the ocean circulation inverse model (OCIM) model grid that has a horizontal resolution of 2° × 2° and vertically 24 layers, with finer resolution in the upper ocean and coarser resolution in the deep ocean. The updated DOC compilation contains 25,869 valid data points (Extended Data Fig. 2) after binning to our model grid and has better coverage than previously widely used ones (ref. 51) that only had 14,034 valid data points in the model grid. The DOC dataset has a slight seasonal bias, with more samples collected in the summer season. However, we think that the influence is minor because: (1) a substantial proportion of the total DOC is composed of refractory DOC; unlike labile and semi-labile DOC, refractory DOC does not exhibit strong seasonality owing to its long residence time in the ocean; (2) we used tracer data from the full-water depth to constrain our model parameters. The deep ocean experiences lesser seasonal variability compared with the surface ocean. Therefore, using full-water-depth data helps to anchor the stability of the inversion. Two NPP products, carbon-based NPP from Sea-viewing Wide Field-of-view Sensor (SeaWiFS CbPM)[52] and CAFE, were downloaded from https://doi.org/10.6084/m9.figshare.19074521. The NPP products were interpolated and averaged by Nowicki et al.[24] to the same model grid as used in this study. The climatological ocean temperature and silicate are from World Ocean Atlas 2018 (refs. 53,54). The projected temperature at 2099 was obtained from a CESM-BGC model prediction under the RCP8.5 scenario[55]. The historical atmospheric $p$CO$_2$ data were obtained from ref. 56 for the period from 1850 to 2015 and were downloaded from https://scrippsco2.ucsd.edu/data/atmospheric_co2/primary_mlo_co2_record.html (ref. 57) for the period from 2016 to 2020.

### Biogeochemical inverse model

A schematic of the structure of the biogeochemical model is shown in Fig. 1. The model couples the cycling of phosphorus (P), carbon (C) and oxygen (O). The phosphorus model is the base model that provides a biological uptake rate ($\gamma(\mathbf{r})$, in which $\mathbf{r}$ is a position coordinate) in P units ($\mathbf{G} \equiv (\gamma[\text{DIP}])$), which is then converted to a DIC uptake rate in the carbon model by incorporating a C:P ratio ($r_{\text{C:P}}$). In the P-cycle model, the DIP assimilation rate is modelled using a spatial pattern obtained from satellite-derived NPP (mg C m$^{-2}$ day$^{-1}$) and a gridded surface DIP climatology as follows

$$\gamma(\mathbf{r}) \equiv \begin{cases} \alpha \dfrac{\left[\dfrac{1}{r_{\text{C:P}}} \dfrac{\text{NPP}(\mathbf{r})}{\text{NPP}_0}\right]^{\beta}}{\dfrac{[\text{DIP}]_{\text{obs}}(\mathbf{r})}{[\text{DIP}]_0}}, & \text{if } z < z_{\text{c}}, \\ 0, & \text{otherwise} \end{cases} \quad (1)$$

in which NPP$_0$ and [DIP]$_0$ are 1 mmol C m$^{-2}$ day$^{-1}$ and 1 μM that are functioned to remove dimensions of NPP and DIP; $\alpha$ and $\beta$ are adjustable parameters that are constrained in the inversion; $r_{\text{C:P}}$ is the C:P ratio that is used to convert NPP from C unit to P unit and modelled according to Galbraith and Martiny[58] ($r_{\text{C:P}} = (0.006 + 0.0069[\text{DIP}]_{\text{obs}})^{-1}$). $z$ and $z_{\text{C}}$ are water depth and the euphotic zone depth, respectively. Photosynthesis is assumed to occur only in the euphotic zone and to be zero below. The euphotic zone is defined as the top two model layers (73.4 m).

**Phosphorus model.** The phosphorus model considers four explicit tracers: dissolved inorganic phosphorus (DIP), dissolved semi-labile organic phosphorus (DOP), dissolved labile organic phosphorus (DOP$_l$) and particulate organic phosphorus (POP). We assign an e-folding

remineralization time ($1/\kappa_l$) of 12 h for DOP$_l$ so that it quickly cycles in the upper ocean, with little being transported below the euphotic zone. We use a parameter $\delta$ to allocate production to labile pools. The remaining production (total production less production to DOP$_l$) is allocated to DOP and POP. The factions $\sigma_P$ and $(1 - \sigma_P - \delta)$ of the production allocated, respectively, to DOP and POP are determined by estimating the parameter $\sigma_P$ through our Bayesian inversion procedure. The advective-diffusive transport of dissolved tracers (DIP, DOP and DOP$_l$ in the P model; DIC, semi-labile dissolved organic carbon (DOC), labile dissolved organic carbon (DOC$_l$), refractory dissolved organic carbon (DOC$_r$) and ALK in the C model; and O$_2$ in the O model) is computed using the OCIM tracer transport matrix, $\mathbf{T}[C] \equiv \nabla \cdot (\vec{U}[C] - \text{K}\nabla[C])$, in which $\vec{U}$ is the velocity vector and K is the diffusive term. $\mathbf{T}$ represents the climatological mean circulation of the ocean. The OCIM tracer transport matrix is constrained using salinity, temperature, sea-surface height, CFC-11, CFC-12, $^{14}$C, $^3$He etc. (see DeVries and Holzer[59] for details). We neglect the advective-diffusive transport of particulate tracers (POP in the P model and PIC and POC in the C model) so that particulate tracers are transported only vertically. The vertical transport of POP is modelled using a sinking flux divergence operator ($\mathbf{F}_{\text{POP}} \equiv \nabla \cdot (\vec{w}[\text{POP}])$), in which $\vec{w}$ is the sinking speed of POP and is directed downward. We choose a sinking speed that increases linearly with depth and a constant dissolution rate, $\kappa_P = (1/30)$ days$^{-1}$, so that the attenuation of the vertical flux of POP follows a power-law function, $F(z) = F(z_0)(z/z_0)^{-b}$, in which $F(z)$ and $F(z_0)$ are fluxes at a depth of $z$ and $z_0$, respectively[60]. A sensitivity test with $\kappa_P = (1/60)$ days$^{-1}$ suggests that the choice of $\kappa_P$ does not markedly influence our results. The exponent $b$ for the P model (C model in the following section) is defined in the following way (ref. 42), $b(\text{P}) = b_{P\theta}T + b_P$, in which $b_{P\theta}$ and $b_P$ are two adjustable parameters and $T$ is the average temperature of the model euphotic zone. The initial guess of $b_{P\theta}$ is set to zero, thereby avoiding any intentional imposition of temperature dependence. The optimization process determines both the sign and magnitude of $b_{P\theta}$. The governing equations for the phosphorus cycle are as follows:

$$\left[\frac{\text{d}}{\text{d}t} + \mathbf{T}\right][\text{DIP}] = -\gamma[\text{DIP}] + \kappa_p[\text{POP}] + \kappa_{\text{dP}}[\text{DOP}] + \kappa_l[\text{DOP}_l] + \kappa_g([\text{DIP}] - \overline{[\text{DIP}]}_{\text{obs}}),$$

$$\left[\frac{\text{d}}{\text{d}t} + \mathbf{T}\right][\text{DOP}] = \sigma_P\gamma[\text{DIP}] - \kappa_{\text{dP}}[\text{DOP}],$$

$$\left[\frac{\text{d}}{\text{d}t} + \mathbf{T}\right][\text{DOP}_l] = \delta\gamma[\text{DIP}] - \kappa_l[\text{DOP}_l],$$

$$\left[\frac{\text{d}}{\text{d}t} + \mathbf{F}_{\text{POP}}\right][\text{POP}] = (1 - \sigma_P - \delta)\gamma[\text{DIP}] - \kappa_P[\text{POP}], \quad (2)$$

in which $\kappa_{\text{dP}}$ is the DOP remineralization rate constant that is a function of temperature defined using a Q$_{10}$ function ($\kappa_{\text{dP}} = \kappa_{P\theta}Q_{10}^{(T-30)/10}$), in which $T$ is water temperature from World Ocean Atlas 2018 (ref. 54). $\kappa_{P\theta}$ and Q$_{10}$ are optimized in the inversion. $\kappa_l$ is the e-folding remineralization time of DOP$_l$, which is fixed at $\kappa_l = (1/12)$ h$^{-1}$. We tested the sensitivity to a smaller $\kappa_l = (1/24)$ h$^{-1}$ and found that the choice of $\kappa_l$ did not substantially change the fittings to the tracers but could alter the export flux of labile organic matter. We therefore include different $\kappa_l$ values in the uncertainty analysis (see the 'Uncertainty analysis' section). $\kappa_g$ is prescribed to $(1/10^6)$ years$^{-1}$ and is used to set the global mean DIP concentration to the observed global mean concentration ($[\text{DIP}]_{\text{obs}}$). $\kappa_P$ is a prescribed POP remineralization rate constant ($\kappa_P = (1/30)$ days$^{-1}$). A sensitivity test shows that increases or decreases in the fraction of DOP$_l$ production ($\delta$ in equations (2) and (3)) does not alter the fit to the observational data nor does it change the inferred export fluxes of POC

and semi-labile DOC. We therefore set $\delta$ to be zero in the first-round optimization.

**Carbon model.** The carbon model explicitly simulates seven tracers: DIC, DOC$_l$, DOC, DOC$_r$, POC, PIC and ALK (Fig. 1b). The DIP assimilation rate **G** is converted to the DIC assimilation rate by incorporating a C:P ratio ($r_{C:P}$) that is allowed to vary spatially according to the modelled DIP concentration, $r_{C:P} = (cc[DIP] + dd)^{-1}$, in which cc and dd are estimated as part of the inversion. As in the P model, we set the allocation to DOC$_l$ to be zero in the first round of optimization. Subsequently, we prescribe the difference between satellite NPP and model organic carbon production as the production for labile DOC$_l$, so that our model production matches satellite NPP exactly. The fraction, $\sigma_C$, of the organic carbon production allocated to POC and DOC pools is estimated as part of the inversion and does not need to be the same as the fraction $\sigma_p$ allocated to the POP and DOP pools. A further adjustable parameter, $\eta$, is used to control the fraction of DOC that is transferred to the refractory pool by bacterial reworking. The remaining DOC fraction $(1 - \eta)$ is remineralized back to DIC. The e-folding decay times of DOC$_r$ ($\kappa_{ur}$ and $\kappa_{dr}$ for the upper and deeper ocean, respectively) are estimated as part of the inversion. POC sinks and is gradually remineralized to DIC in the water column. The downward transport of POC is modelled using a flux divergence operator (**F**$_{POC}$), which is formulated in the same way as the POP sinking flux-divergence operator **F**$_{POP}$ with independent adjustable parameters $b_{C\theta}$ and $b_C$ that are determined as part of the inversion (Extended Data Table 1). Unlike DIP, DIC experiences sea-to-air gas exchange at the surface. This gas exchange is modelled according to the method used for phase 2 of the Ocean Carbon-Cycle Model Intercomparison Project (OCMIP-2)[61] using a recalibrated piston velocity (see the next section). Also, freshwater precipitation and evaporation can greatly affect surface ocean DIC and ALK concentrations. Precipitation will dilute, whereas evaporation will concentrate their concentrations. A virtual flux according to OCMIP-2 (ref. 61) is applied to model for the effects of precipitation and evaporation on DIC and ALK (**F**$_{vDIC}$[DIC]$_s$ and **F**$_{vALK}$[ALK]$_s$, in which [DIC]$_s$ and [ALK]$_s$ are the mean surface-ocean concentrations of DIC and ALK, respectively).

Production of PIC is modelled to be proportional to the production of POC using two adjustable parameters, $r_{Si}$ and $r_{RR}$, that are estimated in the inversion. The parameter $r_{Si}$ adjusts PIC production according to silicate concentration in the surface ocean in linear form ($R_{RR} = r_{Si}[SiO_4^{4-}] + r_{RR}$). The downward transport of PIC is modelled using a flux divergence operator (**F**$_{PIC}$), which generates a PIC flux profile that follows an exponential function $F_{PIC}(z) = F_0 exp((z - z_0)/d)$, in which $d$ is the PIC dissolution length scale, whose value is estimated as part of the inversion (Extended Data Table 1). Compared with a power-law function, an exponential function with a length scale on the order of several thousand metres leads to a much smaller CaCO$_3$ dissolution rate in the shallow water in which CaCO$_3$ is supersaturated[62]. Every mole of PIC production consumes two moles of ALK. By contrast, the dissolution of one mole of PIC releases two moles of ALK (equation (3)). From the perspective of carbon, photosynthesis and remineralization of organic matter do not change alkalinity. However, in the processes of photosynthesis and remineralization, chemical forms of nitrogen change, which influences alkalinity so that a mole of organic carbon production increases alkalinity by $r_{N:C}$ moles, whereas a mole of organic carbon remineralization decreases alkalinity by $r_{N:C}$ moles. The governing equations for carbon cycling are as follows:

$$\left[\frac{d}{dt} + \mathbf{T}\right][DIC] = -(\mathbf{I} + (1 - \sigma_C - \delta)r_{RR})\mathbf{G}r_{C:P} + \eta\kappa_{dC}[DOC] + \kappa_l[DOC_l]$$
$$+ \kappa_r[DOC_r] + \kappa_{PIC}[PIC] + \kappa_p[POC] + \mathbf{F}_{CO_2} + \mathbf{F}_{vDIC}[DIC]_s,$$

$$\left[\frac{d}{dt} + \mathbf{T}\right][DOC] = \sigma_C\mathbf{G}r_{C:P} - \eta\kappa_{dC}[DOC],$$

$$\left[\frac{d}{dt} + \mathbf{F}_{POC}\right][POC] = (1 - \sigma_C - \delta)\mathbf{G}r_{C:P} - \kappa_p[POC],$$

$$\left[\frac{d}{dt} + \mathbf{F}_{PIC}\right][PIC] = (1 - \sigma_C - \delta)R_{RR}\mathbf{G}r_{C:P} - \kappa_{PIC}[PIC],$$

$$\left[\frac{d}{dt} + \mathbf{T}\right][ALK] = -2(1 - \sigma_C - \delta)R_{RR}\mathbf{G}r_{C:P} + r_{N:C}\mathbf{G}r_{C:P}$$
$$- r_{N:C}(\eta\kappa_{dC}[DOC] + \kappa_r[DOC_r] + \kappa_l[DOC_l] + \kappa_p[POC])$$
$$+ 2\kappa_{PIC}[PIC] - \mathbf{F}_{vALK}[ALK]_s + \kappa_g([ALK] - \overline{[ALK]}_{obs}),$$

$$\left[\frac{d}{dt} + \mathbf{T}\right][DOC_l] = \delta\mathbf{G}r_{C:P} - \kappa_l[DOC],$$

$$\left[\frac{d}{dt} + \mathbf{T}\right][DOC_r] = (1 - \eta)\kappa_{dC}[DOC] - \kappa_r[DOC_r]. \tag{3}$$

**Anthropogenic DIC.** To use DIC observations to constrain our inverse model, we have to take into account the changing DIC concentration owing to the invasion of anthropogenic CO$_2$ into the ocean. To obtain a self-consistent estimate of the anthropogenic carbon signal, we performed a time-dependent simulation using equation (3). Starting from an assumed steady state, we time-stepped our carbon-cycle model forward in time from 1850 to 2020, using an implicit trapezoid-rule time-integration scheme for all terms except for the gas exchange, for which we used an explicit Euler forward scheme. In this calculation, we prescribed the surface SST according to a time-dependent reanalysis product (ref. 63). The transient integration was carried out with a time step size of $\Delta t = 2$ months. The atmospheric $p$CO$_2$ was prescribed according to ref. 56 from 1850 to 2015 and according to ref. 57 from 2016 to 2020. We also simulated $\delta^{14}$C to better calibrate the air–sea gas-exchange velocity as described below. The atmospheric $\delta^{14}$C was prescribed according to ref. 64 for the period from 1850 to 2015 and according to ref. 65 from 2016 to 2020. To produce the initial conditions, we assumed that the system was in steady state in 1850 and used Newton's method to find the steady state.

To calibrate the air–sea gas exchange parameterization, we re-optimized the scaling factor in the OCIM2 gas-exchange scheme by minimizing the misfit between our modelled $\delta^{14}$C and the GLODAPv2 $\delta^{14}$C data. See Extended Data Fig. 5a,b for the number of observations as a function of time. To compute the misfit, we sampled our model at the location and times of the bottle measurements in the GLODAPv2 database. Our calibration method followed an iterative two-step process in which we first optimized the air–sea gas exchange through a series of transient carbon-cycle simulations. After obtaining the optimal air–sea gas exchange, we subtracted the excess anthropogenic DIC from the GLODAPv2 measurements to produce an estimate of the natural background DIC for the year 1850. The resulting DIC data and optimal gas-exchange velocity were then used for the optimization of the biogeochemistry model (see the 'Parameter estimation' section). The optimized biogeochemical model was then used to produce an updated initial condition for the transient carbon-cycle simulation and a re-optimization of the air–sea gas-exchange velocity. We repeated this two-step process until we obtained self-consistent estimates of: (1) the optimal biogeochemical parameter values (Extended Data Table 1); (2) the biogeochemical state; (3) the scaling factor for the air–sea gas-transfer velocity, $a = 0.234$ cm h$^{-1}$ (m s$^{-1}$)$^{-2}$; (4) transient DIC; and (5) the transient $\delta^{14}$C signal including the combined effects of radioactive decay, the Suess effect and the bomb radiocarbon signal. Extended Data Figure 5c shows a time series of the excess anthropogenic DIC concentration averaged over the top 100 m of the water

column and for the water column below 100 m. By 2020, the vertical DIC gradient is reduced by 20%.

**Oxygen model.** Oxygen production is modelled by applying a ratio of oxygen to carbon ($r_{O:C}$) to the DIC assimilation rate ($\mathbf{G}r_{C:P}$). The ratio $r_{O:C}$ is optimized in the process of inversion. We convert the DOC and POC remineralization rates ($\eta\kappa_{dC}[DOC] + \kappa_r[DOC_r] + \kappa_l[DOC_l] + \kappa_p[POC]$) to an oxygen consumption rate using the same $r_{O:C}$ ratio and gradually shut down oxygen consumption as the oxygen concentration falls below the critical value ($O_{crit} = 5$ mmol l$^{-1}$) using a hyperbolic equation ($R([O_2]) = 0.5 + 0.5\tanh[([O_2] - O_{crit})/[O_2]_0]$), in which $[O_2]_0$ (1 mmol l$^{-1}$) is used to remove the $O_2$ dimension. Sea-to-air $O_2$ flux ($\mathbf{F}_{O2}$) is modelled according to OCMIP-2 (ref. 61):

$$\left[\frac{d}{dt} + \mathbf{T}\right][O_2] = r_{O:C}\mathbf{G}r_{C:P} + \mathbf{F}_{O2} - r_{O:C}\mathbf{R}(\eta\kappa_{dC}[DOC] + \kappa_r[DOC_r] + \kappa_l[DOC_l] + \kappa_p[POC]) \tag{4}$$

in which the matrix $\mathbf{R}$ is a diagonal matrix, whose elements are given by $R([O_2])$.

## Parameter estimation

The 21 adjustable parameters of the model (Extended Data Table 1) were estimated using a Bayesian inversion method. In this approach, the solutions to our model equations define the tracer fields as implicit functions of the adjustable parameters, which we then compare with the observations to construct a likelihood function. We obtain the P, C and O fields by finding the steady-state solutions of the governing equations for the P, C and O models (equations (2)–(4)). Because the governing equations for the P model are linear, their steady-state solution can be obtained efficiently by direct matrix inversion after setting the time derivatives in equation (2) to zero. We fix the atmospheric $CO_2$ concentration at the preindustrial level (278 ppm) to compute the preindustrial sea-to-air $CO_2$ flux ($\mathbf{F}_{CO2}$). The steady-state solution for the C model is solved using Newton's method because of nonlinearity in $\mathbf{F}_{CO2}$. The governing equation for O is also nonlinear because of the hyperbolic function ($\mathbf{R}$) that turns off oxygen consumption when oxygen concentration is critically low. We solve the oxygen equations using Newton's method.

To find the most probable parameter values, we minimize the negative logarithm of the posterior probability function, which is equivalent to minimizing the negative log-likelihood because we log-transformed our parameters (except of the slopes of exponent $b$) so that they have flat priors:

$$f = \frac{1}{2}(e'_{DIP}\mathbf{W}_P e_{DIP} + e'_{DIC}\mathbf{W}_{DIC}e_{DIC} + e'_{DOC}\mathbf{W}_{DOC}e_{DOC} + e'_{ALK}\mathbf{W}_{ALK}e_{ALK} + e'_{O_2}\mathbf{W}_{O_2}e_{O_2}) + \text{const.}, \tag{5}$$

in which the $e_X$ are column vectors whose elements are given by the difference between modelled and observed concentrations, $e_X = \mathbf{H}_X[X_{mod}] - [X_{obs}]$, in which the X label denotes the specific tracer and $\mathbf{H}_X$ is a rectangular matrix that picks out the model grid boxes that have observations of tracer X. Because there are no measurements that precisely separate DOC into different pools according to their lability, we sum all three pools in the model (DOC, DOC$_l$ and DOC$_r$) and compare the sum to observations. For DIC, we subtracted our estimated anthropogenic DIC from the bottle measurements in the GLODAPv2 database according to the location and time of measurement (see the 'Anthropogenic DIC' section). $\mathbf{W}_X$ is a precision matrix for tracer X and is defined in the following way:

$$\mathbf{W}_X = \frac{1}{\sigma_X^2}\mathbf{V}_X, \tag{6}$$

in which $\mathbf{V}_X$ is a diagonal matrix with the fractional volumes of the model grid boxes ($\mathbf{V} = \text{diag}(\Delta V_i/\Sigma_i\Delta V_i)$, in which the subscript $i$ is the index of the grid boxes that have at least one observation) and $\sigma_X^2$ is the spatial variance of the observations, that is,

$$\sigma_X^2 = ([X_{obs}] - \mu_X)'\mathbf{V}_X([X_{obs}] - \mu_X), \tag{7}$$

with the spatial mean given by

$$\mu_X = \frac{\mathbf{1}'\mathbf{V}_X[X_{obs}]}{\mathbf{1}'\mathbf{V}_X}, \tag{8}$$

in which $\mathbf{V}_X$ is a diagonal matrix with the grid-box volumes and the subscript X represents the grid boxes that have observations of tracer X. The bold $\mathbf{1}$ represents a column vector. The transpose turns it into a row vector. Thus, the numerator yields the volume integral of $X_{obs}$ and the denominator yields the total volume.

The optimization is conducted using MATLAB's fminunc function, which is computationally efficient because we can supply hand-coded first and second derivatives of the objective function with respect to the adjustable parameters. The optimization generally takes fewer than 100 iterations. The most probable model parameter values are presented in Extended Data Table 1. Parameter error bars that correspond to ±1 standard deviations are calculated using Laplace's approximation as described in ref. 66.

## Calculation of carbon flux

The two-dimensional vertical-flux field ($f_{POC}$) is calculated by vertically integrating POC remineralization below the euphotic zone ($f_{POC} = \sum_{i=1}^z \kappa_p POC_i \Delta V_i M_i$, in which $i$ represents the index for deep grid, $\Delta V_i$ is the volume of the $i$th grid box and $M_i$ is a mask that is set to 1 below the euphotic zone and 0 elsewhere). In our model, POC is transported vertically in the water column and is not advected to neighbouring grids. This approximation is appropriate for the coarse horizontal resolution of our model. The non-advective-diffusive flux below the first two layers is calculated on the basis of the following power-law function (also known as the Martin curve function), $f_{POC}(z) = f_{POC}(z_0)(z/z_0)^{-b}$, in which $z_0$ is the euphotic zone depth and $z$ is the depth at which non-advective-diffusive flux is calculated. The exponent, $b$, depends linearly on the surface water temperature[42]. The estimated slope and intercept ($b_C$ and $b_{c\theta}$) for this linear relationship are presented in Extended Data Table 1. The advective-diffusive fluxes of labile and semi-labile organic carbon are calculated using an adjoint method, which tracks the export and subsequent remineralization of DOC, as described by ref. 31. Only DOC respired below the depth of the euphotic zone is counted as export. The flux of TOC is the sum of the non-advective-diffusive flux and fluxes from labile DOC and semi-labile DOC. We ignore the export of refractory DOC because of its negligible contribution.

To compare the non-advective-diffusive flux to CMIP6 models at their consensus reference depth (about 100 m), we scale our estimated non-advective-diffusive flux using a power-law function with our optimized temperature-dependent $b$ exponents.

To compare our export estimates to the geochemical ANCP estimates that are calculated at the base of spatially varying mMLDs obtained from a CESM simulation[35], we estimated the export fluxes of POC and semi-labile DOC to the depth $z = $ mMLD. For cases in which mMLD is deeper than our euphotic zone depth, we scaled the fluxes using the power-law function with our optimized $b$ exponents. For cases in which mMLD is above the base of the euphotic zone, we did not apply the power-law scaling because it tends to amplify errors. In those cases, we used the export flux at the base of the model's euphotic zone for the comparison. Note that the contribution of labile DOC is ignored when scaling flux down to mMLD owing to the short e-folding decay time (12 h or 24 h).

## Uncertainty analysis

The uncertainty analysis is conducted in two ways. First, we use a Monte Carlo method whereby an ensemble of parameter values is drawn from a multivariate normal distribution whose mean is given by our estimated most probable parameter values and whose covariance matrix is given by the inverse of the matrix of second partial derivatives of the negative logarithm of the posterior probability distribution, that is, by the Hessian matrix. For each ensemble member, we solve the steady-state model equations and calculate the organic carbon fluxes. However, the parameters are so well constrained that their uncertainties are small, and the flux uncertainties calculated this way are small. Second, because the DIP uptake model is constructed with two different satellite NPP products (SeaWiFS CbPM and CAFE), and the 21 adjustable parameters are optimized for each NPP field (Extended Data Table 1), the influence of NPP fields on export fluxes are much larger than that of parameter uncertainties. Also, the e-folding remineralization time of labile DOC is prescribed at 12 h and 24 h. We, therefore, report flux uncertainties estimated from the results based on different initial NPP fields and on different labile DOC e-folding decay timescales. The distributions of the standard deviation of key outputs are illustrated in Extended Data Fig. 9.

## DOC sequestration time

To calculate the DOC sequestration time, we injected unit DOC pulses in the model euphotic zone and tracked this DOC as it was transported by the circulation, respired into DIC (according to the timescale given in Extended Data Table 1) and then transported back to the surface, where it was rapidly removed with a loss frequency of $(1 \text{ day})^{-1}$. We then spatially integrated the removal rate for each DOC pulse to obtain residence-time distributions for the DOC exported from the surface of each water column.

## Sequestration-time-partitioned distribution functions

To compute the sequestration-time-partitioned distribution functions, we use the three-dimensional organic carbon respiration rate to construct a Dirac δ-function pulse of labelled regenerated inorganic carbon. The resulting tracer field is then transported using the circulation model until it is removed in the 36.1-m-thick surface layer of the model using a loss frequency of $(1/500) \text{ year}^{-1}$. We integrate the system forward in time for 10,000 years, by which time all of the regenerated-carbon pulse has left the system. We use a second-order-accurate trapezoidal integration rule starting with a time-step size of less than $10^{-4}$ years and gradually increase it to 10 years by the end of the simulation. A sequestration-time density distribution function is obtained by globally integrating the loss rate and the cumulative distribution function is then obtained by integrating the density function for progressively longer times. To obtain the cumulative sequestration-time distribution for the stock of regenerated DIC, we first integrate the tracer field over the whole volume of the ocean and then integrate the resulting stock for progressively longer sequestration times. By year 10,000, the resulting integral is equal to the global inventory of regenerated DIC.

## Data availability

Supporting data used to run the inverse model are available at https://doi.org/10.5281/zenodo.10016054. Model output from the inverse model is available at https://doi.org/10.5281/zenodo.8253973. Source data are provided with this paper.

## Code availability

The code for the inverse model is available at https://doi.org/10.5281/zenodo.8368856.

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

**Acknowledgements** We thank the thousands of scientists and researchers who made hydrographic and organic carbon measurements. We also thank the people and institutes who maintain the public data repositories (GLODAP and WOA) and the Ocean Productivity website maintained by Oregon State University. W.-L.W. was supported by the National Natural Science Foundation of China (42330401) and the Natural Science Foundation of Fujian Province of China 2023J02001. F.W.P. was supported by U.S. National Science Foundation (NSF) OCE-1948842 and OCE-2424014 and U.S. Department of Energy (DOE) DE-SC0021267. W.F. was supported by U.S. DOE DE-SC0021267. Further support for W.F. and Y.L. was provided by the U.S. DOE Reducing Uncertainty in Biogeochemical Interactions through Synthesis and Computation (RUBISCO) Scientific Focus Area (SFA). R.T.L. was supported by U.S. NSF 1829916 and U.S. DOE DE-SC0022177. J.-M.T. was supported by the National Natural Science Foundation of China (92058204 and 41721005) and the Ph.D. Fellowship of the State Key Laboratory of Marine Environmental Science at Xiamen University.

**Author contributions** W.-L.W. conceived the project, with input from F.W.P. W.-L.W. built the inverse model, with contributions from W.F. and F.W.P. W.-L.W. and F.W.P. carried out the formal analyses, with contributions from W.F. W.-L.W. and F.W.P. wrote the manuscript, with input from W.F., F.A.C.L.M., R.T.L., Y.L. and J.-M.T. All authors contributed to the discussion and commented on the manuscript.

**Competing interests** The authors declare no competing interests.

**Additional information**
**Correspondence and requests for materials** should be addressed to Wei-Lei Wang or François W. Primeau.

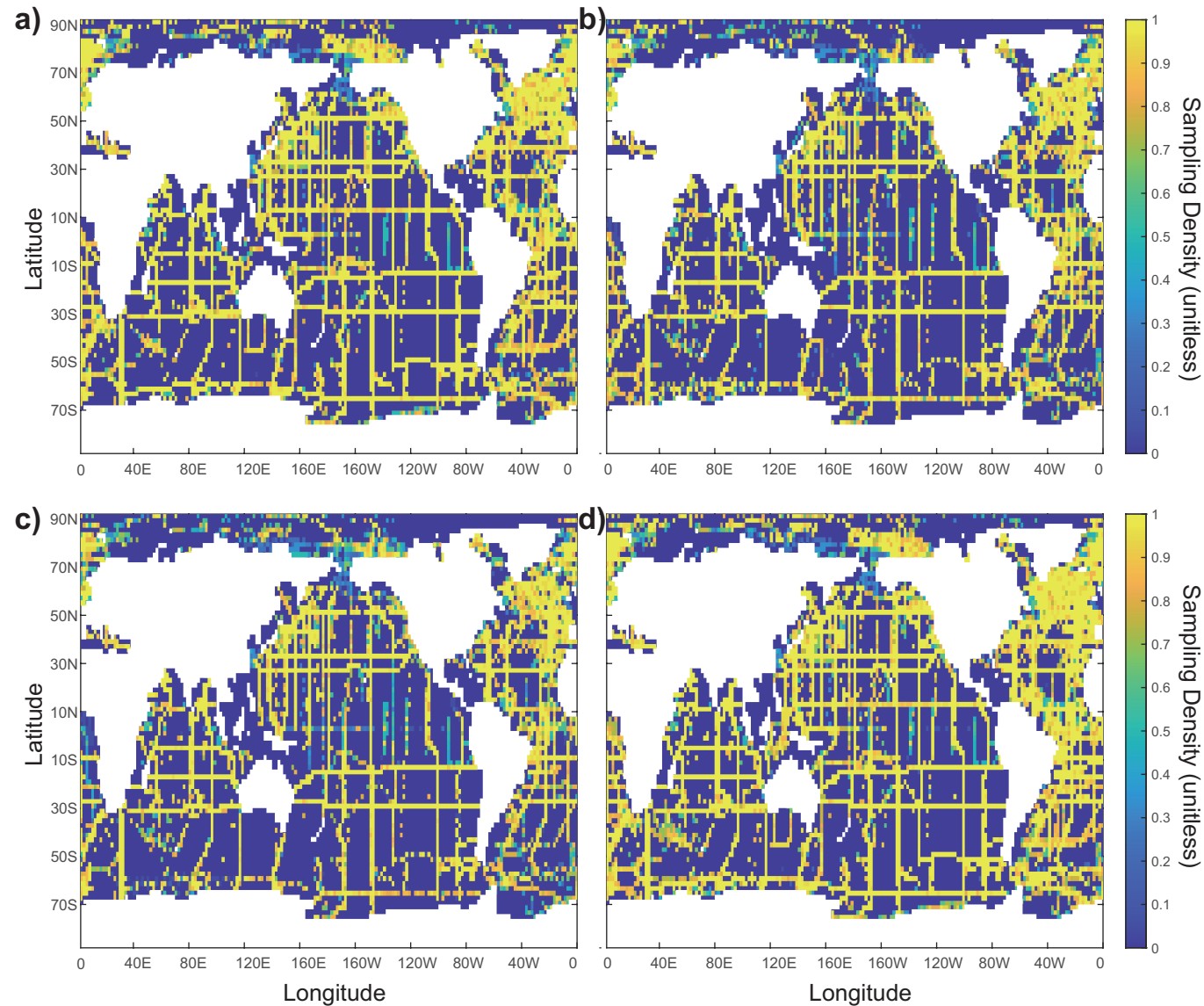

**Extended Data Fig. 1 | Sampling density of DIP, DIC, ALK and O₂. a**, DIP. **b**, DIC. **c**, ALK. **d**, O₂. The observational data, downloaded from the GLODAPv2 (ref. 15), are binned to the OCIM grid. The colour denotes the fraction of the grid boxes in each water column with at least one measurement. For each vertical column, the sampling density is defined as the number of grid boxes with at least one sample divided by the total number of wet grid boxes.

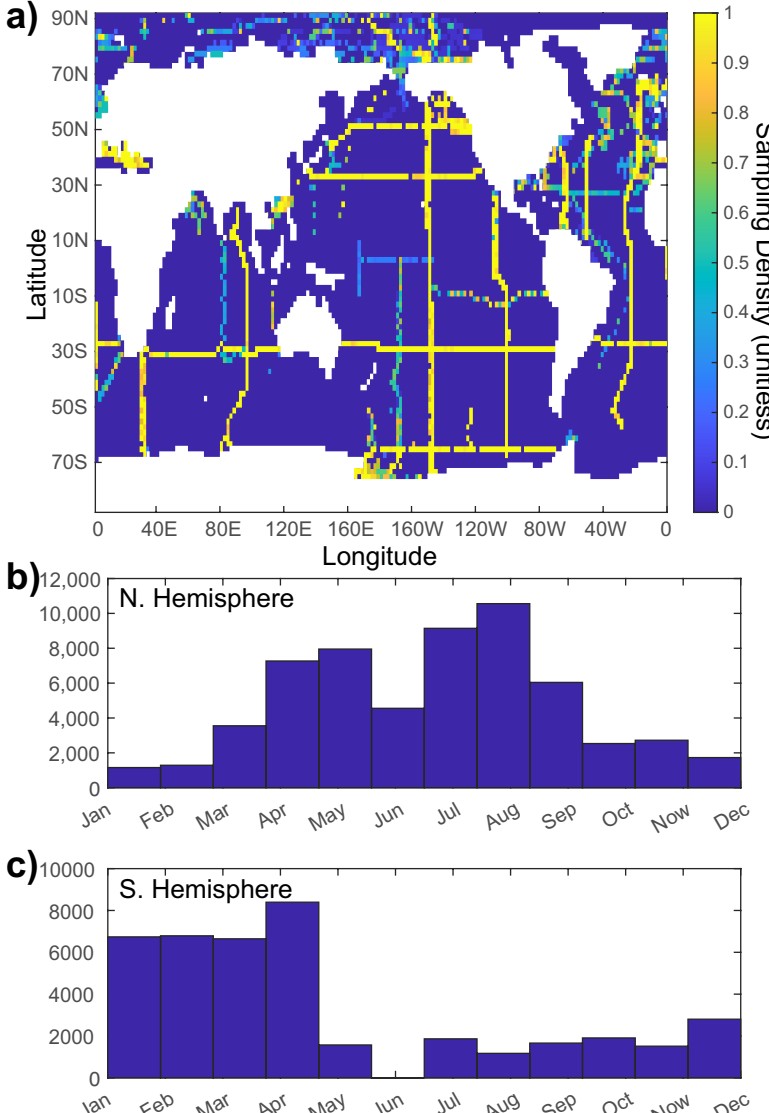

**Extended Data Fig. 2 | Spatial and monthly DOC sampling density.**
**a**, Spatial DOC sampling density. **b**,**c**, Monthly DOC sampling density. The DOC observations, obtained from a recent compilation[16], are interpolated to the OCIM grid. The colour denotes the fraction of the grid boxes in each water column with at least one measurement. For each water column, the sampling density is defined as the total number of grid boxes with at least one sample divided by the total number of wet grid boxes.

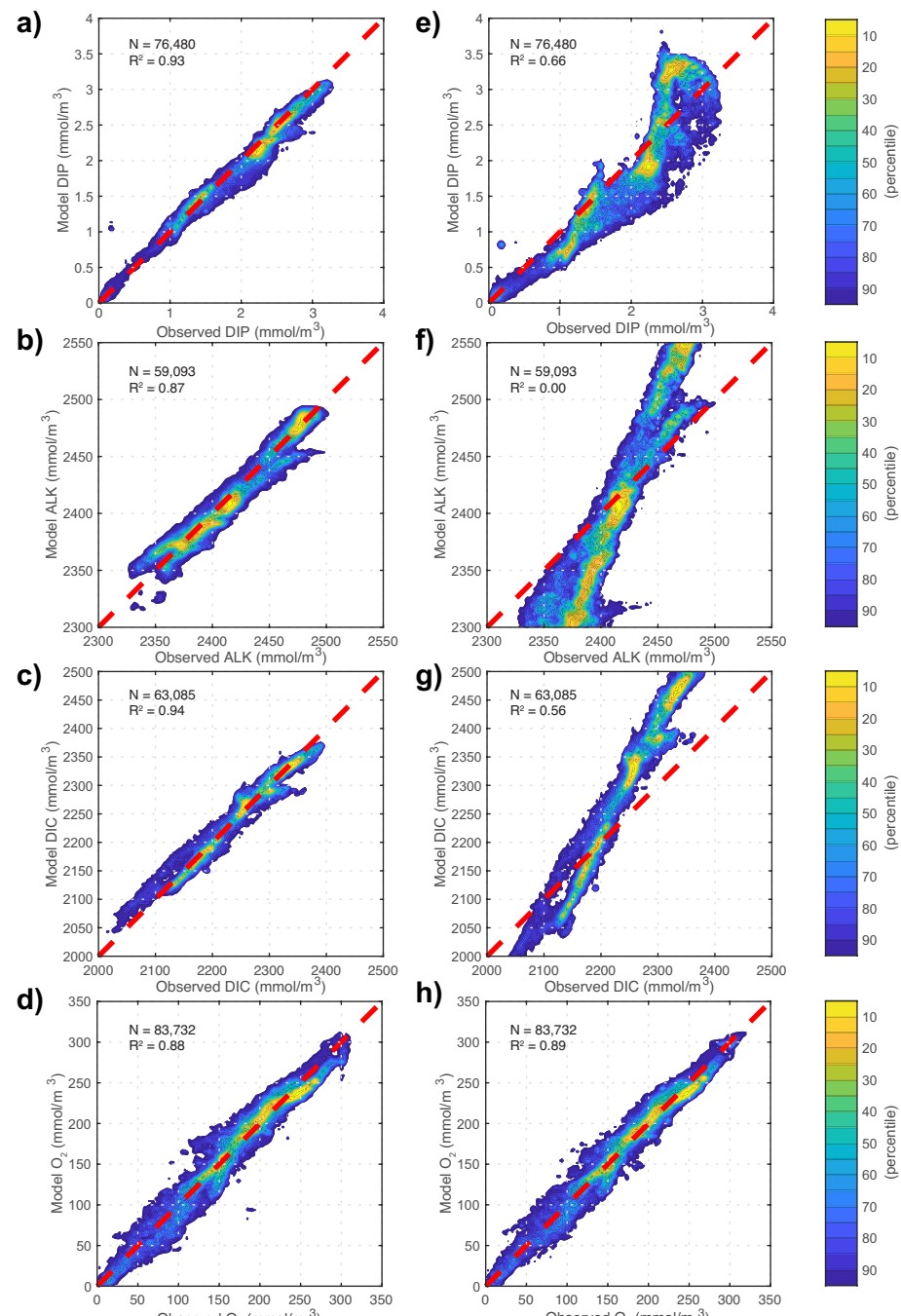

**Extended Data Fig. 3 | Tracer–tracer comparison for DIP, ALK, DIC and O₂.**
**a**–**d**, Observations and optimal model. **e**–**h**, Model constrained using only O₂ and DOC based on the SeaWiFS CAFE NPP field. The plot shows the joint density distribution for the modelled and observed tracer concentrations. The volume under the distribution integrates to 100th percentiles. The colour indicates the fraction of the distribution that falls outside the given contour. The dashed red line shows the one-to-one line. The optimal model captures 93%, 87%, 94% and 88% of the spatial variance of the GLODAPv2 DIP, ALK, DIC and O₂ data, respectively, whereas the model constrained using only DOC and O₂ captures 66%, 0.0%, 56% and 89% of the spatial variance of the GLODAPv2 DIP, ALK, DIC and O₂ data, respectively.

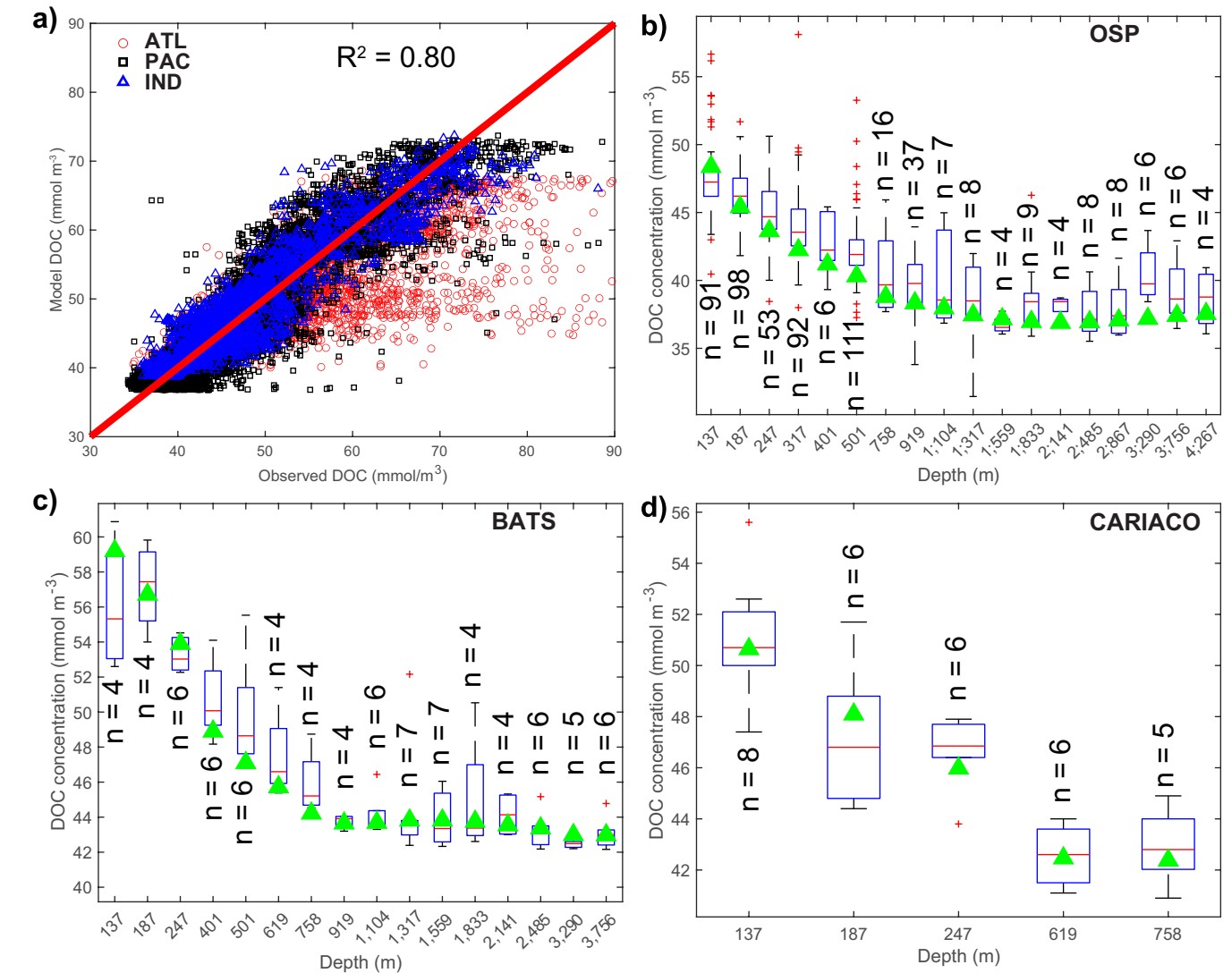

**Extended Data Fig. 4 | Comparison of model DOC (DOC + DOC_r + DOC_l) based on SeaWiFS CbPM NPP field to observations. a**, Tracer–tracer comparison for DOC between the observations and the optimal model. Red circles show DOC observations in the Atlantic Ocean, black squares in the Pacific Ocean, blue triangles in the Indian Ocean and green stars in the Arctic Ocean. The red line shows the one-to-one line. The model captures roughly 80% of the spatial variance of the DOC data. **b**–**d**, Comparisons of model DOC to those measured at ocean stations at different depths. The in situ DOC measurements are interpolated to the model grid. The numbers above/below each box represent the number of measurements at each depth. The box plots summarize the distributions of in situ measurements, which show the 25th, 50th and 75th percentiles binned according to the DOC concentration. The whiskers cover 99.3% of the data, with the remaining points shown as red crosses.

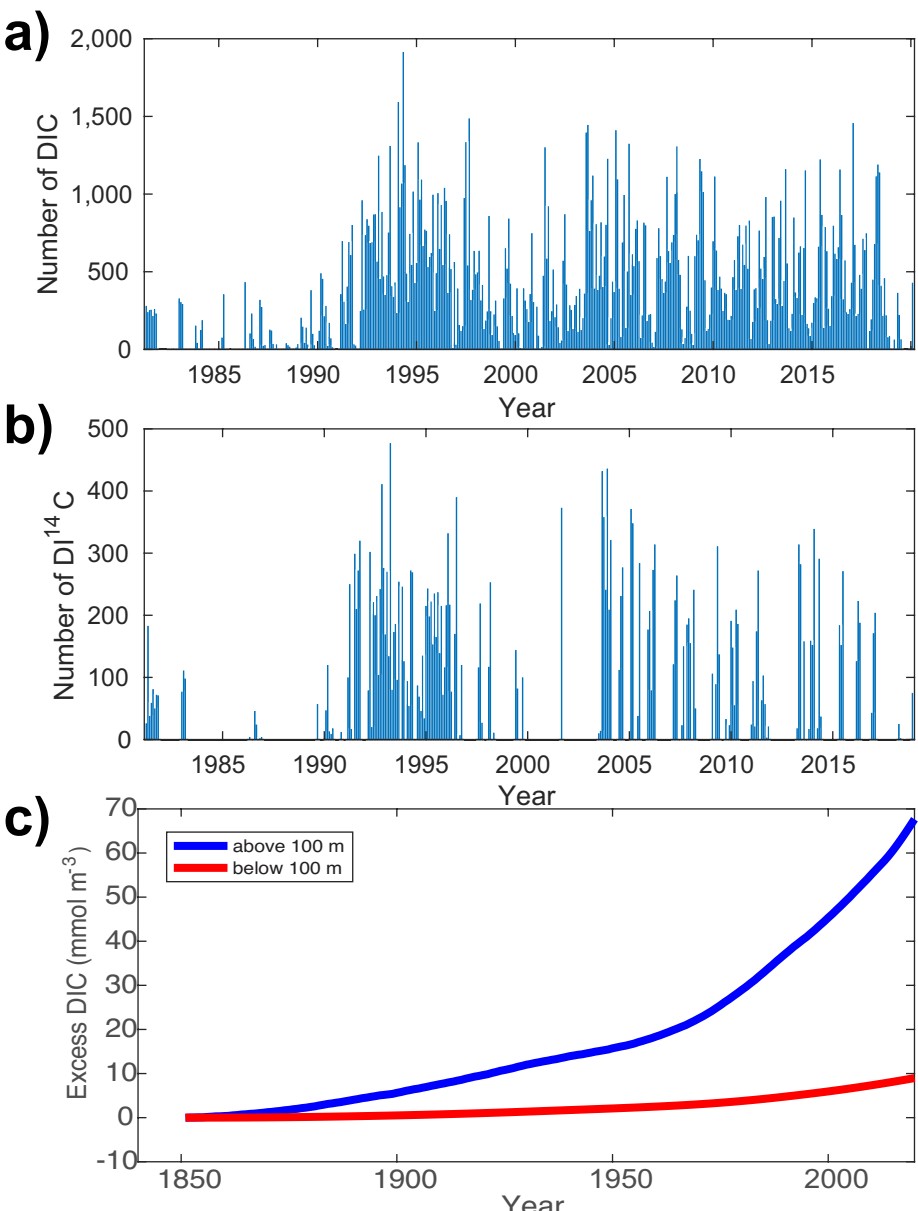

**Extended Data Fig. 5 | DIC and DI¹⁴C observations. a**, Number of hydrographic DIC measurements per month in the GLODAPv2 database as a function of time. **b**, Number of hydrographic δ¹⁴C measurements per month in the GLODAPv2 database as a function of time. **c**, Estimated excess DIC (DIC($t$) − DIC(1850)) computed by averaging the DIC concentration of our optimized model over the top 100 m (blue) and below 100 m (red). For reference, the estimated average background DIC concentration in 1850 was 2,046.8 mmol m⁻³ for the top 100 m of the water column and 2,308.4 mmol m⁻³ for the water column below 100 m, implying a reduction in the vertical DIC gradient of approximately 20% owing to the invasion of anthropogenic $CO_2$ into the ocean. This reduction masks the true strength of the biological pump, unless it is properly accounted for in the model.

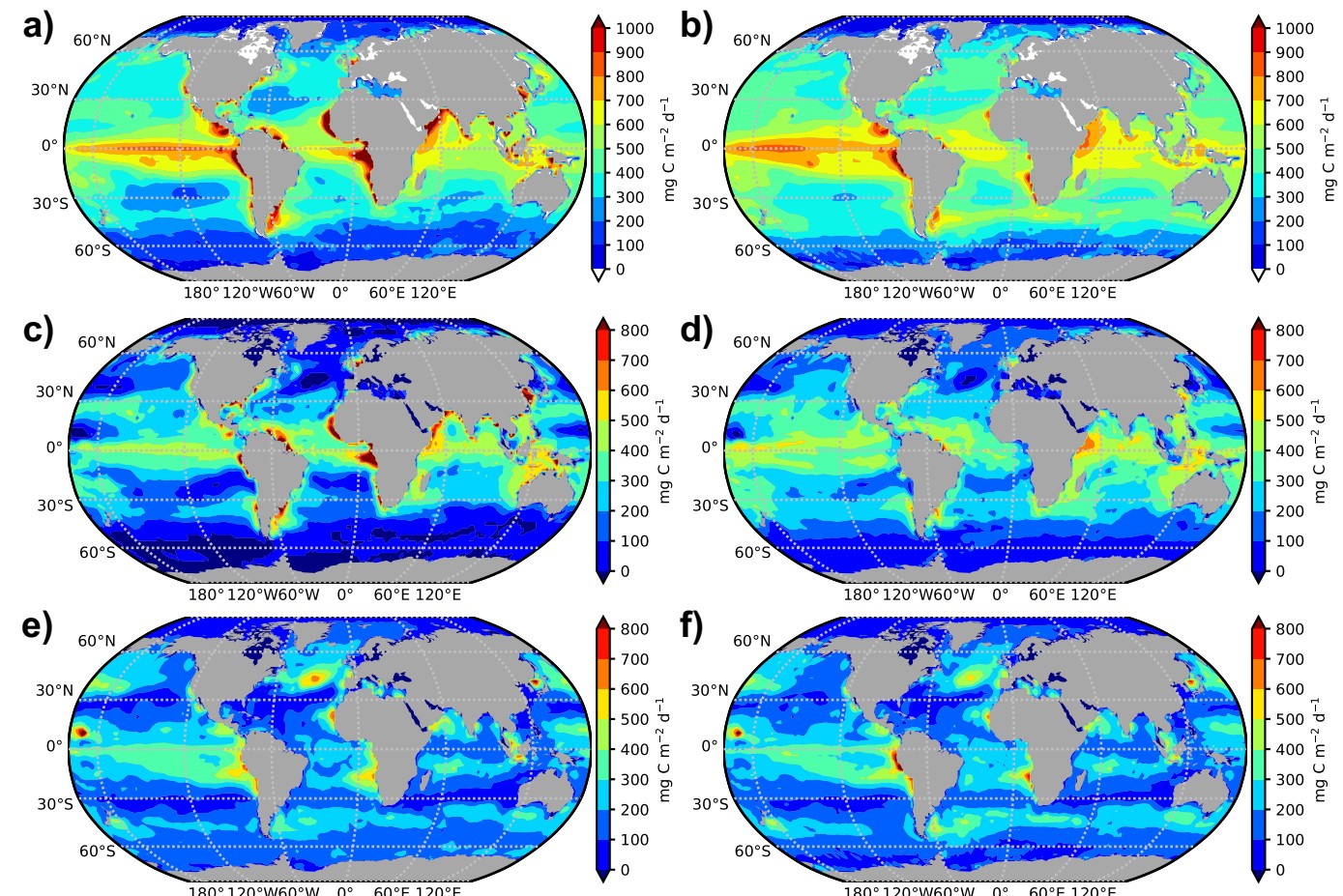

**Extended Data Fig. 6 | Model NPP and organic carbon production. a,b**, The NPP patterns based on SeaWiFS CbPM and CAFE products. **c,d**, The model production of labile organic carbon. **e,f**, The model production of semi-labile and POC. The left column (**a,c,e**) is based on the CbPM NPP product and the right column (**b,d,f**) is based on the CAFE NPP product.

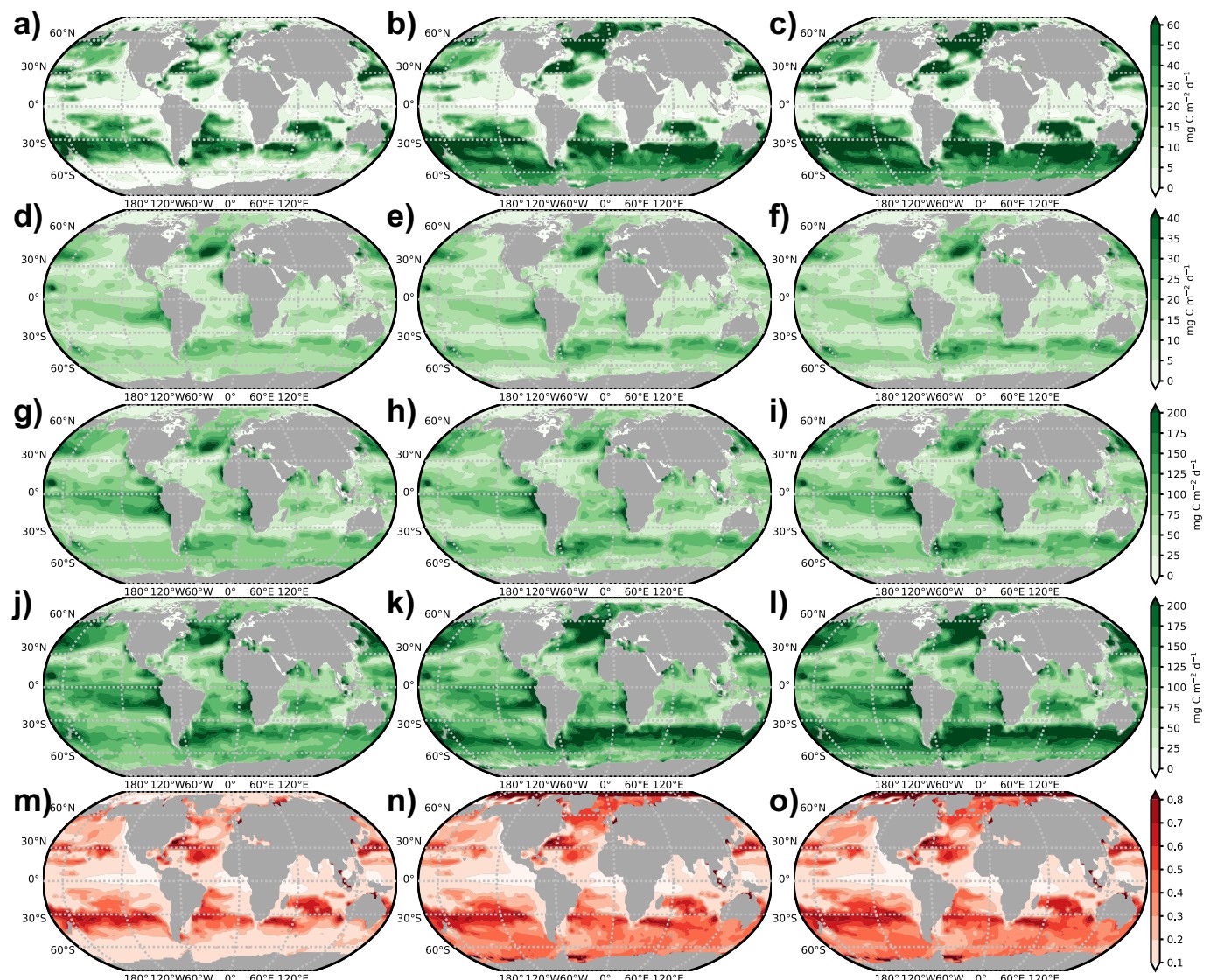

**Extended Data Fig. 7 | Export fluxes and ratios of advective-diffusive to total flux calculated at the base of the model euphotic zone (73.4 m). a–c**, The advective-diffusive flux of labile organic carbon. **d–f**, The advective-diffusive flux of semi-labile organic carbon. **g–i**, The non-advective-diffusive flux. **j–l**, The flux of TOC. **m–o**, Ratios of advective-diffusive flux to total flux. The left column shows results based on the CbPM NPP product and an e-folding remineralization time of 24 h for labile DOC, whereas the middle and right columns are based on the CAFE NPP product and e-folding remineralization times of 12 h and 24 h for labile DOC, respectively.

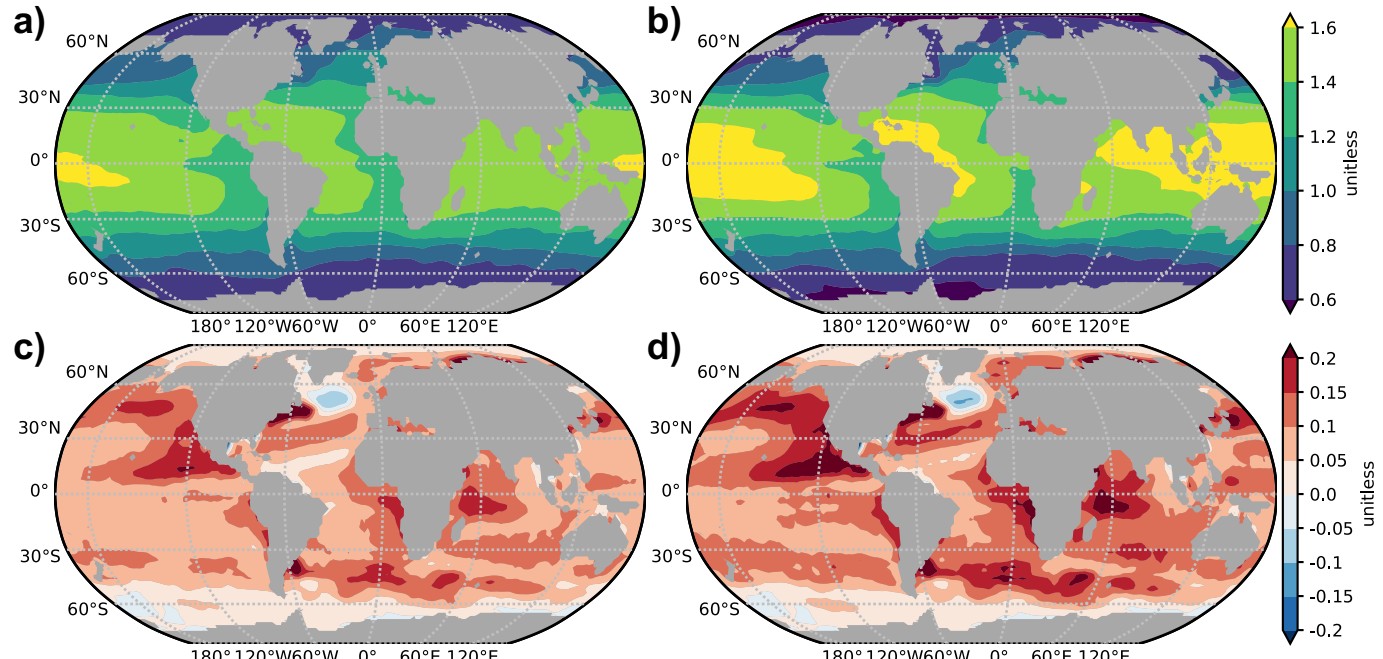

**Extended Data Fig. 8 | Distributions of exponent *b* values for the non-advective-diffusive carbon flux. a,b,** The optimal *b*-value distributions based on SeaWiFS CbPM and CAFE products, respectively. **c,d,** The projected change in the *b*-value according to temperature prediction by a CESM-BGC model prediction under the RCP8.5 scenario in the year 2099 (ref. 51). Larger *b*-values implies that respiration occurs nearer the sea surface.

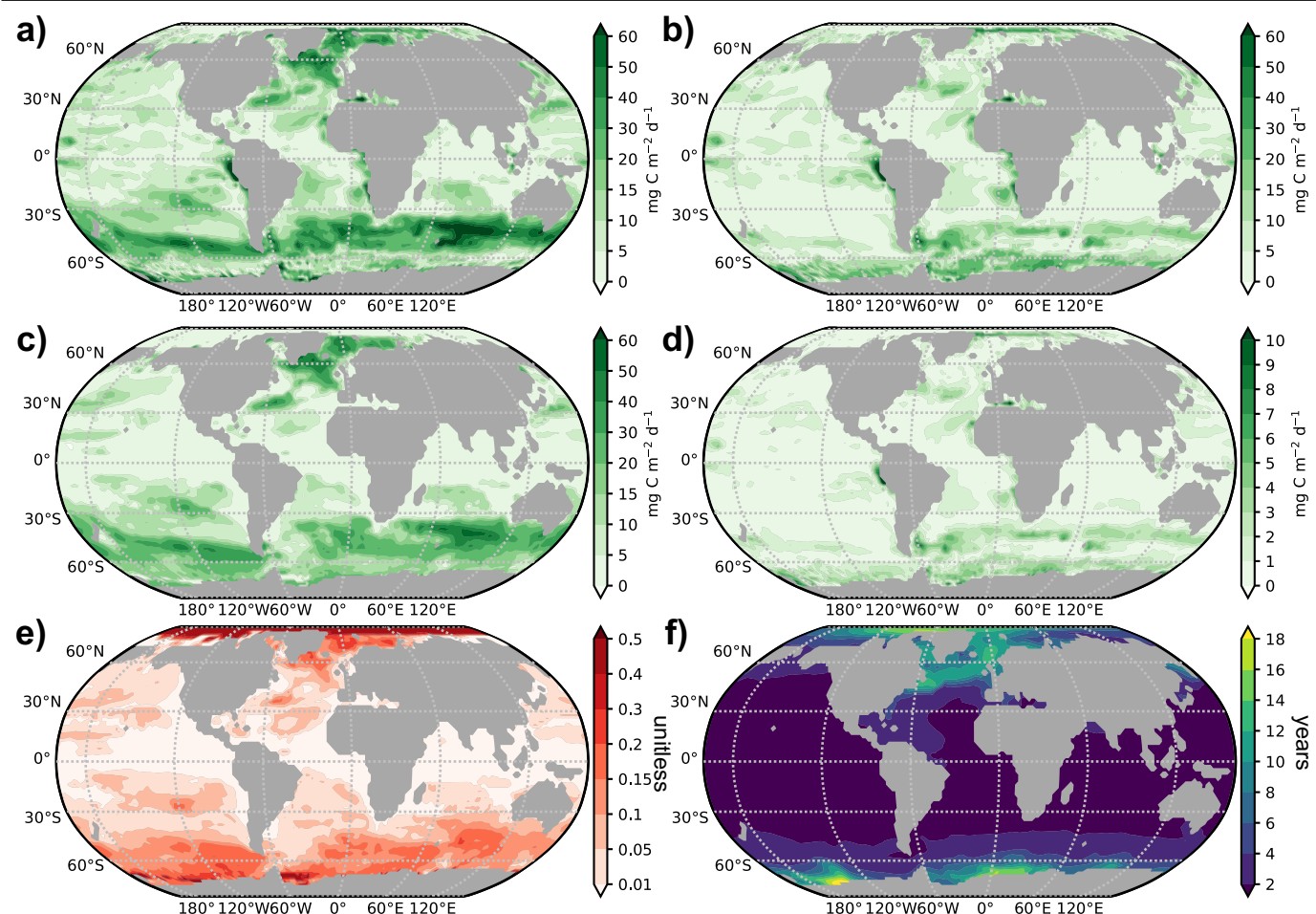

**Extended Data Fig. 9 | Distributions of standard deviations. a**, Standard deviations of TOC flux. **b**, Standard deviations of non-advective-diffusive flux. **c**, Standard deviations of advective-diffusive flux by labile DOC. **d**, Standard deviations of advective-diffusive flux by semi-labile DOC. **e**, Standard deviations of the ratio of advective-diffusive flux to TOC flux. **f**, Standard deviations of DOC residence time in years. These standard deviations are computed from four distinct model configurations, which hinge on two distinct NPP products, namely, CbPM and CAFE, along with two varying e-folding remineralization timescales for labile DOC, specifically, 12 h and 24 h.

**Extended Data Table 1 | Most probable model parameter values with their uncertainties (±1σ)**

| Parameters | M1 | M2 | Representations | Units |
|---|---|---|---|---|
| $\sigma_P$ | $0.35^{+0.01}_{-0.01}$ | $0.48^{+0.01}_{-0.01}$ | Fraction to semi-labile DOP | Unitless |
| $Q_{10P}$ | $2.28^{+0.04}_{-0.04}$ | $1.62^{+0.02}_{-0.02}$ | * | Unitless |
| $\kappa_P$ | $1.86^{+0.06}_{-0.07} \times 10^{-8}$ | $2.95^{+0.15}_{-0.14} \times 10^{-8}$ | * | $s^{-1}$ |
| $b_{P\theta}$ | $0.01^{+0.00}_{-0.00}$ | $0.01^{+0.00}_{-0.00}$ | † | $°C^{-1}$ |
| $b_P$ | $0.80^{+0.01}_{-0.01}$ | $0.83^{+0.01}_{-0.01}$ | † | Unitless |
| $\alpha$ | $2.43^{+0.05}_{-0.05} \times 10^{-8}$ | $1.75^{+0.07}_{-0.06} \times 10^{-8}$ | See Eq.1 | $s^{-1}$ |
| $\beta$ | $0.45^{+0.01}_{-0.01}$ | $1.42^{+0.05}_{-0.05}$ | See Eq.1 | Unitless |
| $\sigma_C$ | $0.09^{+0.00}_{-0.00}$ | $0.10^{+0.00}_{-0.00}$ | Fraction semi-labile DOC | Unitless |
| $\kappa_{ru}$ | $5.74^{+0.11}_{-0.11} \times 10^{-12}$ | $8.69^{2.04}_{1.62} \times 10^{-12}$ | ‡ | $s^{-1}$ |
| $\kappa_{rd}$ | $2.99^{+1.35}_{-1.35} \times 10^{-12}$ | $3.02^{+0.01}_{-0.01} \times 10^{-12}$ | ‡ | $s^{-1}$ |
| $\eta$ | $0.98^{+0.04}_{-0.04}$ | $0.98^{+0.00}_{-0.00}$ | Fraction to rDOC | Unitless |
| $b_{C\theta}$ | $0.03^{+0.00}_{-0.00}$ | $0.04^{+0.00}_{-0.00}$ | † | $°C^{-1}$ |
| $b_C$ | $0.74^{+0.01}_{-0.01}$ | $0.65^{+0.01}_{-0.01}$ | † | Unitless |
| $d$ | $4550^{+29}_{-29}$ | $4860^{+36}_{-36}$ | § | m |
| $Q_{10C}$ | $1.05^{+0.01}_{-0.01}$ | $1.01^{+0.01}_{-0.01}$ | * | Unitless |
| $\kappa_C$ | $5.42^{+0.07}_{-0.07} \times 10^{-9}$ | $5.75^{+0.09}_{-0.09} \times 10^{-9}$ | * | $s^{-1}$ |
| $r_{Si}$ | $0.10^{+0.02}_{-0.03}$ | $0.10^{+0.00}_{-0.00}$ | ‖ | $\mu M^{-1}$ |
| $r_{RR}$ | $2.34^{+0.02}_{-0.02} \times 10^{-2}$ | $2.30^{+0.03}_{-0.03} \times 10^{-2}$ | ‖ | Unitless |
| $cc$ | $8.38^{+0.75}_{-0.70} \times 10^{-4}$ | $4.49^{+0.19}_{-0.20} \times 10^{-3}$ | ¶ | $\mu M^{-1}$ |
| $dd$ | $8.83^{+0.10}_{-0.10} \times 10^{-3}$ | $0.91^{+0.02}_{-0.02} \times 10^{-3}$ | ¶ | Unitless |
| $r_{O:C}$ | $1.77^{+0.00}_{-0.00}$ | $1.79^{+0.00}_{-0.00}$ | O:C ratio | Unitless |

The definitions of the parameters are presented in Methods. M1 and M2 are models parameterized according to SeaWiFS CbPM and CAFE, respectively. The shaded column represents the results reported in the main text.

*$k_{dX} = \kappa_{X\theta} Q_{10}^{(T-30)/10}$, semi-labile DOP or DOC remineralization e-folding time.

†$b(X) = b_{X\theta} T + b_X$, Martin curve exponent, in which $T$ is the average surface ocean temperature (upper approximately 100 m), X represents C or P.

‡Reciprocals of e-folding remineralization time of refractory DOC in the upper ($\kappa_{ru}$) and deep ($\kappa_{rd}$) oceans.

§Length scale of PIC dissolution.

‖$R_{RR} = r_{Si}[SiO_4^{4-}] + r_{RR}$, function of PIC to POC production ratio.

¶Ratio of P:C: $r_{P:C} = cc[DIP] + dd$.