## [Peer Review File · Nature]

Manuscript Title: Biological carbon pump estimate based on multi-decadal hydrographic data

Redactions – unpublished data

Reviewer Comments & Author Rebuttals

Reviewer Reports on the Initial Version:

Referee #1 (Remarks to the Author):

Referees' comments:

Review comment Nature-10-16921

Reducing the uncertainty in the estimate of biological carbon pump strength is one of the key priorities across the global carbon cycling and climate fields, particularly in the context of the proposed marine CO₂ removal strategy (mCDR). The study by Wang et al., leverage the data-driven inverse model, which is based on the global hydrographic datasets (including nutrients, oxygen, DIC, DOC, TA) to tune a set of model parameters, and provide a state-of-art estimate in magnitude and spatial pattern of global carbon export above the sunlit ocean.

One of the groundbreaking merits of this work is solving the long-standing caveat persisting in prior models, and explicitly integrating the carbon export attributed by distinct pathways (including gravitational sinking, zooplankton migration, physical-induced mixing, and transport, etc). This is an extremely timely work in the community, aligning well with the perspective proposed by Boyd et al., (Nature, 2019; a review paper highlighting the necessity for the integrated analysis of distinct carbon export pathways to obtain a more realistic constraint in biological pump potential). With this significant improvement, Wang et al., report an upward biological pump strength (14 Pg C yr⁻¹), which has a far-reaching impact on the reconciliation of the classical conundrum regarding the carbon imbalance between upper layer supply and demand at depth. Secondly, the model is insensitive to the choice of net primary production products and the projection is also well supported by the independent observations of particle export flux determined by the sediment trap at the three time-series locations. The robust estimate (14±0.38 Pg C yr⁻¹) helps us substantially narrow the present large uncertainty in the biological pump strength (5-12 Pg C yr⁻¹). Mechanically, this novel model provides deep insight into the geographic pattern of distinct carbon export pathways. Particularly, the substantial contribution of downward mixing DOC in the high latitude regions helps explain the puzzle of the inverse relationship between ocean productivity and POC flux.

Given the novelty and broad scientific impact, I endorse this work being published in Nature after further modification. Overall, the paper is well-written and logically organized. Tables and figures are shown effectively, and results support the main conclusion. Nevertheless, I think further improvements are required, particularly in model uncertainty analysis and the result interpretation, in order to reach the quality of Nature. Below are my detailed comments.

Major comments :

1) model uncertainty analysis

In the present version, the authors report a pretty low error in the global carbon export estimate (0.38 Pg C yr⁻¹) after considering errors inherited with the choice in the NPP products and uncertainties in the tuned parameters. Beyond that, it is necessary to include more compressive error sources in the uncertainty analysis.

1) NPP products: the present analysis shows the different choices in NPP products are the primary factor responsible for the model uncertainty. I would suggest including another commonly used NPP product (CAFÉ) in error analysis but removing MODIS-CbPM (because the discrepancy of NPP products would be more pronounced compared with the difference related to remote sensing platforms).

2) Training dataset coverage: The majority of shipboard data used for model training were collected during the spring/summer, resulting in an underrepresentation of harsh fall and winter conditions. Particularly, compared with other datasets, DOC remains fairly sparse. I think it requires further analysis or discussion to address how the uneven coverage of the hydrographic dataset impacts the model projection.

3) Depth integration: In this study, for simplicity, the authors defined the base of the euphotic zone as either 74m or 114m. In the real ocean, the euphotic zone might be somewhat different with two choices (i.e., the euphotic zone in the subtropical gyre usually extends to 150m). POC flux is quite sensitive to the integration depth due to the exponential decay pattern. For example, the recent studies global POC flux rapidly decrease from 100 5.7 Pg C yr⁻¹ at the base of the euphotic zone to 2.8 Pg C yr⁻¹ at the depth horizon of 150 m. It would be helpful if the authors can propagate the variability and error of the euphotic zone definition (i.e., assign 10-20m uncertainty in the euphotic zone definition in the uncertainty estimate).

4) Impact of anthropogenic CO₂ in DIC cycling: Since the DIC dataset spans a long temporal coverage, penetration of anthropogenic CO₂ into the ocean interior will leave an imprint in the deep DIC samples. Omitting these long-term trends will result in an overestimation of DIC accumulation (which might lead to an overestimation of biological respiration). Some analysis or discussion is required.

Other comments (encompassing major comments on the result interpretation):

Line 47: how are the novelty and results of your work compared with the prior effort in using the same approach (inverse model) to estimate global carbon export? It would be helpful to add some relevant discussion somewhere.

Line 70: need to define the DOC when it came out for the first time in the main text.

Line 89: it is worthwhile to briefly summarize how the export is computed in your model (based on the respiration at the subsurface, lines 665-674) herein to make the main text more self-explanatory.

Line 140: In the prior paragraph, you mentioned that DOC flux includes suspended POC; however, in this part, the partitioning of DOC flux seems to exclude the suspended POC. I have some trouble understanding the logic herein.

Line 124: the similarity between sediment trap observation and your estimate (the latter is supposed to capture more comprehensive carbon export pathways) also might be the fact that exported carbon via the multiple pathways will be further transformed into the aggregates or the large sinking particle via a variety of physical and biological processes during their journey to the depth. Thus, they are also partly captured by traps. See the relevant discussion in Boyd et al., (2019).

Line 110: The elevated estimate of POC flux from your model output (almost triple ^{234}Th -POC flux) is also a key factor leading to a higher total organic carbon export than prior estimates. The implication is other particle injection pumps contribute substantially to the bulk POC flux. Given that it is a key result, I would suggest adding more discussion on this point. For example, a more detailed discussion about the different types (i.e., seasonal lipid pump, mesopelagic-migrant pump) of migration pump and their implication in carbon storage. Typically, POC export mediated by zooplankton tends to have a longer carbon sequestration time because of its rapid transfer from the surface to the depth. In addition, I note that your POC flux estimate seems to align with the total POC carbon export rates estimated by Boyd et al., (Nature, 2019; gravitational pump of 7 Pg C yr^{-1} + all particle injection pump of 4 Pg yr^{-1}). But it looks like you attribute the physically-mediated POC export (due to the eddy subduction or mixed layer change) to the DOC pathway. It might require some clarification.

Line 110: Since you rely on the bulk respiration at depth to infer the upper carbon export, it can avoid the double-counting issue (it usually happens if researchers sum up different carbon export pathways from the upper layer or different observing platforms) and also nicely capture the footprint of episodic carbon export (i.e., eddy subduction pump). If so, it might be worthwhile pointing out this merit somewhere.

Line 182: In the manuscript, the author conceptually solely attributes the zooplankton-mediated carbon export to the POC portion. I don't think this classification is very accurate. As far as I know, diel and seasonal zooplankton migration also contributes significantly to the vertical DOC export (grazing in the upper layer but excretion of DOC at depth, aka active carbon transport). Therefore, the higher DOC portion in the high latitude might be also related to the enhanced zooplankton migration. It requires some discussion.

Line 178: what does DOCI stand for? Labile DOC export? Need to define all the abbreviations in the main text.

Line 246: The fraction of summer exported carbon that does not escape from the deepest winter mixed layer will be re-entrained and return back to the atmosphere. When comparing with the mesopelagic demand, it would be an opening-eye statement to link with the carbon export estimate at the base of the annual deepest mixed layer.

Line 245: it would be better to keep the unit of aNCP consistency throughout the manuscript (I found it swings among $\text{mmol m}^{-2} \text{ yr}^{-1}$ vs $\text{mol m}^{-2} \text{ d}^{-1}$, $\text{mmol m}^{-2} \text{ yr}^{-1}$ vs $\text{mol /m}^{-2} \text{ /yr}^{-1}$).

Line 144: The geographic pattern of DOC to the total organic carbon in your study is opposite to a recent study by Roshan et al., (10.1038/s41467-017-02227-3). It requires some discussion about the potential reason leading to the discrepancy.

Line 525: should define u and K .

line 672: Again, it is quite confusing to me how you account for the contribution of suspended POC in the DOC export.

Figure S1- S2: the text and label in the axis are incomplete or missing.

Fig S3: 2 alongside O should be subscripted.

Figure S11-12: To facilitate the comparison, I would suggest putting the same prediction parameters from three NPP products in the same rows (or the same figure).

Line 79: Given that hydrographic data are used to tune the model parameters, the validation presented might be not totally independent. Correct me if I am wrong.

Line 663: After you finished up model optimization, how did you make the global projection and get the annual carbon export? Did you apply the tuned parameters to the gridded monthly NPP product and then obtain the different processes? In addition, there are considerable missing values of remotely sensed NPP, particularly in the high-latitude winter season. I am wondering how authors deal with this issue. Need to be clarified.

Referee #2 (Remarks to the Author):

Review of Wang et al. "A robust biological carbon pump revealed by multi-decadal hydrographic tracer data"

This manuscript presents an alternative value for the magnitude of the biological carbon pump that incorporates both particulate and dissolved organic carbon pathways. The manuscript provides an interesting additional estimate of the BCP to add to the several already published estimates. The value given by this work is substantially higher than previous estimates, even those which (like this work) include gravitational sinking, mixing and advection of suspended material, and vertical transport of zooplankton (e.g. Nowicki et al., 2022, GBC). Like this manuscript, the work of Nowicki et al. (2022) is also based on distributions of observed tracers such as oxygen and DOC but utilises a data-assimilating model ensemble to estimate BCP strength.

Major comments:

The authors need to elaborate why their model should be considered an improvement to, or more robust than, Nowicki's. Otherwise their results, although very worthy of publication in a specialist journal, don't merit publication in Nature.

The "biogeochemical implications" section is important, but rather unconvincing as currently written and needs strengthening (see specific comments below). In particular, the results don't add any information on the mechanistic understanding of contributions of different pathways.

Specific comments:

Line 27: specify here that the estimate of 14.26 GtC is the total organic carbon export (i.e. POC+DOC)

Line 42: NPP and SST are used to extrapolate the sparse in situ observations to the global scale. This is slightly different from what is implied here (that NPP and SST are used to study regional differences).

Line 51: another reason for the discrepancies is that the observations are (very often) single points in time and space and so the full annual cycle of, e.g. contributions by migrating zooplankton, cannot be captured.

Line 92: the use of the term "active export" to refer to POC export is rather confusing here as the term "active flux" has already been associated for many years with the specific portion of the flux attributable to vertical migration of zooplankton and other organisms. I suggest finding another term.

Line 92: DOC and slowly sinking POC are different (and slowly sinking POC is different from suspended POC), but here it appears they are all lumped together. As several observational approaches to estimating flux, e.g. sediment traps, Th234 etc., capture both fast and slow sinking POC, when comparing to the observations this way of distinguishing the different components of the flux might introduce a bias.

Lines 95-98: references needed to support the statements here

Lines 98-100: I didn't understand well the point being made in this sentence.

Line 122-123: if the data being shown is obtained from traps, then the station POC flux estimates do in fact include fecal pellets, and therefore capture some portion of the active flux.

Line 130: not clear to me what is meant by "oversampling" here

Line 180-182: until this point, the concepts of the mixed layer pump, subduction pump etc. haven't been introduced. Worth doing so earlier on in the introduction. Also, some explanation needs to be added on why the increase of DOC export is consistent with the mechanisms of these pumps.

Lines 194-195: specify that, of the 2 numbers reported, the model result is given first followed by the observations.

Line 199: here the model result is compared only to the Laws algorithm – what about other satellite-based estimates of export?

Lines 199-200: provide the numbers for ANCP at BATS

Line 202: Figure 4d doesn't exist

Lines 204-205: I think I understand what is meant here, but it's written badly

Lines 206-207: the negative correlation refers to the export ratio vs PP (not export flux vs PP), and furthermore has only been demonstrated for the Southern Ocean

Lines 212-214: this is a lag issue again though, no? see for example Henson et al. (2015, GBC) and Giering et al. (2017, GBC)

Line 214: high export, low productivity conditions have been found episodically too (Henson et al., 2019, GBC)

Lines 218-235: this is a not very insightful or useful paragraph – basically the conclusion is that there are no clues from this study about why certain patterns of export flux are found in observations.

Lines 224-225: I didn't understand the reasoning why this line of evidence shows that "export of DOC may be an essential process"

Lines 233-234: which are the "warm high PP regions"? upwelling zones?

Lines 234-235: this final sentence emphasises that the results shown here shed no mechanistic light on the causes of low export in high PP zones – so why have a whole paragraph speculating about a debate that your results make no contribution to?

Line 254: The models of Nowicki et al. and Devries and Weber (2017, GBC) do account for DOC, and the former also includes vertical migration.

Line 258: "we find that DOC flux..." or indeed slow sinking or suspended POC as these 3 elements are separated in the model (at least as I understood it from lines 92-93)

Lines 264-265: indeed, an improved mechanistic understanding is needed – but this study does not provide that as emphasised in lines 276-277. This is important, as without including mechanistic representation of processes that are considered key to export flux, the response to future climate change cannot be well-predicted.

Figure 1: why have a power-law dependence for the POC flux attenuation, but an exponential dependence for the PIC flux attenuation??

Figure 2: the colour scale on plot a) is unhelpful for distinguishing regional variability. The data should be plotted in $gC\ m^{-2}\ yr^{-1}$ to make the results directly comparable to plots in previous work such as Henson et al. (2012, GBC) or Siegel et al. (2017, GBC). Are the data shown in panels b-e) annual estimates, or

point measurements from the sites? As the model only produces a climatological annual estimate I assume it's the former? What time period are the data averaged over (i.e. it's multiple years, as a guess)? For panel e), change the y-axis so the vertical differences can be seen (include a broken y-axis so the outlier at 137 m depth can still be seen).

Figure 4: panel a) has another unhelpful colourscale. Again, plot in $\text{gC m}^{-2} \text{ yr}^{-1}$ to be comparable with previous studies. Figure 4 caption – I'm not clear what reference depth is used here. Is it the deeper of the euphotic zone depth and the mixed layer depth?

Equation 1: shouldn't the conditional be "if $z > z_c$ "?

Line 532: what sinking speed is chosen? Are the results sensitive to this choice?

Lines 535-536: I'm very surprised that doubling the remineralisation rate doesn't significantly change the results! Surely this should be a first order control?

Lines 536-539: there is a big assumption here which isn't explored in the paper – that b is a function of temperature

Section 4 of Methods: again it wasn't clear from the text here how the export depth horizon is defined

Table S1: an additional column explaining in words what the parameters are would be extremely helpful and stop the reader having to flick between the equations in the methods section and the supplementary material. There are some huge (1-2 orders of magnitude) differences in the estimated parameter values depending on which NPP model is used, e.g. for bp -sigma, beta, rsi and dd. This implies that the model is highly sensitive to the input fields and that multiple different non-exclusive solutions to the optimisation are possible. This doesn't give much faith in the robustness of the model results. The implications of this should be discussed in the manuscript.

Figures S1 and S2: "fraction of the grid boxes" – does this mean vertically?

Figures S3-S5: all basically indistinguishable – are they all needed?

Figures S6-S8: again all basically indistinguishable

Figure S9: is this figure necessary?

Figures S10-S11: again basically indistinguishable

Figure S12: where are the plots with $k_p = (1/30) \text{ day}^{-1}$ – to compare the results of this sensitivity test to? And finally, most of the supplementary figures aren't referenced in the text.

Referee #3 (Remarks to the Author):

This is a thorough, clearly written, and potentially highly impactful study that quantifies the strength of the ocean biological carbon pump, something that is highly uncertain yet key for our understanding of the global carbon cycle. The most noteworthy finding in the study is the estimate of global export production at $\sim 14 \text{ PgC/yr}$ – substantially higher than estimates from satellites and CMIP models. Of this 14 PgC/yr , the study suggests that $\sim 12 \text{ PgC/yr}$ is particulate/gravity export, while $\sim 2 \text{ PgC/yr}$ is dissolved/circulation export. These quantities are determined from a sophisticated inverse modeling approach and constrained by hydrographic observations of biogeochemical quantities.

The modeling approach is sound and the authors have taken great care to explore the sensitivity of

parameter values to various assumptions. For the modeling approach, I have a few minor comments to consider:

-Please address the uncertainties in the OCIM transport matrix, and the influence of these uncertainties on the parameter values.

-Please quantify the sensitivity of the carbon model parameters to the air-sea CO₂ flux parameterization.

-Please address the significant deviations from the 1:1 line on the observed vs. modeled oxygen density distribution plots (Figs S3-S5) at the high oxygen values (225-275 mmol/m³) – these appear regardless of the NPP algorithm selected.

I have two other major points for the authors to consider:

1) While reading, I found myself actively comparing the approach and findings to that of the recent Nowicki et al. paper (<https://agupubs.onlinelibrary.wiley.com/doi/10.1029/2021GB007083>). The export production

estimated in this study (14 PgC/yr) is substantially larger than that of the Nowicki study (10 PgC/yr), yet the approaches are very similar (inverse modeling, constrained by hydrographic observations). This leaves me wondering which approach produces the correct answer. The onus is on this study's authors to demonstrate that their approach is superior, or at least that it is different enough to warrant their much higher estimates. As it stands, the manuscript has only a single sentence that addresses the discrepancy in the findings, and additional discussion is needed.

There are probably multiple ways to demonstrate this point for the reader -- I offer one suggestion here, but it would require substantial effort: Use one of the CMIP models as a testbed for your approach. First, generate a transport matrix from the CMIP model's CFC, T, S, etc. Then, use the CMIP model's NPP in your top down inverse model, and the model's subsampled biogeochemical tracers to constrain the parameters in the inverse model. Finally, quantify how well the inverse estimate of, say, globally integrated POC export, matches that of the original CMIP model. If you wanted to go all-out in this effort, you could also apply the Nowicki model approach to the subsampled CMIP model and then compare the results.

2) One of the hallmarks of a strong paper is when the reader can think of ways to build off the new findings or approach described in the paper. Effective papers are capable of influencing scientific research not only in their subfield or discipline, but also more broadly. Yet, as written, the paper's broader impacts are rather narrow: explaining the relationship between NPP and POC export, the relative importance of POC and DOC in export, and regional distribution of DOC export. I encourage the authors to think even bigger in their impacts, and to address how their findings will influence, e.g., the global carbon cycle, changes in the ocean biological carbon pump and thus atmospheric CO₂ on glacial-interglacial timescales, etc.. I would like to see some of these described in a final or near-final paragraph in the paper.

Because this is a potential nature paper and thus the authors will have their manuscript professionally

edited, I did not focus on in-line text corrections, though I noted that the transport matrix (T) should be bold-faced in all of the model equations, so as to not confuse it with temperature.

Author Rebuttals to Initial Comments:

General comments to all three reviewers and to the editor

We thank all three reviewers for carefully reviewing our manuscript and for many helpful suggestions. We particularly want to thank Reviewer #1 for prodding us to justify our treatment of the effect of anthropogenic CO₂ on our results. As it turns out, the correct treatment of the anthropogenic signal is critical, as we will explain later.

We begin by addressing the concern raised by all three reviewers that we did not adequately explain why our inverse model should be more robust than the recent inversion of Nowicki et al. (2022). In the following, we show the reviewers' comments in red, our responses in blue, and the passages from our revised manuscript in black.

To explain why we believe that our estimates are more robust, we added the following text to our manuscript on lines 102 - 114:

"... What distinguishes our model from previous inverse models^{22,23} is the small number of adjustable parameters and the simultaneous use of multiple tracers to constrain the inversion. In particular, our inverse model uses DIC measurements, which provide the most natural constraint on the BCP. Previous inverse models did not use DIC observations to avoid the need to explicitly model the transient anthropogenic carbon signal in the hydrographic DIC dataset. Here we explicitly simulated the transient DIC signal and found that it contributes a ~20% decline in the vertical DIC gradient produced by the biological pump (Fig S9c, see Methods). Furthermore, by combining ALK and DIC data with an accurate representation of the anthropogenic DIC signal, our model captures the respiration of organic carbon not oxidized by O₂²⁴. In a sensitivity test where we followed Refs^{22,23} by using only O₂ and DOC to constrain the model, we found a significant deterioration of the fits to other tracers (Fig. S5) ..."

Figure S3, reproduced here, shows our model's excellent fit to the DIP, DIC, Alk, and O₂ data. In contrast, figure S5 shows that when we follow Nowicki et al. (2022) and only O₂ and DOC to constrain our model, there is a dramatic deterioration of the fit quality to the DIP, DIC, and Alk data. We believe that multi-tracer constraints significantly increase the robustness of inverse model results.

Fig. S3 Tracer-tracer comparison for DIP, DIC, ALK, and O_2 for the observations and optimal model based on the SeaWiFS CbPM NPP field. The plot shows the joint density distribution for the modeled and observed tracer concentrations. The volume under the distribution integrates to 100 percentiles. The color indicates the fraction of the distribution that fall outside the given contour. The red dash line shows the one-to-one line. The model captures 91%, 92%, 86%, and 86% of the spatial variance of the GLODAPv2 DIP, DIC, ALK, and O_2 data, respectively.

Fig S5 Tracer-tracer comparison for DIP, DIC, ALK, and O_2 for the observations and optimal model based on the SeaWiFS CbPM NPP field. The plot shows the joint density distribution for the modeled and observed tracer concentrations. The volume under the distribution integrates to 100 percentiles. The color indicates the fraction of the distribution that falls outside the given contour. The red dashed line shows the one-to-one line. The model captures 91%, 92%, 86%, and 86% of the spatial variance of the GLODAPv2 DIP, DIC, ALK, and O_2 data, respectively.

To address Reviewer #1's comment

"4) Impact of anthropogenic CO₂ in DIC cycling: Since the DIC dataset spans a long temporal coverage, penetration of anthropogenic CO₂ into the ocean interior will leave an imprint in the deep DIC samples. Omitting these long-term trends will result in an overestimation of DIC accumulation (which might lead to an overestimation of biological respiration). Some analysis or discussion is required." we did major revisions to our manuscript, which we explain here for the benefit of all reviewers.

In the previous version of our study, we corrected for the invasion of anthropogenic DIC only roughly by adding to our pre-industrial DIC a previously-computed estimate of the anthropogenic DIC for the period of the 1990s. Then, to compute the objective function in our optimization, we pooled all the DIC bottle measurements in the GLODAPv2 database as if they were all from the 1990s. However, this crude approach did not account for the fact that the DIC data in the GLODAPv2 database was collected over four decades (see Figure S9a reproduced below), during which it was changing rapidly (see Figure S9c).

For our revised manuscript, we redid our inversion using the total (natural + anthropogenic) DIC signal, carefully computing the residual misfits at the time and locations of the available DIC measurements. We used our model to perform a time-dependent simulation from 1850 to 2020 to obtain a self-consistent estimate of the natural and anthropogenic signals. For this calculation, we took advantage of research on the cycling of ¹³C and ¹⁴C that we conducted in parallel to our work on the biological carbon pump. Specifically, we used our ¹⁴C model to re-calibrate the air-sea gas-exchange velocity using $\delta^{14}\text{C}$ observations from the GLODAPv2 database, allowing us to obtain a more accurate estimate of the transient anthropogenic DIC signal. The result significantly improved the fits of all the principal tracers used in our inversion. For example, the R² improved from 0.91 to 0.93 for DIP, from 0.92 to 0.94 for DIC, from 0.86 to 0.87 for Alk, and from 0.86 to 0.88, without any reduction to the R² = 0.80 for DOC. Moreover, the revised inversion significantly improved the fit to the available $\delta^{13}\text{C}$ data in the GLODAPv2 database, shown here, but not included in the paper because we have a second paper on the isotope model that is in preparation with a different lead author (Fu, Wang, Randerson, and Primeau, In prep.).

Redacted

The top panel shows a tracer-tracer plot for $\delta^{13}\text{C}$ using the carbon cycle model from our previous inversion, and the bottom panel shows the tracer-tracer plot for the revised model. Again, there is a remarkable improvement in the quality of the fit, even though $\delta^{13}\text{C}$ was not used to optimize our carbon-cycle model.

We revised the Methods section to explain how we now treat the anthropogenic DIC signal in our inverse model (see lines 721-754).

“2.3 Anthropogenic DIC

To use DIC observations to constrain our inverse model, we have to take into account the changing DIC concentration due to the invasion of anthropogenic CO_2 into the ocean. To obtain

a self-consistent estimate of the anthropogenic carbon signal we performed a time-dependent simulation using Eq. 3. Starting from an assumed steady state, we time-stepped our carbon-cycle model forward in time from 1850 to 2020, using an implicit trapezoid-rule time-integration scheme for all terms except for the gas-exchange where we used an explicit Euler-forward scheme. In this calculation we prescribed the surface SST according to a time-dependent reanalysis product Ref¹⁴. The transient integration was carried out with a time-step size of $\Delta t = 2$ months. The atmospheric $p\text{CO}_2$ was prescribed according to Ref¹⁵ for 1850 to 2015 and according to Ref¹⁶ from 2016 to 2020. We also simulated $\delta^{14}\text{C}$ in order to better calibrate the air-sea gas exchange velocity as described below. The atmospheric $\delta^{14}\text{C}$ was prescribed according to Ref¹⁷ for the period from 1850 to 2015 and from Ref¹⁸ from 2016 to 2020. To produce the initial conditions, we assumed that the system was in steady state in 1850 and used Newton's method to find the steady state.

To calibrate the air-sea gas exchange parameterization, we re-optimized the scaling factor in the OCIMP2 gas-exchange scheme by minimizing the misfit between our modeled $\delta^{14}\text{C}$ and the GLODAPv2 $\delta^{14}\text{C}$ data. See Fig. S9a and S9b for the number of observations as a function of time. To compute the misfit we sampled our model at the location and times of the bottle measurements in the GLODAPv2 database. Our calibration method followed an iterative two-step process in which we first optimized the air-sea gas exchange through a series of transient carbon-cycle simulations. After obtaining the optimal air-sea gas exchange, we subtracted the excess anthropogenic DIC from the GLODAPv2 measurements to produce an estimate of the natural background DIC for the year 1850. The resulting DIC data and optimal gas-exchange velocity were then used for the optimization of the biogeochemistry model (see Section 3). The optimized biogeochemical model was then used to produce an updated initial condition for the transient carbon-cycle simulation and a re-optimization of the air-sea gas-exchange velocity. We repeated this two-step process until we obtained self-consistent estimates of (i) the optimal biogeochemical parameter values (see Table S1), (ii) the biogeochemical state, (iii) the scaling factor for the air-sea gas transfer velocity, $a = 0.234 \text{ cm h}^{-1} / (\text{m s}^{-1})^2$, (iv) transient DIC, and (v) the transient $\delta^{14}\text{C}$ signal including the combined effects of radioactive decay, the Suesse effect, and the bomb radiocarbon signal. Figure 9(c) shows a time-series of the excess anthropogenic DIC concentration averaged over the top 100 m of the water column and for the water column below 100 m. By 2020 the vertical DIC gradient is reduced by 20%.”

Fig. S9 a) Number of hydrographic DIC measurements per month in the GLODAPv2 database as a function of time. b) Number of hydrographic $\Delta^{14}\text{C}$ measurements per month in the GLODAPv2 database as a function of time. c) Estimated excess DIC, ($\text{DIC}(t) - \text{DIC}(1850)$) computed by averaging our optimized model's DIC concentration over the top 100 m (blue) and below 100 m (red). For reference, the estimated average background DIC concentration in 1850 was $2046.8 \text{ mmol}/\text{m}^3$ for the top 100 m of the water column and $2308.4 \text{ mmol}/\text{m}^3$ for the water column below 100 m, implying a reduction in the vertical DIC gradient of approximately 20% due to the invasion of anthropogenic CO_2 in the ocean. This reduction masks the true strength of the biological pump unless it is properly accounted for in the model.

We now give a point-by-point response to each reviewer's comments.

Referee #1 (Remarks to the Author):

Review comment Nature-10-16921

Reducing the uncertainty in the estimate of biological carbon pump strength is one of the key priorities across the global carbon cycling and climate fields, particularly in the context of the proposed marine CO₂ removal strategy (mCDR). The study by Wang et al., leverage the data-driven inverse model, which is based on the global hydrographic datasets (including nutrients, oxygen, DIC, DOC, TA) to tune a set of model parameters, and provide a state-of-art estimate in magnitude and spatial pattern of global carbon export above the ocean.

One of the groundbreaking merits of this work is solving the long-standing caveat persisting in prior models, and explicitly integrating the carbon export attributed by distinct pathways (including gravitational sinking, zooplankton migration, physical-induced mixing, and transport, etc). This is an extremely timely work in the community, aligning well with the perspective proposed by Boyd et al., (Nature, 2019; a review paper highlighting the necessity for the integrated analysis of distinct carbon export pathways to obtain a more realistic constraint in biological pump potential). With this significant improvement, Wang et al., report an upward biological pump strength (14 Pg C yr⁻¹), which has a far-reaching impact on the reconciliation of the classical conundrum regarding the carbon imbalance between upper layer supply and demand at depth. Secondly, the model is insensitive to the choice of net primary production products and the projection is also well supported by the independent observations of particle export flux determined by the sediment trap at the three time-series locations. The robust estimate (14±0.38 Pg C yr⁻¹) helps us substantially narrow the present large uncertainty in the biological pump strength (5-12 Pg C yr⁻¹). Mechanically, this novel model provides deep insight into the geographic pattern of distinct carbon export pathways. Particularly, the substantial contribution of downward mixing DOC in the high latitude regions helps explain the puzzle of the inverse relationship between ocean productivity and POC flux.

Given the novelty and broad scientific impact, I endorse this work being published in Nature after further modification. Overall, the paper is well-written and logically organized. Tables and figures are shown effectively, and results support the main conclusion. Nevertheless, I think further improvements are required, particularly in model uncertainty analysis and the result interpretation, in order to reach the quality of Nature. Below are my detailed comments.

We thank the reviewer for their positive comments.

Major comments :

1) model uncertainty analysis

In the present version, the authors report a pretty low error in the global carbon export estimate (0.38 Pg C yr⁻¹) after considering errors inherited with the choice in the NPP products and uncertainties in the tuned parameters. Beyond that, it is necessary to include more compressive error sources in the uncertainty analysis.

We believe this issue is closely related to point 3 raised by the same reviewer because the main source of uncertainty in any estimate of export flux is the great sensitivity of the flux to the choice of depth horizon for computing the flux. We therefore addressed both issues together in the revised manuscript using the alternative perspective provided by the sequestration-time-partitioned organic matter production. See lines 30-37 and 219-243.

“Putting the non-advective-diffusive flux and advective-diffusive fluxes together, our globally integrated TOC flux is $15.00 \pm 1.12 \text{ Pg C yr}^{-1}$ at the base of the euphotic zone ($\sim 73 \text{ m}$, Fig. 4a). This number is highly sensitive to the export horizon due to strong remineralization in the upper ocean. For example, the export flux rate decreases by $\sim 30\%$ in the 27 m interval from the base of the model euphotic zone to the 100 m -depth horizon that is widely used by Earth System Models (ESMs) as a reference export depth. An alternative perspective on this sensitivity is provided by the sequestration-time distribution functions for the organic carbon production rate and for the stock of regenerated DIC (Fig. 5). The total organic carbon production with sequestration times greater than three months is $11.09 \pm 1.02 \text{ Pg C yr}^{-1}$. If we include only residence times greater than one year the total export flux decreases to $8.25 \pm 0.30 \text{ Pg C yr}^{-1}$ and if we include only residence times greater than three years the total export flux is $6.30 \pm 0.09 \text{ Pg C yr}^{-1}$. The distribution functions show that the total flux is dominated by small residence-time export, but that the small residence-time fluxes contribute negligibly to the standing stock of regenerated DIC, pointing to the rapid recycling of much of the organic matter production on short timescales. For residence times less than 1 year (yellow regions in Fig. 5), the accuracy of export fluxes is highly uncertain due to three main factors. Firstly, the circulation model lacks representation of the seasonal cycle. Secondly, the short-residence time fluxes are sensitive to the precise mathematical formulation of biological production and respiration models. Lastly, the inverse model, which is constrained by carbon, oxygen, and nutrient stocks, is insensitive to the partition of the export flux distribution that does not impact these stocks. Indeed, significant contributions to the standing stock (Fig. 5b) only become apparent when residence times approach approximately one year.”

Figure 5. Sequestration-time distribution functions for the organic carbon flux and the stock of regenerated DIC. a) Sequestration-time-partitioned organic carbon production. The curves show the cumulative net primary production fluxes with sequestration times greater than τ separated into contributions from labile DOC (red), semi-labile DOC (black), refractory DOC (green), and POC (blue). The sequestration times are measured from the time when the organic carbon is respired into DIC to the time when the regenerated DIC is transported back to the model’s 36.1 m-thick surface layer. b) Sequestration-time-partitioned standing stock of regenerated DIC. The curves show the cumulative stock with sequestration times less than τ . All curves correspond to climatological-mean estimates integrated over the whole ocean volume. The error bars, indicated by the shaded regions correspond to $\pm 1\sigma$ computed from four inverse models in which the e -folding lifetime of labile DOC was either 12 hours or 24 hours and the biological carbon production was patterned using either the CbPM or the CAFE NPP products. The posterior parametric uncertainty makes a negligible contribution to the displayed error bars. For $\tau > 1$ year (green regions) the inverse model produces a robust estimate of the export flux distribution.

We also added a section describing how the sequestration-time partitioned distribution function is computed on lines 862-876.

6. Sequestration-time partitioned distribution functions

To compute the sequestration-time partitioned distribution functions we use the three-dimensional organic carbon respiration rate to construct a Dirac δ -function pulse of labeled regenerated inorganic carbon. The resulting tracer field is then transported using the circulation model until it is removed in the model’s 36.1 m-thick surface layer using a loss frequency of (1 year / 500). We integrate the system forward in time for 10,000 years by which time all of the

regenerated-carbon pulse has left the system. We use a second-order accurate trapezoidal integration rule starting with a time-step size of less than 10^{-4} years and gradually increase it to 10 years by the end of the simulation. A sequestration-time density distribution function is obtained by globally integrating the loss rate and the cumulative distribution function is then obtained by integrating the density function for progressively longer times. To obtain the cumulative sequestration-time distribution for the stock of regenerated DIC we first integrate the tracer field over the whole volume of the ocean and then integrate the resulting stock for progressively longer sequestration times. By year 10,000 the resulting integral is equal to the global inventory of regenerated DIC.

1) NPP products: the present analysis shows the different choices in NPP products are the primary factor responsible for the model uncertainty. I would suggest including another commonly used NPP product (CAFÉ) in error analysis but removing MODIS-CbPM (because the discrepancy of NPP products would be more pronounced compared with the difference related to remote sensing platforms).

We followed the reviewer's suggestion and redid our analysis using the SeaWiFE CbPM and CAFE. The change in NPP products improved the consistency between the optimal parameters obtained from the models with the different NPP inputs compared to those used in our previous analysis. See Table S1 for the optimal parameter values.

2) Training dataset coverage: The majority of shipboard data used for model training were collected during the spring/summer, resulting in an underrepresentation of harsh fall and winter conditions. Particularly, compared with other datasets, DOC remains fairly sparse. I think it requires further analysis or discussion to address how the uneven coverage of the hydrographic dataset impacts the model projection.

Firstly, we utilized an updated compilation of DOC measurements (Hansell et al., 2021), which encompasses 25,869 valid data points after binning to the model grid (Fig. S2a). The new dataset provides better coverage compared to previously utilized ones (e.g., Letscher and Moore (2015)) that had only 14,034 valid data points in the model grid. The Hansell et al. 2022 database has a slight seasonal bias, but does include data from all seasons (Fig. S2b,c).

Secondly, a significant proportion of the total DOC is composed of refractory DOC. Unlike labile and semi-labile DOC, refractory DOC does not exhibit strong seasonality due to its long residence time in the ocean.

Lastly, we mentioned in the manuscript that we utilized tracer data from the full-water depth to constrain our model parameters. The deep ocean experiences lesser seasonal variability compared to the surface ocean. Therefore, employing full-water depth data helps to anchor the inversion's stability.

We have added the following text on lines 594-600, and included a histogram figure for monthly DIC sampling density.

“The DOC dataset has a slight seasonal bias with more samples collected in the summer season. However, we think that the influence is minor because 1) a significant proportion of the total DOC is composed of refractory DOC. Unlike labile and semi-labile DOC, refractory DOC does not exhibit strong seasonality due to its long residence time in the ocean; 2) we utilized tracer data from the full-water depth to constrain our model parameters. The deep ocean experiences lesser seasonal variability compared to the surface ocean. Therefore, employing full-water depth data helps to anchor the inversion's stability.”

Fig. S2 a) Spatial, b) and c) monthly DOC sampling density. The DOC observations, obtained from a recent compilation², are interpolated to the OCIM grid. The color denotes the fraction of the grid boxes in each water column with at least one measurement. For each water column, the sampling density is defined as the total number of grid boxes with at least one sample divided by the total number of wet grid boxes.

3) Depth integration: In this study, for simplicity, the authors defined the base of the euphotic zone as either 74m or 114m. In the real ocean, the euphotic zone might be somewhat different with two choices (i.e., the euphotic zone in the subtropical gyre usually extends to 150m). POC flux is quite sensitive to the integration depth due to the exponential decay pattern. For example, the recent studies global POC flux rapidly decrease from 100 5.7 Pg C yr⁻¹ at the base of the euphotic zone to 2.8 Pg C yr⁻¹ at the depth horizon of 150 m. It would be helpful if the authors can propagate the variability and error of the euphotic zone definition (i.e., assign 10-20m uncertainty in the euphotic zone definition in the uncertainty estimate).

To facilitate scaling, we use a uniform euphotic zone depth of 74.3 m in the revision, and scale carbon export to 100 m depth that is widely used in earth system models. Therefore, our model results can be easily compared to previous results.

We agree with the reviewer that the integrated carbon export is sensitive to reference depth, however, we decide not to include the variation of reference depth in the error estimates because there is no consensus on which depth should be used. See also our response to point 1.

4) Impact of anthropogenic CO₂ in DIC cycling: Since the DIC dataset spans a long temporal coverage, penetration of anthropogenic CO₂ into the ocean interior will leave an imprint in the deep DIC samples. Omitting these long-term trends will result in an overestimation of DIC accumulation (which might lead to an overestimation of biological respiration). Some analysis or discussion is required.

The reviewer was correct on this point and we are very grateful for bringing this important point to our attention. We addressed in our general comments to all three reviewers.

Other comments (encompassing major comments on the result interpretation):

Line 47: how are the novelty and results of your work compared with the prior effort in using the same approach (inverse model) to estimate global carbon export? It would be helpful to add some relevant discussion somewhere.

We addressed this point in our comments to all three reviewers.

Line 70: need to define the DOC when it came out for the first time in the main text.

Done.

Line 89: it is worthwhile to briefly summarize how the export is computed in your model (based on the respiration at the subsurface, lines 665-674) herein to make the main text more self-explanatory.

Done. We added the following text to the manuscript on lines 136 – 137

“... , which is calculated by integrating POC remineralization below the euphotic zone,...”

Line 140: In the prior paragraph, you mentioned that DOC flux includes suspended POC; however, in this part, the partitioning of DOC flux seems to exclude the suspended POC. I have some trouble understanding the logic herein.

In the revised manuscript, we make it clear that sinking POC and vertically migrating zooplankton are lumped together in the *non-advective-diffusive export flux* and that the export of DOC and suspended POC are lumped together in the *advective-diffusive export flux*. We added the following text to the manuscript on lines 88 – 96.

“Here, using an inverse biogeochemical model for the cycling of phosphorus (P), carbon (C), and oxygen (O) (Fig.1), we estimate the global distribution of the export-flux separated into contributions from advective-diffusive flux of dissolved organic carbon and suspended particles, which encompasses fluxes mediated by physical transports such as the mixed layer pump¹⁶ and subduction pump¹⁷, and DOC contribution to the biological pump^{18,19}, and non-advective-diffusive vertical flux that includes contributions from the gravitational pump¹², zooplankton migration pump²⁰, and seasonal lipid pump²¹.”

Line 124: the similarity between sediment trap observation and your estimate (the latter is supposed to capture more comprehensive carbon export pathways) also might be the fact that exported carbon via the multiple pathways will be further transformed into the aggregates or the large sinking particle via a variety of physical and biological processes during their journey to the depth. Thus, they are also partly captured by traps. See the relevant discussion in Boyd et al., (2019).

Good point. We added the following discussion to our manuscript (see lines 158 – 164)

“For the coastal CARIACO station, the higher fluxes measured in sediment traps have several possible explanations. One, our model may not have adequate resolution. Two, the bias may be due to blooms, which may be poorly represented in our climatological-mean inverse model. Lastly, sediment traps may overestimate particle flux in coastal regions because of augmented ‘statistical funnels’ of particle collection³⁰ or sediment catchment of large aggregates mediated by a spectrum of physical and biological processes¹².”

Line 110: The elevated estimate of POC flux from your model output (almost triple 234Th-POC flux) is also a key factor leading to a higher total organic carbon export than prior estimates. The implication is other particle injection pumps contribute substantially to the bulk POC flux. Given that it is a key result, I would suggest adding more discussion on this point. For example, a more detailed discussion about the different types (i.e., seasonal lipid pump, mesopelagic-migrant pump) of migration pump and their implication in carbon storage. Typically, POC export mediated by zooplankton tends to have a longer carbon sequestration time because of its rapid transfer from the surface to the depth. In addition, I note that your POC flux estimate seems to align with the total POC carbon export rates estimated by Boyd et al., (Nature, 2019; gravitational pump of 7 Pg C yr⁻¹+ all particle injection pump of 4 Pg yr⁻¹). But it looks like you attribute the physically-mediated POC export (due to the eddy subduction or mixed layer change) to the DOC pathway. It might require some clarification.

We thank the reviewer for their suggestion. We define export according to the timescale for the vertical transfer of the organic carbon. Fluxes by fast-sinking POC (gravitational pump) and vertical zooplankton movements (vertical migration pump and seasonal lipid pump), which transport carbon vertically with no appreciable lateral transport, are assigned to non-advective-diffusive export. Fluxes induced by organic carbon detrainment caused by changes of mixed layer depth (mixed layer depth pump) (Dall'Olmo et al., 2016) and by physical subduction (subduction pump) (Omand et al., 2015) are assigned to advective-diffusive export. In the revised manuscript, we consistently use the terms advective-diffusive export and non-advective-diffusive export to avoid any confusion. We present the regional variability of the different pathways, which is only mildly touched upon in Boyd 2019. However, we are unable to discuss the relative importance of different pumps in vertical flux or in mixing flux, because our tracer inversion lumps them together. One benefit of doing so is that we can avoid double counting of various export pathways.

Line 110: Since you rely on the bulk respiration at depth to infer the upper carbon export, it can avoid the double-counting issue (it usually happens if researchers sum up different carbon export pathways from the upper layer or different observing platforms) and also nicely capture the footprint of episodic carbon export (i.e., eddy subduction pump). If so, it might be worthwhile pointing out this merit somewhere.

Good point. We added the following text to the manuscript on lines 134–135.

"... We infer the strength and distribution of the total BCP from tracer distributions, which avoids counting the same export pathways multiple times¹²."

Line 182: In the manuscript, the author conceptually solely attributes the zooplankton-mediated carbon export to the POC portion. I don't think this classification is very accurate. As far as I know, diel and seasonal zooplankton migration also contributes significantly to the vertical DOC export (grazing in the upper layer but excretion of DOC at depth, aka active carbon transport). Therefore, the higher DOC portion in the high latitude might be also related to the enhanced zooplankton migration. It requires some discussion.

It is true that zooplankton excrete DOC at depth, but likely most such DOC are labile and are quickly remineralized. If such DOC release by migrating zoo was significant, we would see it in a vertical DOC concentration profile. However, there is no such signals in our DOC profile. In any event, our new terminology based on advective-diffusive and non-advective-diffusive export should avoid any misconception about the meaning of the export fluxes we are estimating.

Line 178: what does DOC_l stand for? Labile DOC export? Need to define all the abbreviations in the main text.

Yes, DOC_l represents labile DOC. We clarify this in the text (now on line 632).

"..., labile dissolved organic carbon (DOC_l), ..."

Line 246: The fraction of summer exported carbon that does not escape from the deepest winter mixed layer will be re-entrained and return back to the atmosphere. When comparing with the

mesopelagic demand, it would be an opening-eye statement to link with the carbon export estimate at the base of the annual deepest mixed layer.

We agree. In the revised manuscript, we have scaled the carbon flux to the maximum mixed layer depth when we did the comparison to mesopelagic carbon demand. The comparison to previous estimates is therefore updated on lines 297-308.

“For example, at the Porcupine Abyssal Plain (PAP) site in the North Atlantic Ocean, our TOC export ($201.5 \pm 29.4 \text{ mg C m}^{-2} \text{ d}^{-1}$ between 73-1000 m) exceeds the community respiration ($48\text{-}167 \text{ mg C m}^{-2} \text{ d}^{-1}$ between 50-1000 m) determined in the summer season⁴⁴ when the net community production is relatively low⁴⁵. At station ALOHA, our annual TOC flux between mMLD and 1000 m ($45.1 \pm 4.0 \text{ mg C m}^{-2} \text{ d}^{-1}$) has overlapping error bars with in situ determinations of respiration rates by bacteria and zooplankton between 150-1000 m ($32.5\text{-}96.6 \text{ mg C m}^{-2} \text{ d}^{-1}$)⁴⁶. At the Japanese time-series site K2 station also in the Pacific, our TOC flux between mMLD and 1000 m ($82.1 \pm 2.4 \text{ mg C m}^{-2} \text{ d}^{-1}$) falls short of the lower end of in situ determinations ($106.1\text{-}249.8 \text{ mg C m}^{-2} \text{ d}^{-1}$)⁴⁶ at the depth interval of 150-1000 m, suggesting that other mechanisms may play significant role at this location.”

Line 245: it would be better to keep the unit of aNCP consistency throughout the manuscript (I found it swings among $\text{mmol m}^{-2} \text{ yr}^{-1}$ vs $\text{mol m}^{-2} \text{ d}^{-1}$, $\text{mmol m}^{-2} \text{ yr}^{-1}$ vs $\text{mol /m}^{-2} \text{ /yr}^{-1}$).

Thank you for pointing this out. All units are converted to $\text{mg C m}^{-2} \text{ day}^{-1}$.

Line 144: The geographic pattern of DOC to the total organic carbon in your study is opposite to a recent study by Roshan et al., (10.1038/s41467-017-02227-3). It requires some discussion about the potential reason leading to the discrepancy.

There are two improvements in the revision. First our revised inverse model better accounts for the effect of anthropogenic DIC. Second, our revised model includes labile DOC and can match different satellite NPP products without affecting the quality of fit to DIC, DIP, Alk, O₂, and DOC. The spatial pattern of our revised DOC export estimate roughly matches the export pattern by Roshan and DeVries (2017) (See Fig. 3a b and Fig. S7). However, the total exported DOC is different from their values. We have added corresponding discussion as follows (lines 171-178),

“... . Our advective-diffusive flux of semi-labile organic carbon is close to a previous estimate of 1.8 Pg C yr^{-1} at 100 m reference depth¹⁸, but is somewhat lower than the estimated $2.31 \pm 0.6 \text{ Pg C yr}^{-1}$ at the same depth of 73.4 m obtained from interpolated DOC observations and a circulation model¹⁹. When we include the export of labile organic carbon, our estimate surpasses any previous estimates. The previous estimate¹⁹, which considered only one group of DOC, may have included signals from both labile and semi-labile organic carbon, hence explaining why their number falls in between.”

Line 525: should define u and K.

u and K have been defined on the lines 633-635.

“..., and O₂ in the O model) is computed using the OCIM tracer transport matrix, $\mathbf{T}[C] \equiv \nabla \cdot (\vec{U}[C] - \mathbf{K}\nabla[C])$, where \vec{U} is velocity vector and K is diffusive term.”

line 672: Again, it is quite confusing to me how you account for the contribution of suspended POC in the DOC export.

Because export fluxes of suspended POC and DOC are mediated by the same physical processes, they are grouped in what we call the "advective-diffusive export". We believe that this treatment is okay for most of the ocean, because the vertical gradient of suspended POC is much lower (less than a few μM) than that of DOC (dozens of μM).

Figure S1- S2: the text and label in the axis are incomplete or missing.
Corrections have been made.

Fig S3: 2 alongside O should be subscripted.
Corrections have been made.

Figure S11-12: To facilitate the comparison, I would suggest putting the same prediction parameters from three NPP products in the same rows (or the same figure).
The figures have been combined; the new figure is Fig. S7

Line 79: Given that hydrographic data are used to tune the model parameters, the validation presented might be not totally independent. Correct me if I am wrong.

In an effort to show the importance of using multiple tracers to constrain the model, we did a test inversion in which we optimized the model parameters using only DOC and O_2 tracer data. As we show in Fig S5, this leads to some modest improvements to the fit to DOC and O_2 , but produces dramatic deterioration in the quality of fit to DIP, DIC, and Alkalinity. This demonstrates that there is indeed independent information available from all the tracers and that this information can be extracted with careful modeling.

Line 663: After you finished up model optimization, how did you make the global projection and get the annual carbon export? Did you apply the tuned parameters to the gridded monthly NPP product and then obtain the different processes? In addition, there are considerable missing values of remotely sensed NPP, particularly in the high-latitude winter season. I am wondering how authors deal with this issue. Need to be clarified.

Our export estimates are not tightly coupled to the NPP productions. After the optimization, we obtained 3D tracer distributions (DIP, DOP, POP, DOP_i , DIC, DOC, DOC_i , DOC_r , PIC, POC, ALK, and O_2) and optimal parameter values (Table S1). The non-advective-diffusive flux is calculated by integrating particle remineralization below the euphotic zone ($F_{\text{ sinking }} = \sum_{i=1}^N \kappa_p [\text{POC}]_i V_i$). The advective-diffusive is calculated using adjoint method (Primeau et al., 2013), which also depends on tracer distributions and optimal parameters. Our inverse model produces a self-consistent biogeochemical state estimate that includes the tracer concentrations and the associated fluxes and carbon transformations.

We used the pre-processed and interpolated NPP products from the Nowicki et al., (2022) study. The method to deal with missing data is described by Nowicki et al. as follows: "For locations that do not

have NPP data for certain months (due to sea ice coverage or low solar zenith angles), the missing values were filled with either 10% of the maximum monthly NPP during the year, or the lowest monthly NPP, whichever is lower (DW17; Yao & Schlitzer, 2013)." We have cited the literature. Our estimated organic carbon export with residence times of one year or longer are not sensitive to NPP.

Referee #2 (Remarks to the Author):

Review of Wang et al. "A robust biological carbon pump revealed by multi-decadal hydrographic tracer data"

This manuscript presents an alternative value for the magnitude of the biological carbon pump that incorporates both particulate and dissolved organic carbon pathways. The manuscript provides an interesting additional estimate of the BCP to add to the several already published estimates. The value given by this work is substantially higher than previous estimates, even those which (like this work) include gravitational sinking, mixing and advection of suspended material, and vertical transport of zooplankton (e.g. Nowicki et al., 2022, GBC). Like this manuscript, the work of Nowicki et al. (2022) is also based on distributions of observed tracers such as oxygen and DOC but utilises a data-assimilating model ensemble to estimate BCP strength.

We thank the reviewer for his/her comments. We have adequately addressed the difference between our model the previous inverse model in the reply to all reviewers.

Major comments:

The authors need to elaborate why their model should be considered an improvement to, or more robust than, Nowicki's. Otherwise their results, although very worthy of publication in a specialist journal, don't merit publication in Nature.

We addressed this point in our comments to all three reviewers.

The "biogeochemical implications" section is important but rather unconvincing as currently written

and needs strengthening (see specific comments below). In particular, the results don't add any information on the mechanistic understanding of contributions of different pathways.

Our model results have several implications. They help explain the counterintuitive observations that low-export efficiency in high-production oceans; it then closes the budget between carbon export and carbon demand in the mesopelagic oceans; it also emphasizes on the importance to include non-advective-diffusive flux in earth system models. Furthermore, we have added a new paragraph to discuss the biogeochemical implication as follows (see lines 330-347),

“Our global inversion strongly supports the finding, discovered using neutrally buoyant sediment traps, that POC attenuation is temperature dependent⁵². Geographically, non-advective-diffusive vertical fluxes attenuate faster when surface waters are warmer and penetrate deeper in the water column when surface waters are cold (Fig. S8). Notably, our non-advective-diffusive vertical flux includes not only POC flux but also any fluxes with significant non-advective-diffusive vertical transport (e.g., seasonal lipid pump²¹, zooplankton migration pump²⁰). The deeper penetration of non-advective-diffusive vertical fluxes at high latitudes may be due in part to the strong seasonal lipid pump prevalent in those regions¹². Taken at face value, the temperature dependence implies that global warming will cause a stronger non-advective-diffusive vertical-flux attenuation with depth (increased *b*-value in panels c and d of Fig. S8), which will leave more carbon in the upper ocean and atmosphere⁵³. Our findings thus could help explain atmospheric CO₂ variations during glacial-interglacial cycles⁵⁴. The combined vertical flux by sinking particles and vertically migrating zooplankton is more efficient in cold waters than in warm waters (evidenced by lower *b*-value in high latitudes, Fig. S8), which suggests that the removal of CO₂ from the atmosphere would be stronger during cold climates thus act to shift the air-sea CO₂ balance to the ocean.”

Specific comments:

Line 27: specify here that the estimate of 14.26 GtC is the total organic carbon export (i.e. POC+DOC)

Corrected. Thank you. Our updated estimate is 15.00 ± 1.12 Pg C yr⁻¹ at the base of euphotic zone. The text is now on lines 27-29.

“By implicitly accounting for all export pathways, our estimate of total organic-carbon export at the base of the model euphotic zone (73.4 m) is 15.00 ± 1.12 Pg C yr⁻¹, ...”

Line 42: NPP and SST are used to extrapolate the sparse in situ observations to the global scale. This is slightly different from what is implied here (that NPP and SST are used to study regional differences).

Thank you. We have changed the wording as follows (lines 55-57).

“Oceanographers rely on empirical relationships between the ef-ratio and satellite-based measurements of NPP and sea surface temperature (SST) to obtain global-scale export patterns.”

Line 51: another reason for the discrepancies is that the observations are (very often) single points in time and space and so the full annual cycle of, e.g. contributions by migrating zooplankton, cannot be captured.

Thank you for pointing this out. We have added the following discussion to the manuscript (lines 65-71).

“One possible cause for these discrepancies is that most observations provide only snapshots of the ocean at the time of sample collection, whereas episodic signals may be missed in models. Another possible explanation is that the empirical algorithms focus almost entirely on the contribution from sinking particles and often neglect possibly significant contributions from vertically migrating zooplankton and the transport of dissolved and non-sinking particulate organic carbon by subducting and overturning water masses (a.k.a., the particle injection pump¹²).”

Line 92: the use of the term "active export" to refer to POC export is rather confusing here as the term "active flux" has already been associated for many years with the specific portion of the flux attributable to vertical migration of zooplankton and other organisms. I suggest finding another term.

We now use 'advective-diffusive flux' to represent fluxes mediated by physical transports, and 'non-advective-diffusive flux' to represent fluxes mediated by sinking POC, vertical migrator, seasonal lipid pump. The corresponding discussion is on lines 90-96.

“... we estimate the global distribution of the export-flux separated into contributions from advective-diffusive flux of dissolved organic carbon and suspended particles, which encompasses fluxes mediated by physical transports such as the mixed layer pump¹⁶ and subduction pump¹⁷, and DOC contribution to the biological pump^{18,19}, and non-advective-diffusive vertical flux that includes contributions from the gravitational pump¹², zooplankton migration pump²⁰, and seasonal lipid pump²¹.”

Line 92: DOC and slowly sinking POC are different (and slowly sinking POC is different from suspended POC), but here it appears they are all lumped together. As several observational approaches to estimating flux, e.g. sediment traps, Th234 etc., capture both fast and slow sinking POC, when comparing to the observations this way of distinguishing the different components of the flux might introduce a bias.

We agree that sediment traps and Th-234 do capture the flux generated by slowly sinking POC. On occasions, this slowly sinking flux (1-10 m d⁻¹) can make up a substantial amount of the total POC flux out of the surface layers (see Alonso-González et al. (2010) and Riley et al. (2012)). Alonso-González et al. (2010) and Riley et al. (2012) provide a detailed picture of the biogeochemistry of the slow sinking pool. While the slow sinking pool is important right below the surface layer, it is unlikely that such pool resist remineralization for very long and therefore descends through the mesopelagic zone. Riley et al., 2012 found no evidence for ballasting of the slow sinking flux and they concluded that the slow sinking particles were entirely remineralised in the upper mesopelagic zone. Similarly, Alonso Gonzales et al., found that the slow sinking particles are more labile than the

fast-sinking pool. This suggests that the slow sinking pool, when important, is rather restricted to the upper mesopelagic at the upmost. In our study, we compare our outputs to observations in the entire water columns well below the surface layer (base of Ez or mMLD) where the contribution of slow sinking particles is likely very limited. Therefore, we believe that our comparison is not significantly biased by the way we distinguish slow sinking particles relative what observations provide.

Lines 95-98: references needed to support the statements here

A reference has been added (line 132).

Lines 98-100: I didn't understand well the point being made in this sentence.

We have rewritten the paragraph to make it clear (See lines 122-135).

“In our model, which has a horizontal mesh resolution of $2^\circ \times 2^\circ$ and 24 vertical layers, we define export according to the timescale for the vertical transfer of the organic carbon. Fluxes by fast-sinking POC (gravitational pump) and vertical zooplankton movements (vertical migration pump and seasonal lipid pump), which transport carbon vertically with no appreciable lateral transport, are assigned to non-advective-diffusive vertical export. Fluxes induced by organic carbon detrainment caused by changes of mixed layer depth (mixed layer depth pump)¹⁶ and physical subduction (subduction pump)¹⁷ are assigned to advective-diffusive export. It is important to note that while our model’s DOC pool includes what would be characterized as suspended POC in field measurements and therefore is missing from the DOC measurement database, we believe the difference is negligible for most of the ocean. This is because the concentration of suspended POC is much lower (less than a few μM) than that of DOC (dozens of μM)¹⁸. We infer the strength and distribution of the total BCP from tracer distributions, which avoids counting the same export pathways multiple times¹².”

Line 122-123: if the data being shown is obtained from traps, then the station POC flux estimates do in fact include fecal pellets, and therefore capture some portion of the active flux.

Good point. We added relevant discussion as follows (see lines 149-154).

“The non-advective-diffusive vertical flux is consistent with measurements from deep-water sediment traps²⁷ at ocean stations ALOHA, OSP, BATS, and CARIACO, where extensive measurements exist (Fig. 2), even though such POC-flux measurements only partially include the contribution from vertical zooplankton migration (fecal pellets).”

Line 130: not clear to me what is meant by “oversampling” here

We changed “oversampling” to “overestimation” to avoid misunderstanding. The following updated text is on lines 161-164.

“Lastly, sediment traps may overestimate particle flux in coastal regions because of augmented ‘statistical funnels’ of particle collection³⁰ or catchment of large aggregates mediated by a spectrum of physical and biological processes¹². ”

Line 180-182: until this point, the concepts of the mixed layer pump, subduction pump etc. haven't been introduced. Worth doing so earlier on in the introduction. Also, some explanation needs to be added on why the increase of DOC export is consistent with the mechanisms of these pumps.

Thank you for this suggestion. We introduced the two types of pumps in the Introduction (see lines 92-94), and briefly explained their mechanisms on lines 125-127.

Lines 92-94:

“..., which encompasses fluxes mediated by physical transports such as the mixed layer pump¹⁶ and subduction pump¹⁷, and DOC contribution to the biological pump^{18,19}, ...”

Lines 127-129

“Fluxes induced by organic carbon detrainment caused by changes of mixed layer depth (mixed layer depth pump)¹⁶ and by physical subduction (subduction pump)¹⁷ are assigned to advective-diffusive export.”

Lines 194-195: specify that, of the 2 numbers reported, the model result is given first followed by the observations.

Thank you for your suggestion. We have made the following changes on lines 257-265.

“We further compared our TOC flux rate with those measured using mass balance calculations at ALOHA, BATS, and OSP. Our model results (mean with $\pm\sigma$) at the base of maximum mixed layer depth (mMLD) have overlapping errorbars with mass balance estimates at ALOHA (45.99 ± 23.00 this study vs. 82.15 ± 23.00 mg C m⁻² d⁻¹) and at OSP (52.65 ± 3.29 this study vs. 75.56 ± 19.71 mg C m⁻² d⁻¹)¹³. Our estimate at the BATS station (23.00 ± 3.28 mg C m⁻² d⁻¹ at mMLD) is significantly lower than ANCP by Emerson¹³ (124.83 ± 39.42 mg C m⁻² d⁻¹ at 150 m), but two-fold higher than the ANCP determined using O₂ and DI¹³C in the Western North Atlantic around the BATS station at 100 m depth (82.13 ± 13.14 this study vs. 39.42 mg C m⁻² d⁻¹ Ref.³⁶).”

Line 199: here the model result is compared only to the Laws algorithm – what about other satellite-based estimates of export?

We have removed the comparison to Laws satellite algorithm because it is considered to be less accurate than the geochemical tracer estimates (Yang et al., 2019).

Lines 199-200: provide the numbers for ANCP at BATS
Numbers have been added.

Line 202: Figure 4d doesn't exist

We meant to say Fig. 4b, c. Corrections have been made.

Lines 204-205: I think I understand what is meant here, but it's written badly

We have changed the wording of this sentence on lines 267-268.

“Our results shed light on the low export efficiency observed, counterintuitively, in high-productivity regions (e.g., the Southern Ocean).”

Lines 206-207: the negative correlation refers to the export ratio vs PP (not export flux vs PP), and furthermore has only been demonstrated for the Southern Ocean

Thank you for pointing this out. We have made the following changes on lines 268-271.

“In-situ determined export ratios are negatively correlated with NPP⁸, which contradicts the empirical relationships that relate the ef-ratio to temperature and NPP^{3,37} by assuming a positive relationship between NPP and the ef-ratio.”

Lines 212-214: this is a lag issue again though, no? see for example Henson et al. (2015, GBC) and Giering et al. (2017, GBC)

Giering et al. (2017) and Henson et al. (2015) present work on the time lag in the mesopelagic (between POC flux at about 100m and POC flux at > 500m). This is done in the context of interpreting flux attenuation estimates or transfer efficiency estimates. In lines 212-214 of the previous manuscript however, we were referring to surface processes (namely surface bacterial recycling and zooplankton grazing) in the context of explaining variability in export efficiency. Regardless of the time lag induced by aggregation processes (See Stange et al. (2017) for instance), both bacterial recycling and zooplankton grazing do reduce the amount of the primary production available for aggregation (and therefore, for subsequent export). This implies that both processes could well be possibilities to explain the occurrence of low particle export in high primary production regions. The context is therefore slightly different from what is stated in Henson et al. (2015) and Giering et al. (2017).

We have modified the sentence on lines 275-277, and also as follows ,

“...; (H3) Grazing-mediated export varies inversely with PP in surface ocean (low grazing in high PP regions reduces the amount of zooplankton mediated export)^{11,38}; ...”

Line 214: high export, low productivity conditions have been found episodically too (Henson et al., 2019, GBC)

Henson et al. (2019) report low productivity/high export efficiency regions, which is different from low productivity/high export. The export cannot be higher than the PP. Low productivity/high export efficiency regions export efficiently but not so much given that they are low PP to start with. In any case, hypothesis H1, H2, H3 and H4 would be valid in such low productivity/high export. For the case of H3, high grazing in low PP regions may enhance the amount of zooplankton mediated export and leads to high export efficiency (see Le Moigne et al. (2016)).

Lines 218-235: this is a not very insightful or useful paragraph – basically the conclusion is that there are no clues from this study about why certain patterns of export flux are found in observations. We have removed this paragraph. Instead, we have added more discussions about model implications.

Lines 224-225: I didn't understand the reasoning why this line of evidence shows that "export of DOC may be an essential process"

We followed your previous suggestion, and removed this paragraph.

Lines 233-234: which are the "warm high PP regions"? upwelling zones?

We followed your previous suggestion, and removed this paragraph.

Lines 234-235: this final sentence emphasises that the results shown here shed no mechanistic light on the causes of low export in high PP zones – so why have a whole paragraph speculating about a debate that your results make no contribution to?

We followed your previous suggestion, and removed this paragraph.

Line 254: The models of Nowicki et al. and Devries and Weber (2017, GBC) do account for DOC, and the former also includes vertical migration.

The sentence has been modified as follows on line 309-311,

"Only a few global carbon export models^{22,23} and algorithms account for the advective-diffusive export of DOC and suspended POC^{4,37,47} or the export mediated by zooplankton migration²⁶."

Line 258: "we find that DOC flux..." or indeed slow sinking or suspended POC as these 3 elements are separated in the model (at least as I understood it from lines 92-93)

Correct. We are now using the term "advective-diffusive flux" to refer to the flux of dissolved organic carbon (DOC) and the flux of suspended particulate organic carbon (POC). This change in terminology aims to minimize any potential confusion in our discussion.

Lines 264-265: indeed, an improved mechanistic understanding is needed – but this study does not provide that as emphasised in lines 276-277. This is important, as without including mechanistic representation of processes that are considered key to export flux, the response to future climate change cannot be well-predicted.

The problem is that mechanistic models are uncertain as can be seen by the spread in the biological carbon pump simulated in the most recent CMIP6 models which have an 8 GtC/yr spread (See Fig 1b from Henson et al., 2022 and Fig. A reproduced here.)

Fig. A: Absolute change in globally averaged export flux in 19 coupled climate models forced with the SSP5–8.5 scenario⁷² taken from the Coupled Model Inter-comparison Project phase 6 (CMIP6) archive (Reproduced from Henson et al., 2022)

Incorporating mechanistic description of the export pathways into the model inevitably introduces more model uncertainty because we do not have sufficient mechanistic understanding to unambiguously specify them parametrically. Hence, our inverse model provides a top-down estimate based on the integrated effects of organic matter export on well observed tracers, particularly those of carbon (DIC and DOC) and oxygen O_2 , but also Alk and DIP. Our assumption is that, irrespective of the pathway that export follows, the organic matter exported will utilize oxygen and eventually decompose to inorganic carbon. Consequently, in order to achieve the observed tracer distributions, a certain quantity of organic matter must have been exported from the euphotic zone.

Our model can detect strong temperature dependence on the value of the exponent, b value, in the Martin curve. Geographically, non-advective-diffusive fluxes attenuate faster when surface waters are warmer and penetrate deeper in the water column when surface waters are cold (Fig. S8). The result has implications on how the ocean will respond to the future climate change as we answer your major comment #2.

Figure 1: why have a power-law dependence for the POC flux attenuation, but an exponential dependence for the PIC flux attenuation??

The exponential function is a commonly-used model to describe PIC flux profile (see ‘Ocean Biogeochemical Dynamics by Sarmiento and Gruber’ on p.374). This is because compared to a power-law function, an exponential function with a length scale on the order of ~4500 m leads to a much smaller CaCO₃ dissolution rate in the shallow water where CaCO₃ is supersaturated.

Since the caption of figure 1 is already crowded, we have added the following text on lines 699-701.

“Compare to a power-law function, an exponential function with a length scale on the order of several thousand meters leads to a much smaller CaCO₃ dissolution rate in the shallow water where CaCO₃ is supersaturated¹³.”

Figure 2: the colour scale on plot a) is unhelpful for distinguishing regional variability. The data should be plotted in gC m⁻² yr⁻¹ to make the results directly comparable to plots in previous work such as Henson et al. (2012, GBC) or Siegel et al. (2017, GBC). Are the data shown in panels b-e) annual estimates, or point measurements from the sites? As the model only produces a climatological annual estimate I assume it's the former? What time period are the data averaged over (i.e. it's multiple years, as a guess)? For panel e), change the y-axis so the vertical differences can be seen (include a broken y-axis so the outlier at 137 m depth can still be seen).

We have changed the color scale, but have maintained the mono-color type of colormap in accordance of the requirement by Nature journals. To ensure consistency, all units have been converted to mg C m⁻² d⁻¹.

In panels b)-e), the sediment trap data presented are multi-year collections covering a sampling period of 1988-2011 for the BATS station, 1988-2010 for the ALOHA station, 1987-2006 for the Ocean Station Papa, and 1995-2012 for the CARIACO station. Since sediment traps are deployed in the water for several months, their measurements represent an average for a relatively extended period instead of snapshot. The box plots in the figures display all measurements determined in each station. We have made this clear in the caption.

We have included a break in the y-axis for panel e.

Figure 4: panel a) has another unhelpful colourscale. Again, plot in gC m⁻² yr⁻¹ to be comparable with previous studies. Figure 4 caption – I'm not clear what reference depth is used here. Is it the deeper of the euphotic zone depth and the mixed layer depth?

We have updated the color scale for panel a. To ensure consistency, all units have been converted to mg C m⁻² d⁻¹.

The reference depth is the model euphotic zone depth (~73 m).

Equation 1: shouldn't the conditional be "if $z > z_c$ "?

No, z is water depth, and z_c is the euphotic zone depth. $z < z_c$ defines the euphotic zone.

Line 532: what sinking speed is chosen? Are the results sensitive to this choice?

The flux divergence operator is built following Kriest and Oschlies (2008). We do not arbitrarily assign a constant sinking speed because sinking speeds of different particles span a large range. Instead, we define a remineralization curve for bulk POC, and sinking speed is implicitly determined by POC dissolution rate (κ_p) and the exponent b value that is optimized in the inversion. We have added corresponding reference to the manuscript (lines 641-644).

"... We choose a sinking speed that increases linearly with depth and a constant dissolution rate, $\kappa_p = (1/30) \text{ days}^{-1}$, so that the attenuation of the vertical flux of POP follows a powerlaw function, $F(z) = F(z_0)(z/z_0)^{-b}$, where $F(z)$ and $F(z_0)$ are fluxes at a depth of z and z_0 , respectively¹⁰...."

Lines 535-536: I'm very surprised that doubling the remineralisation rate doesn't significantly change the results! Surely this should be a first order control?

In our model, the remineralization curve of POC is jointly determined by remineralization rate (κ_p) and exponent b value. κ_p is prescribed and exponent b is optimized in the inversion. If we increase POC remineralization rate by doubling the value of κ_p , the optimization routine will adjust the b value, so that the POC remineralization curve remains relatively stable, and best matches the observed tracer data.

Lines 536-539: there is a big assumption here which isn't explored in the paper – that b is a function of temperature

The temperature dependence of b is not an assumption. Instead, it was determined based on in-situ data from sediment traps and surface temperature (Marsay et al. 2015). In this study, we re-optimized the linear relationship (slope and intercept) in the inversion.

Section 4 of Methods: again, it wasn't clear from the text here how the export depth horizon is defined

We thank you for your suggestion. To make it clear, we have adopted a uniform euphotic zone depth, top two model layers (73.4 m). We report two exports at both the model euphotic zone and at 100 m depth that is widely used in ESMs. To normalize the export from the model euphotic zone to 100 m, we use Martin curve function with optimal exponent b values. Meanwhile, the estimate of export flux is highly sensitive to the choice of depth horizon. We therefore provide an alternative perspective using the sequestration-time-partitioned organic matter production (Fig. 5).

Table S1: an additional column explaining in words what the parameters are would be extremely helpful and stop the reader from having to flick between the equations in the methods section and the supplementary material. There are some huge (1-2 orders of magnitude) differences in the estimated parameter values depending on which NPP model is used, e.g. for bp-sigma, beta, rsi and dd. This implies that the model is highly sensitive to the input fields and that multiple different non-exclusive solutions to the optimization are possible. This doesn't give much faith in the robustness of the model results. The implications of this should be discussed in the manuscript.

We thank the reviewer for pointing this out. We have added explanations for each parameter in the table or footnotes.

In previous version, we averaged the downloaded satellite NPP data to get an annual climatology. The missing data in high latitude oceans were interpolated from spatial neighbors. According to the suggestions of reviewer #1, we have adopted the processed NPP products by Nowicki et al., and have removed MODIS VGPM and used CbPM and CAFE products. This change has led to greater consistency in the model parameters between the two models, as demonstrated in Table S1.

Figures S1 and S2: "fraction of the grid boxes" – does this mean vertically?

Correct. The sampling density are for each water column grid box. We added the following text to the caption.

"The color denotes the fraction of the grid boxes in each water column with at least one measurement. For each vertical column, the sampling density is defined as the number of grid boxes with at least one sample divided by the total number of wet grid boxes."

Figures S3-S5: all basically indistinguishable – are they all needed?

We have removed the original Fig. S4 and Fig. S5.

Figures S6-S8: again, all basically indistinguishable

We have removed the corresponding Figures.

Figure S9: is this figure necessary?

This figure has been removed.

Figures S10-S11: again, basically indistinguishable

The figures have been removed.

Figure S12: where are the plots with $k_p = (1/30) \text{ day}^{-1}$ –to compare the results of this sensitivity test to?

The plots with $k_p = (1/30) \text{ day}^{-1}$ are the results reported in the main text. The pattern of $k_p = (1/60) \text{ day}^{-1}$ is indistinguishable from the main results, based on the previous comments of Reviewer #2, we decided not to include the figure.

And finally, most of the supplementary figures aren't referenced in the text.

We have double checked the text and made sure that all figures are correctly cited.

Referee #3 (Remarks to the Author):

This is a thorough, clearly written, and potentially highly impactful study that quantifies the strength of the ocean biological carbon pump, something that is highly uncertain yet key for our understanding of the global carbon cycle. The most noteworthy finding in the study is the estimate of global export production at $\sim 14 \text{ PgC/yr}$ – substantially higher than estimates from satellites and CMIP models. Of this 14 PgC/yr , the study suggests that $\sim 12 \text{ PgC/yr}$ is particulate/gravity export, while $\sim 2 \text{ PgC/yr}$ is dissolved/circulation export. These quantities are determined from a sophisticated inverse modeling

approach and constrained by hydrographic observations of biogeochemical quantities.

We thank the reviewer for the positive comments.

The modeling approach is sound and the authors have taken great care to explore the sensitivity of parameter values to various assumptions. For the modeling approach, I have a few minor comments to consider:

-Please address the uncertainties in the OCIM transport matrix, and the influence of these uncertainties on the parameter values.

The OCIM tracer transport operator eliminates most of the biases present in free running OGCMs by parameterizing the effects of unresolved eddies by adding eddy stresses to the horizontal momentum and optimizing these extra terms using a variational data assimilation. Unfortunately, this is a large-scale parameter optimization problem and it is difficult to sample the posterior probability distribution for the inferred eddy stresses. As a result, we do not have an accurate estimate of the circulation uncertainty. There have been sensitivity analyses in which multiple OCIM operators were created by prescribing different values for the background isopycnal and diapycnal diffusivities (eg. DeVries and Holzer 2019). However, these additional transport operators are not proper samples from the posterior probability distribution and are in fact quite improbable based on their extremely low likelihood values. When we tested them with our biogeochemical model optimization, we obtained very poor fits to the oxygen and carbon data. As a result, we are not able to quantify the uncertainty due to the OCIM transport operator.

-Please quantify the sensitivity of the carbon model parameters to the air-sea CO₂ flux parameterization.

We have recalibrated the air-sea CO₂ flux parameterization using ¹⁴C observations. The corresponding method description is shown below and on the lines 736-754.

“To calibrate the air-sea gas exchange parameterization, we re-optimized the scaling factor in the OCIMP2 gas-exchange scheme by minimizing the misfit between our modeled $\Delta^{14}\text{C}$ and the GLODAPv2 $\Delta^{14}\text{C}$ data. See Fig. S9a and S9b for the number of observations as a function of time. To compute the misfit we sampled our model at the location and times of the bottle measurements in the GLODAPv2 database. Our calibration method followed an iterative two-step process in which we first optimized the air-sea gas exchange through a series of transient carbon-cycle simulations. After obtaining the optimal air-sea gas exchange, we subtracted the excess anthropogenic DIC from the GLODAPv2 measurements to produce an estimate of the natural background DIC for the year 1850. The resulting DIC data and optimal gas-exchange velocity were then used for the optimization of the biogeochemistry model (see Section 3). The optimized biogeochemical model was then used to produce an updated initial condition for the transient carbon-cycle simulation and a re-optimization of the air-sea gas-exchange velocity. We repeated this two-step process until we obtained self-consistent estimates of (i) the optimal biogeochemical parameter values (see Table S1), (ii) the biogeochemical state, (iii) the scaling factor for the air-sea gas transfer velocity, $a = 0.234 \text{ cm h}^{-1} / (\text{m s}^{-1})^2$, (iv) transient DIC, and (v) the transient $\Delta^{14}\text{C}$ signal including the combined effects of radioactive decay, the Suesse effect, and the bomb radiocarbon signal. Figure 9(c) shows a time-series of the excess anthropogenic DIC concentration averaged over the top 100 m of the water column and for the water column below 100 m. By 2020 the vertical DIC gradient is reduced by 20%.”

-Please address the significant deviations from the 1:1 line on the observed vs. modeled oxygen density distribution plots (Figs S3-S5) at the high oxygen values (225-275 mmol/m³) – these appear regardless of the NPP algorithm selected.

The deviations disappeared in our new simulations after dealing with anthropogenic DIC correctly. Please see our reply to all three reviewers for how we dealt with the anthropogenic DIC.

I have two other major points for the authors to consider:

1) While reading, I found myself actively comparing the approach and findings to that of the recent Nowicki et al. paper (<https://agupubs.onlinelibrary.wiley.com/doi/10.1029/2021GB007083>). The export production estimated in this study (14 PgC/yr) is substantially larger than that of the Nowicki study (10 PgC/yr), yet the approaches are very similar (inverse modeling, constrained by hydrographic observations). This leaves me wondering which approach produces the correct answer. The onus is on this study's authors to demonstrate that their approach is superior, or at least that it is different enough to warrant their much higher estimates. As it stands, the manuscript has only a single sentence that addresses the discrepancy in the findings, and additional discussion is

needed.

There are probably multiple ways to demonstrate this point for the reader -- I offer one suggestion here, but it would require substantial effort: Use one of the CMIP models as a testbed for your approach. First, generate a transport matrix from the CMIP model's CFC, T, S, etc. Then, use the CMIP model's NPP in your top down inverse model, and the model's subsampled biogeochemical tracers to constrain the parameters in the inverse model. Finally, quantify how well the inverse estimate of, say, globally integrated POC export, matches that of the original CMIP model. If you wanted to go all-out in this effort, you could also apply the Nowicki model approach to the subsampled CMIP model and then compare the results.

Both our model and the model of Nowicki et al. depend on an offline advective-diffusive transport operator that is optimized using tracers. To derive such an offline transport operator from an earth system model involves tremendous amount of effort and is beyond the scope of the current study. However, we believe that our model provides a more robust estimate of the biological carbon pump because it is constrained by more tracers than the model produced by Nowicki et al. study. See our general comments to all the reviewers.

2) One of the hallmarks of a strong paper is when the reader can think of ways to build off the new findings or approach described in the paper. Effective papers are capable of influencing scientific research not only in their subfield or discipline, but also more broadly. Yet, as written, the paper's broader impacts are rather narrow: explaining the relationship between NPP and POC export, the relative importance of POC and DOC in export, and regional distribution of DOC export. I encourage the authors to think even bigger in their impacts, and to address how their findings will influence, e.g., the global carbon cycle, changes in the ocean biological carbon pump and thus atmospheric CO₂ on glacial-interglacial timescales, etc.. I would like to see some of these described in a final or near-final paragraph in the paper.

This is a good point. We have added the following discussion about our model implication as follows and also see our manuscript on lines 330-367.

“Our global inversion strongly supports the finding, discovered using neutrally buoyant sediment traps, that POC attenuation is temperature dependent⁵². Geographically, non-advective-diffusive vertical fluxes attenuate faster when surface waters are warmer and penetrate deeper in the water column when surface waters are cold (Fig. S8). Notably, our non-advective-diffusive vertical flux includes not only POC flux but also any fluxes with significant non-advective-diffusive vertical transport (e.g., seasonal lipid pump²¹, zooplankton migration pump²⁰). The deeper penetration of non-advective-diffusive vertical fluxes at high latitudes may be due in part to the strong seasonal lipid pump prevalent in those regions¹². Taken at face value, the temperature dependence implies that global warming will cause a stronger non-advective-diffusive vertical-flux attenuation with depth (increased b -value in panels c and d of Fig. S8), which will leave more carbon in the upper ocean and atmosphere⁵³. Our findings thus could help explain atmospheric CO₂ variations during glacial-interglacial cycles⁵⁴. The combined vertical flux by sinking particles and vertically migrating zooplankton is more efficient in cold waters than in warm waters (evidenced by lower b -value in high latitudes, Fig. S8), which suggests that the removal of CO₂ from the atmosphere would be stronger during cold climates thus act to shift the air-sea CO₂ balance to the ocean.

Closing Remarks

One strength of our inverse model is that the estimated export fluxes are not sensitive to satellite-estimated NPP. This is a major difference with export estimates based on the ef -ratio, which suffer from the compound uncertainties in the ef -ratio and in the algorithm used to estimate NPP²⁸. In contrast, our inverse model infers carbon export from the respiration signal imprinted in the full water column DIC, DOC, DIP, alkalinity, and oxygen observations. Unlike prognostic ESMs, our top-down estimate avoids the need for incorporating uncertain and possibly incorrect parameterizations of complex processes for which we have insufficient understanding. However, our model has its own limitations. For example, our advection-diffusion transport model represents the climatological annual-mean circulation and lacks seasonality. Therefore, we are unable to diagnose how export changes seasonally. Future developments of our inverse model should consider the effect of seasonal variation. Finally, the successful integration of DIC and oxygen measurements in our model was contingent upon an accurate estimation of the transient anthropogenic carbon signal. Our estimate shows that vertical DIC gradient in the ocean has decreased by approximately 20% due to the invasion of anthropogenic CO₂ (Fig. S9c and Methods Section 2.3). We therefore expect that future improvements in anthropogenic carbon uptake estimates will need to take into account the multi-tracer constraints we used here.”

Because this is a potential nature paper and thus the authors will have their manuscript professionally edited, I did not focus on in-line text corrections, though I noted that the transport matrix (T) should be bold-faced in all of the model equations, so as to not confuse it with temperature.

We have carefully gone through the text to make sure there are no typos. Thank you.

Reference:

- Alonso-González, I. J., Arístegui, J., Lee, C., Sanchez-Vidal, A., Calafat, A., Fabrés, J., Sangrá, P., Masqué, P., Hernández-Guerra, A., & Benítez-Barrios, V. (2010). Role of slowly settling particles in the ocean carbon cycle. *Geophysical Research Letters*, *37*(13), n/a-n/a. <https://doi.org/10.1029/2010gl043827>
- Bianchi, D., Weber, T. S., Kiko, R., & Deutsch, C. (2018). Global niche of marine anaerobic metabolisms expanded by particle microenvironments. *Nature Geoscience*, *11*(4), 263-268. <https://doi.org/10.1038/s41561-018-0081-0>
- Boyd, P. W., Claustre, H., Levy, M., Siegel, D. A., & Weber, T. (2019). Multi-faceted particle pumps drive carbon sequestration in the ocean. *Nature*, *568*(7752), 327-335. <https://doi.org/10.1038/s41586-019-1098-2>
- Cochran, J. K., Miquel, J. C., Armstrong, R., Fowler, S. W., Masqué, P., Gasser, B., Hirschberg, D., Szlosek, J., Rodriguez y Baena, A. M., Verdeny, E., & Stewart, G. M. (2009). Time-series measurements of ²³⁴Th in water column and sediment trap samples from the northwestern Mediterranean Sea. *Deep Sea Research Part II: Topical Studies in Oceanography*, *56*(18), 1487-1501. <https://doi.org/10.1016/j.dsr2.2008.12.034>
- Dall'Olmo, G., Dingle, J., Polimene, L., Brewin, R. J. W., & Claustre, H. (2016). Substantial energy input to the mesopelagic ecosystem from the seasonal mixed-layer pump. *Nat. Geosci.*, *9*, 820-825. <https://doi.org/10.1038/ngeo2818>
- DeVries, T. & Holzer, M. Radiocarbon and Helium Isotope Constraints on Deep Ocean Ventilation and Mantle - ³He Sources. *Journal of Geophysical Research: Oceans* **124**, 3036-3057 (2019). <https://doi.org/10.1029/2018jc014716>
- DeVries, T., & Weber, T. (2017). The export and fate of organic matter in the ocean: New constraints from combining satellite and oceanographic tracer observations. *Global Biogeochemical Cycles*, *31*(3), 535-555. <https://doi.org/10.1002/2016gb005551>
- Emerson, S. (2014). Annual net community production and the biological carbon flux in the ocean. *Global Biogeochem. Cycles*, *28*(1), 14-28. <https://doi.org/10.1002/2013gb004680>
- Giering, S. L., Sanders, R., Martin, A. P., Henson, S. A., Riley, J. S., Marsay, C. M., & Johns, D. G. (2017). Particle flux in the oceans: Challenging the steady state assumption. *Global Biogeochemical Cycles*, *31*(1), 159-171.
- Hansell, D. A., Carlson, C. A., Amon, R. M. W., Álvarez-Salgado, X. A., Yamashita, Y., Romera-Castillo, C., & Bif, M. B. (2021). Compilation of dissolved organic matter (DOM) data obtained from the global ocean surveys from 1994 to 2020 (NCEI Accession 0227166) [DOC]. *NOAA National Centers for Environmental Information*. <https://doi.org/10.25921/s4f4-ye35>
- Hansell, D. A., Carlson, C. A., Repeta, D. J., & Schlitzer, R. (2009). Dissolved Organic Matter in the Ocean a Controversy Stimulates New Insights. *Oceanography*, *22*(4), 202-211. <https://doi.org/10.5670/oceanog.2009.109>
- Henson, S., Le Moigne, F., & Giering, S. (2019). Drivers of carbon export efficiency in the global ocean. *Global Biogeochemical Cycles*, *33*(7), 891-903.
- Henson, S. A. *et al.* Uncertain response of ocean biological carbon export in a changing world. *Nature Geoscience* **15**, 248-254 (2022). <https://doi.org/10.1038/s41561-022-00927-0>
- Henson, S. A., Yool, A., & Sanders, R. (2015). Variability in efficiency of particulate organic carbon export: A model study. *Global Biogeochemical Cycles*, *29*(1), 33-45.

- Kriest, I. & Oschlies, A. On the treatment of particulate organic matter sinking in large-scale models of marine biogeochemical cycles. *Biogeosciences* **5**, 55-72 (2008).
- Kwon, E. Y., Primeau, F., & Sarmiento, J. L. (2009). The impact of remineralization depth on the air–sea carbon balance. *Nat. Geosci.*, *2*(9), 630-635.
<https://doi.org/10.1038/ngeo612>
- Le Moigne, F. A. C., Henson, S. A., Cavan, E., Georges, C., Pabortsava, K., Achterberg, E. P., Ceballos-Romero, E., Zubkov, M., & Sanders, R. J. (2016). What causes the inverse relationship between primary production and export efficiency in the Southern Ocean? *Geophys. Res. Lett.*, *43*(9), 4457-4466.
<https://doi.org/10.1002/2016gl068480>
- Letscher, R. T., & Moore, J. K. (2015). Preferential remineralization of dissolved organic phosphorus and non-Redfield DOM dynamics in the global ocean: Impacts on marine productivity, nitrogen fixation, and carbon export. *Global Biogeochem. Cycles*, *29*(3), 325-340. <https://doi.org/10.1002/2014gb004904>
- Marsay, C. M., Sanders, R. J., Henson, S. A., Pabortsava, K., Achterberg, E. P., & Lampitt, R. S. (2015). Attenuation of sinking particulate organic carbon flux through the meso-oceanic ocean. *Proc Natl Acad Sci U S A*, *112*(4), 1089-1094.
<https://doi.org/10.1073/pnas.1415311112>
- Martinez-Garcia, A., Sigman, D. M., Ren, H., Anderson, R. F., Straub, M., Hodell, D. A., Jaccard, S. L., Eglinton, T. I., & Haug, G. H. (2014). Iron fertilization of the Suboceanic ocean during the last ice age. *Science*, *343*(6177), 1347-1350.
<https://doi.org/10.1126/science.1246848>
- Mouw, C. B., Barnett, A., McKinley, G. A., Gloege, L., & Pilcher, D. (2016). Global ocean particulate organic carbon flux merged with satellite parameters. *Earth System Science Data*, *8*(2), 531-541. <https://doi.org/10.5194/essd-8-531-2016>
- Nowicki, M., DeVries, T., & Siegel, D. A. (2022). Quantifying the Carbon Export and Sequestration Pathways of the Ocean's Biological Carbon Pump. *Global Biogeochemical Cycles*, *36*(3), e2021GB007083.
<https://doi.org/https://doi.org/10.1029/2021GB007083>
- Omand, M. M., D'Asaro, E. A., Lee, C. M., Perry, M. J., Briggs, N., Cetinic, I., & Mahadevan, A. (2015). Eddy-driven subduction exports particulate organic carbon from the spring bloom. *Science*, *348*(6231), 222-225. <https://doi.org/10.1126/science.1260062>
- Primeau, F. W., Holzer, M., & DeVries, T. (2013). Southern Ocean nutrient trapping and the efficiency of the biological pump. *Journal of Geophysical Research: Oceans*, *118*(5), 2547-2564. <https://doi.org/10.1002/jgrc.20181>
- Riley, J. S., Sanders, R., Marsay, C., Le Moigne, F. A. C., Achterberg, E. P., & Poulton, A. J. (2012). The relative contribution of fast and slow sinking particles to ocean carbon export. *Global Biogeochemical Cycles*, *26*(1), n/a-n/a.
<https://doi.org/10.1029/2011gb004085>
- Roshan, S., & DeVries, T. (2017). Efficient dissolved organic carbon production and export in the oligotrophic ocean. *Nature Communications*, *8*(1), 2036.
<https://doi.org/10.1038/s41467-017-02227-3>
- Sarmiento, J. L. & Gruber, N. *Ocean biogeochemical dynamics*. P.374 (Princeton university press, 2006).
- Siegel, D., & Deuser, W. (1997). Trajectories of sinking particles in the Sargasso Sea: modeling of statistical funnels above deep-ocean sediment traps. *Deep Sea Research Part I: Oceanographic Research Papers*, *44*(9-10), 1519-1541.

- Siegel, D. A., Granata, T. C., Michaels, A. F., & Dickey, T. D. (1990). Mesoscale eddy diffusion, particle sinking, and the interpretation of sediment trap data. *Journal of Geophysical Research: Oceans*, *95*(C4), 5305-5311.
- Stange, P., Bach, L. T., Le Moigne, F. A., Taucher, J., Boxhammer, T., & Riebesell, U. (2017). Quantifying the time lag between organic matter production and export in the surface ocean: Implications for estimates of export efficiency. *Geophysical Research Letters*, *44*(1), 268-276.
- Yang, B., Emerson, S. R., & Quay, P. D. (2019). The Subtropical Ocean's Biological Carbon Pump Determined From O₂ and DIC/DI¹³C Tracers. *Geophysical Research Letters*, *46*(10), 5361-5368. <https://doi.org/10.1029/2018gl081239>

Reviewer Reports on the First Revision:

Referees' comments:

Referee #1 (Remarks to the Author):

I appreciated the authors' earnest response and their dedicated effort in addressing a school of potential concerns I raised regarding the model's accuracy and interpretations of its results. Particularly, I am delighted to see that my key suggestion about the inclusion of accounting for the impact of anthropogenic carbon invasion in the DIC dataset has led to a substantial improvement in the model's predictive skill and projections. Also, I am very impressed by the addition of Figure 5 in the revised version's main text, depicting the sequestration-time distribution of multiple carbon export pathways. This figure offers readers a captivating perspective on how these pathways are interconnected with integration depth and their implications for long-term carbon storage.

Overall, I am fully satisfied with the authors' response and would recommend publishing this work in Nature. I believe that this exceptional study will generate significant interest/impact within the scientific community, particularly these in the fields of marine biogeochemistry, paleoclimatology, climate change, Earth system modeling, and marine carbon dioxide removal.

Referee #3 (Remarks to the Author):

The authors have addressed all of my previous concerns, and the manuscript is now in an acceptable form for publication.

Referee #4 (Remarks to the Author):

The exact same text below is also uploaded as a pdf ("naturereview.pdf") for ease of reading.

(I am a reviewer who is checking that Reviewer 2's suggestions were incorporated.)

This paper is well-written and adds another important new estimate of total export. With the improvements made at the suggestion of the first round of reviewers to account for anthropogenic DIC, the methodology is much more robust now. The added elaboration on why the model should be considered an improvement to Nowicki's is also well explained after revision. The use of both CbPM and CAFE, and move away from Laws, for NPP estimates further improved the paper as well.

There are 2 primary concerns raised by Reviewer 2 that I don't believe have been fully addressed.

The first primary concern not addressed is regarding the biogeochemical implications, which I will walk through by referring to the author's revised text under the section "Biogeochemical

implications.”

Line 267-268: “Our results shed light on the low export efficiency observed, counterintuitively, in high-productivity regions (e.g., the Southern Ocean).” Though the results do indeed calculate export in the Southern Ocean and other high-productivity regions, I don’t see how they are able to speak to the reasons for low export efficiency in the Southern Ocean.

Line 279-280: “They could explain why high ef-ratios are rarely observed in highly productive regions.” Are high ef-ratios really rare in productive regions? Or do you mean just in the Southern Ocean? If you mean just the Southern Ocean, please say so. Otherwise please cite.

Line 282-283: “Our high-to-intermediate advective-diffusive organic carbon flux in the Southern Ocean indicates that H1 may also explain the low POC export efficiency.” What is meant by high-to-intermediate? Looking at Figure 3, I’m not sure that I see this. It also depends on what you mean by “Southern Ocean.” Can you be more specific about where you are referring to? Where are those in situ Southern Ocean measurements taken that you want to compare to - closer to 30S or closer to 60S? Where are the high-to-intermediate advective-diffusive fluxes in the Southern Ocean that you’re modeling? I’m also not sure why high advective-diffusive flux would necessarily imply low POC export efficiency. Can you explain why the two should necessarily be correlated or coupled?

Line 283-287: “We, therefore, hypothesize that in addition to the reduction in POC export by strong surface microbial recycling and low grazing-mediated export, the advective-diffusive export of DOC and suspended POC might be an essential feature of high productivity, low POC export regimes.” As noted above, I don’t believe that the results of this paper truly shed light on this or help narrow down which of the 4 hypotheses are true. Furthermore, “high productivity, low POC export regimes” were never well-defined in this paper. I’m still not sure where exactly these are and whether you mean just the Southern Ocean or other places, too. In general, I think this paper would benefit a lot from grouping different regimes/biomes and discussing results from that type of framework, rather than only discussing general patterns (e.g., right now authors often say high lat, mid lat oceans, or similar, which is highly non-specific).

Line 290-292: “Our model results also shed light on the paradox that the organic carbon exported from the surface ocean appears insufficient to support mesopelagic zooplankton and prokaryotic carbon demand.” Though it is great to have an updated estimate of export, I’m not sure that this statement is true. At 100 m, global export is estimated by the authors to be 10.64 Pg C/yr, which is squarely in the range of current estimates (5-12 Pg C/yr), as the authors themselves point out. Why then does this estimate shed any more light than previous studies?

Line 297-398: Same as above. The authors point to 2 places where their calculated export is potentially greater than or equal to community respiration, but then points to 1 place where their calculated export is less than community respiration. This does not support their assertion that their new results really help us better reconcile community respiration and needed export rates. For K2 station, they say other mechanisms may be at play, but that may most definitely be the case at ALOHA and PAP as well.

Line 326-329: "Our results highlight the importance of including the advective-diffusive flux of DOC and suspended POC when estimating the strength of the BCP and motivate the need to improve satellite-based carbon export algorithms so that they better account for export mediated by mixing and other fluid transport." I really like this point and it should be the real takeaway of the paper and the biogeochemical implications section. I don't think the other points (as I noted above) are very well-supported; rather I believe they may be a confusing and unnecessary stretch as currently worded.

Line 330-331: "Our global inversion strongly supports the finding, discovered using neutrally buoyant sediment traps, that POC attenuation is temperature dependent. Geographically, non-advective-diffusive vertical fluxes attenuate faster when surface waters are warmer and penetrate deeper in the water column when surface waters are cold (Fig. S8)." I don't see how the results would support such a strong statement like that in the first sentence. First of all, temperature dependence is built into your formulation of b . Second of all, patterns of b generally matching patterns of upper ocean temperatures does not simply mean that the temperatures caused the patterns (correlation doesn't imply causation). I'm also not even sure how well these patterns match because you don't show that explicitly (Fig. S8 only shows b , not temperature) or do any computation to relate the two variables. Third of all, you are only referring to results from neutrally buoyant sediment traps and only using one paper to say that POC attenuation is temperature dependent. There is a ton of other literature showing that other factors are equally or even more important. Please look into this further. Here are just 2 examples:

Cram, J. A., Weber, T., Leung, S. W., McDonnell, A. M., Liang, J. H., & Deutsch, C. (2018). The role of particle size, ballast, temperature, and oxygen in the sinking flux to the deep sea. *Global Biogeochemical Cycles*, 32(5), 858-876.

Weber, T., Cram, J. A., Leung, S. W., DeVries, T., & Deutsch, C. (2016). Deep ocean nutrients imply large latitudinal variation in particle transfer efficiency. *Proceedings of the National Academy of Sciences*, 113(31), 8606-8611.

Line 338-347: While none of it is totally wrong or unfounded, this entire section makes me a little uneasy because it's ignoring so many other factors and seems to me trying to play up the role of temperature based on a tenuous, unproven relationship (at least within the context of this paper). For example, just because b values are lower in high latitudes, that doesn't necessarily mean a process is more efficient in cold waters than in warm waters when you haven't shown that temperature as opposed to anything else (particle size, ballast, oxygen, nutrient availability, etc.) is really the primary driver. It also does not add anything new to the conversation that many have not already conjectured about before. See Henson et al. (2022) in *Nature Geoscience*, which the authors already cite, for just 1 example.

The second primary concern not addressed is regarding the temperature dependence of b .

I have already begun discussing my concerns about this in the comments for Line 330-331 and Line 338-347 above.

I'd add that I don't believe the authors' response to Reviewer 2's comment about Lines 536-539 was satisfactory. The temperature dependence of b is INDEED an assumption. The authors cite 1 paper to

say why it isn't, but there is plenty of other literature showing that temperature is not the most important (2 papers cited above, for example). Furthermore, a linear relationship is assumed, such that even if temperature dependence of b IS true, a linear relationship does not necessarily have to be. I don't think that this invalidates the methodology by any means, but it is important to add notes to the paper saying that this is an important (maybe? or maybe not?) assumption that your methodology makes. You can say you would test this assumption with future work or you can try to add in different b dependencies (say, on mineral ballast or particle size) and see what happens to your results. I also think the authors need to have a more holistic understanding and citation of this debate on what drives b in general and in the paper (if they want to make claims about what b depends on - otherwise you don't need it).

One last small thing:

It was generally not clear to me what model configurations were presented in the main results. Were they CAFE or CbPM or an average of the 2, etc.? I'd also appreciate maps of the standard deviations between the different model configurations associated with Fig 2a, 3a-d, 4a, at least in the supplementary material. The estimated range for the global value is great, but in general I would like to see more geographical detail.

Concluding thoughts:

This paper is a nice advance in our everlasting quest to better constrain total export rates. The methodology is robust and solid (though a few notes about assumptions need to be added). However, due to the inability to truly separate out mechanistic drivers of export between different regions and therefore the inability to make broader claims of significance for biogeochemical implications, I think it may be better suited for a more specialized journal.

Author Rebuttals to First Revision:

Point-by-point response to editorial suggestions

DATA: Please note that all data have to be made available in a public data repository and the DOI provided prior to publication. Please see our FAIR data in Earth science editorial for more details.

The supporting data to run the inverse model and model outputs have been uploaded to public repositories. We have added the data availability statement in the text as follows,

“Supporting data used to run the inverse model is available at DOI: 10.5281/zenodo.8253967. Model outputs from the inverse model are available at DOI: 10.5281/zenodo.8253973.”

CODE: We strongly encourage that all code is made available in a public repository and the DOI provided.

We have uploaded the code for the inverse model to the following GITHUB repository: at [git@github.com:weileiw/Carbon-Export.git](https://github.com/weileiw/Carbon-Export.git)

LENGTH: Your paper is with 4200 words and five figures around our length limit and should not increase in length during revision. Please keep in mind that important technical details that are not central to the main message of the paper can be moved into the Methods section or, if necessary, a Supplementary Information section (see below).

TITLES: Titles cannot exceed 75 characters (including spaces); they must not contain punctuation and should reflect the main new finding of the paper (not the methodology).

The title now has 75 characters with spaces.

SUMMARY PARAGRAPH: All Nature papers begin with a fully referenced paragraph, typically no longer than 200 words, aimed at readers in other disciplines. This paragraph starts with a 2- to 3-sentence, basic introduction to the field; continues with a 1-sentence statement of the main findings starting 'Here we show' or an equivalent phrase; and finally, concludes with 2 to 3 sentences putting the main findings into general context so it is clear how the results described in the paper have moved the field forward. A downloadable, annotated example is available at <https://www.nature.com/nature/for-authors/formatting-guide>. In some cases it may be necessary to exceed this limit in order to explain complex material for readers in other fields – in such cases, summary paragraphs can be up to 230 words in length. The extra length, however, is for introduction and context, and not for additional technical information.

We have shortened the summary paragraph, and it now has 228 words in length.

MAIN TEXT: If further introductory material is necessary, the main text can begin with up to 500 words of introduction expanding on the background to the work (some overlap with the summary paragraph is acceptable), before proceeding to a concise, focused account of the new research and findings, and ending with 1 or 2 short paragraphs of discussion. Sections are

separated with subheadings to aid navigation. Subheadings may be up to 40 characters (including spaces) and should not be generic (such as 'concluding remarks').

We have two paragraphs of introduction with a total of 404 words.

STATISTICS: Authors should ensure that any statistical analysis used is sound and that it conforms to the journal's guidelines (see <https://www.nature.com/nature/for-authors/formatting-guide> for guidance).

We have checked the statistics and made sure they conform the guidelines.

METHODS: At the end of the main text document (after the main figure legends), there should be a section entitled "Methods", which provides a more detailed discussion of the additional methodological information that would allow other researchers to replicate the results (we define "Methods" quite broadly, so this is not limited to details of experimental protocols – supplementary discussion and analysis can also be included). The Methods section will not appear in the print version but will be fully copy-edited and appear online in the full-text HTML and PDF versions. The Methods section should be written as concisely as possible but should contain all elements necessary to allow interpretation and reproduction of the results. If there are additional references in the Methods section, their numbering should continue from the last reference in the main paper, and the list should follow the Methods section. If the methods require chemical structures, figures or tables, these should be supplied as Extended Data (see below). For mathematically complex methods, or methods that require an unusually large number of figures or tables (beyond what can be accommodated as Extended Data), the entire Methods section should instead be supplied as a separate Supplementary Information.

The METHODS session follows the guidelines.

REFERENCES: As a guideline, Articles allow up to 50 references in the main text; additional references can be cited in (and listed after) the Methods section, as detailed above starting from 51.

We now have a total of 50 references in the main text. Additional references are correctly numbered in following the Method session.

MAIN TEXT STATEMENTS: We require authors to provide a detailed Author Contribution statement immediately after the acknowledgements; the specific contributions of each author must be listed (not all authors are individually listed at present). It is also a condition of publication that authors include an Author Information statement indicating how to access information regarding reprints and permissions, stating whether or not there is a financial or non-financial competing interest, and naming the author to whom correspondence and requests for materials should be addressed. Please ensure that this section is included in the manuscript file after the Methods (but before the Extended Data legends) - it will not appear in the print version but will appear online in the full-text HTML and PDF versions. For details of "end note" style and an example see <https://www.nature.com/nature/for-authors/formatting-guide>.

Author information has been added. Contribution statement has been modified to include all individual authors.

DATA AVAILABILITY STATEMENT: All published manuscripts reporting original research in Nature Portfolio journals must include a data availability statement. The data availability statement must make the conditions of access to the “minimum dataset” that are necessary to interpret, verify and extend the research in the article, transparent to readers. This minimum dataset may be provided through deposition in public community/discipline-specific repositories, custom proprietary repositories for certain types of datasets, or general repositories like Figshare, Zenodo and Dryad. Providing large datasets in supplementary information is strongly discouraged and the preferred approach is to make data available in repositories. More information on Nature Portfolio’s reporting standards and preparing your Data Availability Statement can be found here: <https://www.nature.com/nature-portfolio/editorial-policies/reporting-standards#reporting-requirements>

Model outputs and supporting data have been uploaded to public repositories, and Data availability statement has been changed to include the DOIs.

FIGURE LEGENDS: These should be listed sequentially after the references in the main text and not in the figures files. Each legend should begin with a brief title for the whole figure and continue with a short description of each panel and the symbols used. Any error bars in the figures must be defined (for example, s.d., s.e.m.) and the value of n indicated; see <https://www.nature.com/nature/for-authors/formatting-guide> for further explanation.

Figure legends conform to the guidelines.

DISPLAY ITEMS: We ask that you take stock of all the data that have been generated throughout the review process and ensure that only the data most central to the conclusions are presented in the main figures. Figures should be comprehensible to readers in other or related disciplines, and assist their understanding of the paper. We encourage authors who are describing complex processes to include a schematic of the main finding as part of the Extended Data to aid readers unfamiliar with the immediate discipline. Figures should be as small and simple as is compatible with clarity. All panels of a figure should be logically connected; each panel of a multipart figure should be sized so that the whole figure can be reduced by the same amount and reproduced on the printed page at the smallest size at which essential details are visible. For guidance, Nature’s standard figure sizes are 89 mm (one column), 120 mm (one and a half columns), or exceptionally 183 mm (two columns) wide; the full depth of a Nature page is 247 mm. All panels of figures should be presented on a single page and assembled into a rectangular shape for publication; please indicate any essential alignments (parts horizontal, vertical, spacings of stereo pairs, etc.). Tables should be prepared using the Table menu in Microsoft Word.

Display items are submitted in eps format and can be scaled to fit either 1 column and 2 columns.

FIGURE FORMATTING: Lettering in all figures (labelling of axes and so on) should be in

uniform, sans-serif font, in lower-case type, and large enough to permit substantial reduction for publication (minimum font size 5 pt). Separate parts of a figure are labelled a, b, etc. Units have a single space between the number and the unit, and follow SI nomenclature or the nomenclature common to a particular field. Thousands are separated by commas (1,000). Unusual units or abbreviations are defined in the legend. Scale bars rather than magnification factors should be used.

The figure format conforms to the guidelines.

IMAGE PRESENTATION: Authors should be aware that any image provided for publication, either in print or online (as Extended Data or Supplemental Information), may be subject to a quality control process to check for image integrity and manipulation. For a full discussion of our standards regarding how images should be prepared and presented, see www.nature.com/authors/editorial_policies/image.html.

The manuscript does not include any image.

EXTENDED DATA: Please include your current Supplementary Table/Figures as Extended Data. Nature is now integrating the supplemental figures and tables into the final version of most papers. Extended Data do not appear in the printed version of the paper but are included online within the full-text HTML and at the end of the online PDF. Extended Data are an integral part of the paper and only data that directly contribute to the main message should be presented. All Extended Data must be referred to in the main text, figure legends and/or Methods section, and their figure legends should be listed sequentially at the end of the main text, not in the Extended Data files. Authors should assemble the Extended Data into a maximum of ten, A4 size, multi-panelled display items, submitted as individual JPEG, TIFF or EPS files. They must be provided at the same quality as figures for print, but there are important differences in their formatting. More specific instructions are provided in the Extended Data Formatting Guide (http://s3-service-broker-live-19ea8b98-4d41-4cb4-be4c-d68f4963b7dd.s3.amazonaws.com/uploads/ckeditor/attachments/7823/3h_Extended_data.pdf).

We have transferred the supplementary figures and table to Extended data.

SUPPLEMENTARY INFORMATION: Supplementary Information is online-only material published with the manuscript (<https://www.nature.com/nature/for-authors/supp-info>). For most papers, there should be no need for Supplementary Information beyond that already provided as Methods and Extended Data, the aim being to avoid unnecessary fragmentation of the paper online. Exceptions to this rule include large datasets that cannot be accommodated within Extended Data; video material; and more complex ‘‘Supplemental Methods’’ (and any associated references) that do not readily fit within the constraints of the Methods/Extended Data formats. Please note that after the paper has been formally accepted you can only provide amended Supplementary Information files for critical changes to the scientific content, not for style. You should clearly explain what changes have been made if you do resupply any such files.

There is no supplementary information, since we have transferred the original supplementary figures and table to Extended data.

SOURCE DATA: To further increase transparency, we encourage authors to provide, in spreadsheet form, the data underlying the graphical representations used in the figures. This is in addition to our well-established data-deposition policy for specific types of experiments and large datasets. Readers of the online manuscript will be able to access the Source Data directly from the figure legend. Spreadsheets can only be submitted in .xls, .xlsx or .csv formats. One file per figure is permitted; thus, if there is a multi-panelled figure the Source Data for each panel should be clearly labeled in the csv/Excel file; alternatively, the data for a figure can be included in multiple, clearly labeled sheets within an Excel file. File sizes of up to 30 MB are permitted; however, it is expected that the vast majority of Source Data files will be considerably smaller than this. When submitting these files with your manuscript, please select the file type “Source Data” and use the title field in the file description tab to indicate the figure to which the Source Data pertain.

The source data for each figure is organized in spreadsheet, and has been submitted with the manuscript.

Nature is committed to improving transparency in authorship. As part of our efforts in this direction, we are now requesting that all authors identified as ‘corresponding author’ create and link their Open Researcher and Contributor Identifier (ORCID) with their account on the Manuscript Tracking System prior to acceptance. ORCID helps the scientific community achieve unambiguous attribution of all scholarly contributions. You can create and link your ORCID from the home page of the Manuscript Tracking System by clicking on ‘Modify my Springer Nature account’ and following the instructions in the link below. Please also inform all co-authors that they can add their ORCIDs to their accounts and that they must do so prior to acceptance. If you experience problems in linking your ORCID, please contact the Platform Support Helpdesk.

The corresponding authors have linked to their ORCIDs.

Point-by-point response to Referees' comments

Referees' comments:

Referee #1 (Remarks to the Author):

I appreciated the authors' earnest response and their dedicated effort in addressing a school of potential concerns I raised regarding the model's accuracy and interpretations of its results. Particularly, I am delighted to see that my key suggestion about the inclusion of accounting for the impact of anthropogenic carbon invasion in the DIC dataset has led to a substantial improvement in the model's predictive skill and projections. Also, I am very impressed by the addition of Figure 5 in the revised version's main text, depicting the sequestration-time distribution of multiple carbon export pathways. This figure offers readers a captivating perspective on how these pathways are interconnected with integration depth and their implications for long-term carbon storage.

Overall, I am fully satisfied with the authors' response and would recommend publishing this work in Nature. I believe that this exceptional study will generate significant interest/impact within the scientific community, particularly these in the fields of marine biogeochemistry, paleoclimatology, climate change, Earth system modeling, and marine carbon dioxide removal.

We are deeply appreciative of this referee's constructive suggestions during the first round of review and for the positive assessment of our study and revised manuscript.

Referee #3 (Remarks to the Author):

The authors have addressed all of my previous concerns, and the manuscript is now in an acceptable form for publication.

We thank the reviewer for their constructive suggestions during the first round of review and for the positive assessment of our revised manuscript.

Referee #4 (Remarks to the Author):

The exact same text below is also uploaded as a pdf ("naturereview.pdf") for ease of reading.

(I am a reviewer who is checking that Reviewer 2's suggestions were incorporated.)

This paper is well-written and adds another important new estimate of total export. With the improvements made at the suggestion of the first round of reviewers to account for anthropogenic DIC, the methodology is much more robust now. The added elaboration on why the model should be considered an improvement to Nowicki's is also well explained after revision. The use of both CbPM and CAFE, and move away from Laws, for NPP estimates further improved the paper as well.

We thank the reviewer for their positive comments.

There are 2 primary concerns raised by Reviewer 2 that I don't believe have been fully addressed.

The first primary concern not addressed is regarding the biogeochemical implications, which I will walk through by referring to the author's revised text under the section "Biogeochemical implications."

We have revised the manuscript to take into account the referee's concerns. Below we give a point-by-point response with the reviewers' comments in red, our responses in blue, and the passages from our revised manuscript in black.

Line 267-268: "Our results shed light on the low export efficiency observed, counterintuitively, in high-productivity regions (e.g., the Southern Ocean)." Though the results do indeed calculate export in the Southern Ocean and other high-productivity regions, I don't see how they are able to speak to the reasons for low export efficiency in the Southern Ocean.

What we were trying to say is that for a given net community production, a larger carbon export via an advective-diffusive pathway leaves less organic carbon to be exported via non-advective-diffusive fluxes, which would imply a low export ratio. We have revised the manuscript to clarify our meaning (see lines 300 - 308),

"More importantly, in-situ observations often miss such mixing events because sea-going measurements usually take place during the summer when there is less vertical mixing in the water column. This is a possible reason why POC export ratios determined in situ are negatively correlated with NPP in the Southern Ocean⁸. Indeed, we find that up to 70% of the production is exported via the advective-diffusive pathway in the latitudes between the subtropical and subantarctic fronts (Fig. 3c). The negative correlation between POC export ratio and NPP contradicts the empirical relationships that relate the ef-ratio to temperature and NPP^{3,45} by assuming a positive relationship between NPP and the ef-ratio."

Line 279-280: "They could explain why high ef-ratios are rarely observed in highly productive regions." Are high ef-ratios really rare in productive regions? Or do you mean just in the Southern Ocean? If you mean just the Southern Ocean, please say so. Otherwise please cite.

We meant the Southern Ocean. We have made this clear in the revised manuscript on lines as follows (lines 302 - 306),

"This is a possible reason why POC export ratios determined in situ are negatively correlated with NPP in the Southern Ocean⁸. Indeed, we find that up to 70% of the production is exported via the advective-diffusive pathway in the latitudes between the subtropical and subantarctic fronts (Fig. 3c)."

Line 282-283: "Our high-to-intermediate advective-diffusive organic carbon flux in the Southern Ocean indicates that H1 may also explain the low POC export efficiency." What is meant by high-to-intermediate? Looking at Figure 3, I'm not sure that I see this. It also depends on what

you mean by “Southern Ocean.” Can you be more specific about where you are referring to? Where are those in situ Southern Ocean measurements taken that you want to compare to - closer to 30S or closer to 60S? Where are the high-to-intermediate advective-diffusive fluxes in the Southern Ocean that you’re modeling?

We thank the referee for their comments. The advective-diffusive organic carbon fluxes are plotted in both Fig. 3a and 3b, the former for the labile DOC and the latter for semi-labile DOC. They are also plotted in Extended Data Fig. 7 (a-f) for different model configurations. The ratio of advective-diffusive flux to total organic carbon flux is illustrated in Fig. 3c and Extended Fig. 7 (m-o). Figure 3c shows that the ratio of advective-diffusive flux to total organic carbon flux is high south of 30S and Figure 3b shows that the export by advective-diffusive fluxes is significant south of 40S, corresponding to the regions where the observed low ef-ratios were documented¹.

I’m also not sure why high advective-diffusive flux would necessarily imply low POC export efficiency. Can you explain why the two should necessarily be correlated or coupled?

As mentioned in our response to the referee’s previous comment, what we were trying to say is simply that for a given net community production, a larger carbon export via an advective-diffusive pathway leaves less organic carbon to be exported via non-advective-diffusive fluxes, which would imply a low export ratio. We have made this clear in the revised manuscript as follows (see lines 300 - 308),

“More importantly, in-situ observations often miss such mixing events because sea-going measurements usually take place during the summer when there is less vertical mixing in the water column. This is a possible reason why POC export ratios determined in situ are negatively correlated with NPP in the Southern Ocean⁸. Indeed, we find that up to 70% of the production is exported via the advective-diffusive pathway in the latitudes between the subtropical and subantarctic fronts (Fig. 3c). The negative correlation between POC export ratio and NPP contradicts the empirical relationships that relate the ef-ratio to temperature and NPP^{3,45} by assuming a positive relationship between NPP and the ef-ratio.”

Line 283-287: “We, therefore, hypothesize that in addition to the reduction in POC export by strong surface microbial recycling and low grazing-mediated export, the advective-diffusive export of DOC and suspended POC might be an essential feature of high productivity, low POC export regimes.” As noted above, I don’t believe that the results of this paper truly shed light on this or help narrow down which of the 4 hypotheses are true. Furthermore, “high productivity, low POC export regimes” were never well-defined in this paper. I’m still not sure where exactly these are and whether you mean just the Southern Ocean or other places, too. In general, I think this paper would benefit a lot from grouping different regimes/biomes and discussing results from that type of framework, rather than only discussing general patterns (e.g., right now authors often say high lat, mid lat oceans, or similar, which is highly non-specific).

The review is correct that the discussion of different hypotheses to explain the low export ratio in highly productive regions was inadequate and that the conclusions we can draw from our inverse modeling results are rather limited. We have, therefore, removed much of this discussion, to focus on what the reviewer rightly calls the paper’s ‘true’ take-home message (the importance of

advective-diffusive flux). Removing this discussion has also allowed us to keep the manuscript within the editor's recommended length.

Line 290-292: "Our model results also shed light on the paradox that the organic carbon exported from the surface ocean appears insufficient to support mesopelagic zooplankton and prokaryotic carbon demand." Though it is great to have an updated estimate of export, I'm not sure that this statement is true. At 100 m, global export is estimated by the authors to be 10.64 Pg C/yr, which is squarely in the range of current estimates (5-12 Pg C/yr), as the authors themselves point out. Why then does this estimate shed any more light than previous studies?

While our integrated total falls within the scope of previous estimates, our geographic distribution differs from these past assessments. Notably, a comprehensive estimation of carbon demand in the deep ocean has been lacking on a global scale, having only been determined at select oceanic stations at specific seasons. It's essential to underline that our model is constrained simultaneously by several full water-column tracers, particularly DIC, ALK, DOC, and O₂. As a result, it encapsulates the balance between carbon export and the deep-ocean carbon demand.

Line 297-398: Same as above. The authors point to 2 places where their calculated export is potentially greater than or equal to community respiration, but then points to 1 place where their calculated export is less than community respiration. This does not support their assertion that their new results really help us better reconcile community respiration and needed export rates. For K2 station, they say other mechanisms may be at play, but that may most definitely be the case at ALOHA and PAP as well.

We thank the referee for pointing this out. In the revised manuscript, rather than delving into explanations for the precise mechanisms contributing to budget balance at each site, we've highlighted, more generally, that the divergence between export and consumption could arise from overlooking certain export pathways or seasonal variations. Our revised discussion focuses instead on the fact that our model results are able to fit the observations with a model built on a balance between production and respiration (lines 234 - 255):

"Budgets based on in situ observations often struggle to establish a balance between community production and respiration (e.g., Ref^{37,38}), either because they fail to account for all processes that deliver organic carbon to the mesopelagic ocean or because they are limited to measurements during a specific season. Our model, which represents an annual-mean balance between community production and respiration, is able to simultaneously fit full water-column observations of DIC, DOC, ALK, and O₂ showing that there is no difficulty in closing the budget provided one accounts for both advective-diffusive and non-advective-diffusive export pathways. At the Porcupine Abyssal Plain (PAP) site in the North Atlantic Ocean, our TOC export ($201.5 \pm 29.4 \text{ mg C m}^{-2} \text{ d}^{-1}$ between 73-1000 m) exceeds the in-situ community respiration ($48\text{-}167 \text{ mg C m}^{-2} \text{ d}^{-1}$ between 50-1000 m) measured in the summer season³⁹ when NCP is relatively low³⁸. At station ALOHA, our annual TOC flux between mMLD and 1000 m ($45.1 \pm 4.0 \text{ mg C m}^{-2} \text{ d}^{-1}$) overlaps with in situ measurements of heterotrophic respiration rates between 150-1000 m ($32.5\text{-}96.6 \text{ mg C m}^{-2} \text{ d}^{-1}$)³⁷. However, at the Japanese time-series site K2 station, also in the Pacific, our TOC flux between mMLD and 1000 m ($82.1 \pm 2.4 \text{ mg C m}^{-2} \text{ d}^{-1}$) falls short of the lower end of in situ determinations ($106.1\text{-}249.8 \text{ mg C m}^{-2} \text{ d}^{-1}$)³⁷ at the depth interval of 150-1000

m. Such disparities could potentially arise because our model represents an annual mean, while the in-situ measurements were conducted during specific seasons. Future development of a seasonal inverse model could contribute to narrowing this difference. The disparities might also be influenced by the inherent uncertainties associated with in situ measurements. In light of these potential factors, we advocate for an increased number of in situ observations focused on year-round whole community carbon demand within the twilight zone.”

Line 326-329: “Our results highlight the importance of including the advective-diffusive flux of DOC and suspended POC when estimating the strength of the BCP and motivate the need to improve satellite-based carbon export algorithms so that they better account for export mediated by mixing and other fluid transport.” I really like this point and it should be the real takeaway of the paper and the biogeochemical implications section. I don’t think the other points (as I noted above) are very well-supported; rather I believe they may be a confusing and unnecessary stretch as currently worded.

We thank the reviewer for the positive comments and suggestion of reducing the discussion of other implications. We have therefore combined this section with the discussion of low-export-efficiency in high-productive Southern Oceans. We have removed the hypotheses explaining the phenomenon but stating that missing the advective-diffusive flux could be a potential reason for the observed low-export-efficiency in high-productive Southern Oceans.

Line 330-331: “Our global inversion strongly supports the finding, discovered using neutrally buoyant sediment traps, that POC attenuation is temperature dependent. Geographically, non-advective-diffusive vertical fluxes attenuate faster when surface waters are warmer and penetrate deeper in the water column when surface waters are cold (Fig. S8).” I don’t see how the results would support such a strong statement like that in the first sentence. First of all, temperature dependence is built into your formulation of b . Second of all, patterns of b generally matching patterns of upper ocean temperatures does not simply mean that the temperatures caused the patterns (correlation doesn’t imply causation). I’m also not even sure how well these patterns match because you don’t show that explicitly (Fig. S8 only shows b , not temperature) or do any computation to relate the two variables. Third of all, you are only referring to results from neutrally buoyant sediment traps and only using one paper to say that POC attenuation is temperature dependent. There is a ton of other literature showing that other factors are equally or even more important. Please look into this further. Here are just 2 examples:

Cram, J. A., Weber, T., Leung, S. W., McDonnell, A. M., Liang, J. H., & Deutsch, C. (2018). The role of particle size, ballast, temperature, and oxygen in the sinking flux to the deep sea. *Global Biogeochemical Cycles*, 32(5), 858-876.

Weber, T., Cram, J. A., Leung, S. W., DeVries, T., & Deutsch, C. (2016). Deep ocean nutrients imply large latitudinal variation in particle transfer efficiency. *Proceedings of the National Academy of Sciences*, 113(31), 8606-8611.

We thank the reviewer for these important comments. We have revised the manuscript to make it clear that while we choose to make the b values linear functions of temperature, the parameter values are not prescribed, but inferred from the hydrographic data through our inverse model. In the revised manuscript we now write (lines 256 - 269):

“Numerous mechanisms have been proposed to explain the spatial variations of carbon flux, with prominent factors including particle size and sinking velocities, community structure, remineralization dependence on temperature and oxygen, and ballast effect^{40,41}. Earth system models that incorporate these mechanisms in varying degrees exhibit a wide range of carbon flux ($\sim 5 - 12 \text{ Pg C yr}^{-1}$)²⁶ and have clearly identifiable biases in their simulated oxygen and carbon distributions. In contrast, our inverse model avoids overparameterization, by not including explicit representations of each of these processes. Nevertheless, it provides a good fit to the tracer data with a simple temperature-dependent parameterization for the remineralization of organic carbon. Specifically, our model adopts a powerlaw parameterization with a temperature dependent exponent $b = b_{c\theta}T + b_c$ for non-advective-diffusive carbon fluxes (See Methods). Our inversion infers a temperature dependence, $b_{c\theta} = 0.03 \text{ }^\circ\text{C}^{-1}$ (Extended Data Table 1) that is approximately 50% smaller than the value estimated using a limited sediment trap dataset of POC fluxes⁴², but is otherwise in agreement with the sign of the temperature effect.”

We also added to the discussion to make it clear that the temperature dependence in our inferred b values can be viewed as a proxy for other mechanisms that correlate with temperature (lines 269 - 283):

“Geographically, non-advective-diffusive vertical fluxes attenuate faster when surface waters are warmer and penetrate deeper in the water column when surface waters are cold (Extended Data Fig. 8). Notably, our non-advective-diffusive vertical flux includes not only the classical gravitational POC flux but also any fluxes with significant non-advective-diffusive vertical transport such as fluxes related to seasonal lipid pump²¹ and zooplankton migration pump²⁰. It's also worth noting that in high-latitude low-temperature oceans, the prevalence of large phytoplankton with ballast shells and shorter food webs promotes non-advective-diffusive vertical fluxes. Conversely, in warm subtropical gyres, the prevalence of small phytoplankton and longer food webs reduces non-advective-diffusive flux^{40,41}. The deeper penetration in higher latitudes, coupled with an overall lower temperature dependence compared to the trap-derived value ($0.03 \text{ }^\circ\text{C}^{-1}$ this study vs. $0.062 \text{ }^\circ\text{C}^{-1}$ Ref.⁴²) underscores the intricate interplay of different mechanisms. In our inverse model, the dependence of the powerlaw exponent on temperature, serves as a proxy for any mechanism that correlates with surface temperature. Future research will need to unravel these mechanisms.”

Line 338-347: While none of it is totally wrong or unfounded, this entire section makes me a little uneasy because it's ignoring so many other factors and seems to me trying to play up the role of temperature based on a tenuous, unproven relationship (at least within the context of this paper). For example, just because b values are lower in high latitudes, that doesn't necessarily mean a process is more efficient in cold waters than in warm waters when you haven't shown that temperature as opposed to anything else (particle size, ballast, oxygen, nutrient availability, etc.) is really the primary driver. It also does not add anything new to the conversation that many have not already conjectured about before. See Henson et al. (2022) in Nature Geoscience, which the authors already cite, for just 1 example.

We hope the reviewer will find that our revised discussion of the role of temperature better reflects what can be concluded from our inversion.

The second primary concern not addressed is regarding the temperature dependence of b .

I have already begun discussing my concerns about this in the comments for Line 330-331 and Line 338-347 above.

I'd add that I don't believe the authors' response to Reviewer 2's comment about Lines 536-539 was satisfactory. The temperature dependence of b is INDEED an assumption. The authors cite 1 paper to say why it isn't, but there is plenty of other literature showing that temperature is not the most important (2 papers cited above, for example). Furthermore, a linear relationship is assumed, such that even if temperature dependence of b IS true, a linear relationship does not necessarily have to be. I don't think that this invalidates the methodology by any means, but it is important to add notes to the paper saying that this is an important (maybe? or maybe not?) assumption that your methodology makes. You can say you would test this assumption with future work or you can try to add in different b dependencies (say, on mineral ballast or particle size) and see what happens to your results. I also think the authors need to have a more holistic understanding and citation of this debate on what drives b in general and in the paper (if they want to make claims about what b depends on - otherwise you don't need it).

We are grateful for the reviewer's comments because they have made us aware that our previous presentation could be misinterpreted. In the previous iteration of our manuscript, we cited the work of Marsay et al. (2015) because it used the same functional form for the parameterization of b in terms of temperature that we use in our inverse model. It was not meant imply that temperature was the driving mechanism for the variations in b . The reviewer is correct that correlation does not imply causation and that while we do not exclude the possibility that there is a mechanistic relationship between temperature and the vertical transfer of carbon, our model does not require it. Moreover, the fact that we use a linear relationship does not imply that the 'true' relationship, assuming there is one, is necessarily linear because we can interpret our coefficients as simply being related to the coefficients in a Taylor expansion of a more general relationship, i.e. $b = f(T) \rightarrow b \approx f(T_0) + f'(T_0)(T - T_0) \rightarrow b_{C\theta} = f'(T_0)$ and $b_C = f(T_0) - f'(T_0)T_0$.

The relationship between b and temperature can implicitly incorporate other mechanisms, and temperature may serve as a useful proxy of other mechanisms. We have added the following discussion in the text (lines 271 - 283).

“Notably, our non-advective-diffusive vertical flux includes not only the classical gravitational POC flux but also any fluxes with significant non-advective-diffusive vertical transport such as fluxes related to seasonal lipid pump²¹ and zooplankton migration pump²⁰. It's also worth noting that in high-latitude low-temperature oceans, the prevalence of large phytoplankton with ballast shells and shorter food webs promotes non-advective-diffusive vertical fluxes. Conversely, in warm subtropical gyres, the prevalence of small phytoplankton and longer food webs reduces non-advective-diffusive flux^{40,41}. The deeper penetration in higher latitudes, coupled with an overall lower temperature dependence compared to the trap-derived value ($0.03 \text{ }^\circ\text{C}^{-1}$ this study vs. $0.062 \text{ }^\circ\text{C}^{-1}$ Ref.⁴²) underscores the intricate interplay of different mechanisms. In our inverse model, the dependence of the powerlaw exponent on temperature, serves as a proxy for any

mechanism that correlates with surface temperature. Future research will need to unravel these mechanisms,”

One last small thing:

It was generally not clear to me what model configurations were presented in the main results. Were they CAFE or CbPM or an average of the 2, etc.? I'd also appreciate maps of the standard deviations between the different model configurations associated with Fig 2a, 3a-d, 4a, at least in the supplementary material. The estimated range for the global value is great, but in general I would like to see more geographical detail.

We thank the referee for their suggestion. We have clearly indicated the model configuration in figure captions. We have added “The results are based on the CbPM NPP product and an e-folding remineralization time of 12 hours for labile DOC.” at the end of caption of each figure in the main text.

We also plotted maps of standard deviations between different model configurations. The new figure is presented as Extended Data Fig. 9, and reproduced below.

Extended Data Fig. 9 | Distributions of standard deviations (STD). a) STD of total organic carbon flux, b) STD of non-advective-diffusive flux, c) STD of advective-diffusive flux by labile DOC, d) STD of advective-diffusive flux by semi-labile DOC, e) STD of the ratio of advective-

diffusive flux to total organic carbon flux, and f) STD of DOC residence time in years. These standard deviations are computed from four distinct model configurations, which hinge on two distinct NPP (Net Primary Productivity) products, namely CbPM and CAFE, along with two varying e-folding remineralization time scales for labile DOC, specifically 12 and 24 hours.

Concluding thoughts:

This paper is a nice advance in our everlasting quest to better constrain total export rates. The methodology is robust and solid (though a few notes about assumptions need to be added). However, due to the inability to truly separate out mechanistic drivers of export between different regions and therefore the inability to make broader claims of significance for biogeochemical implications, I think it may be better suited for a more specialized journal.

We thank the review for their positive comments.

The strength of the inverse modeling approach we have taken is that it is agnostic to the detailed mechanistic processes that are responsible for our inferred export patterns. The biogeochemical implications we discussed in the paper were not meant to be definitive — we only wanted to give examples of how our results could be used to evaluate the relative consistency of the various hypotheses about carbon export mechanisms.

The biggest breakthrough of our research is that by combining multiple tracer constraints our inverse model is able to separate the effect the biological pump from those of the anthropogenic DIC invasion on the DIC fields, thus paving the way for a marine carbon-cycle data assimilation system that includes the direct assimilation of DIC measurements to infer potential changes in the strength of the biological pump superimposed on the anthropogenic carbon signal.

Reviewer Reports on the First Revision:

Referees' comments:

Referee #4 (Remarks to the Author):

The authors have thoroughly addressed my comments and the paper should now be published.
Great job!

Author Rebuttals to Second Revision:

Reply to referees' comments:

Referees' comments:

Referee #4 (Remarks to the Author):

The authors have thoroughly addressed my comments and the paper should now be published. Great job!

We thank the reviewer for their constructive suggestions during previous round of review and for the positive assessment of our revised manuscript.